# 70 km long-range Raman distributed optical fibre sensing through enhanced anti-distortion coding and waveform reconstruction

Fan Zhang [1,2], Jian Li [1,2,3] ✉, Lulei Li [1,2], Kangyi Cao [1,2] & Mingjiang Zhang [1,2,3] ✉

The practical implementation of Raman distributed optical fibre sensing has been fundamentally constrained by the inherent low signal-to-noise ratio (SNR), particularly for sensing distance exceeding 30 km. We propose a paradigm that combines enhanced anti-distortion coding processing, advanced Raman scattering waveform reconstruction preprocessing, and Haar wavelet denoising to transcend this physical limitation. The proposed pre-processing framework simultaneously optimises complementary sequences correlation, effectively mitigates disturbances of transient effects and improves sensing performance. The experimental demonstration achieves performance metrics: 70.0 km sensing distance with 1.58 m spatial resolution, while maintaining 0.91 °C measurement accuracy and 5.39 °C temperature resolution. The tripartite synergistic mechanism, consisting of the waveform reconstruction pre-processing framework compensating for and suppressing transient effects, coding gain improving the baseline SNR, and the Haar wavelet transform removing residual noise, breaks through the theoretical trade-off constraint between SNR and sensing distance in traditional schemes. The proposed approach demonstrates a potential for application in the fields of long-range infrastructure monitoring and environmental sensing.

In 1928, physicist Raman C.V. published a paper titled A type of secondary radiation in *Nature*[1], and since then, the phenomenon of Raman scattering has been extensively and intensively studied. Over the past decades, Raman distributed optical fibre sensing has emerged as a research frontier[2–7], distinguished by its fast measurement speed, simple structure and inherent temperature sensitivity[8–11]. It has been widely applied in safety monitoring of infrastructure systems, including transportation networks, power transmission facilities and large-scale linear installations, owing to its unique capability for continuous temperature measurement along the entire length of the sensing fibre[12–16]. However, in contrast to Rayleigh and Brillouin scattering light[17], Raman scattering light exhibits lower intensity characteristics. Since the Raman scattering signal is 50 dB weaker than the incident detection signal[18,19], the Raman distributed optical fibre sensing has the technical bottleneck of lower SNR, which seriously hampers the advancement of long-sensing-distance applications. While increasing the pulse width effectively enhances the SNR of Raman scattering signal[20], this improvement occurs at the expense of degraded spatial resolution which creates a technical bottleneck that cannot balance the sensing distance and spatial resolution.

[1]College of Physics and Optoelectronics, Taiyuan University of Technology, Taiyuan, China. [2]Key Laboratory of Advanced Transducers and Intelligent Control Systems (Ministry of Education and Shanxi Province), Taiyuan University of Technology, Taiyuan, China. [3]Shanxi Key Laboratory of Precision Measurement Physics, Taiyuan University of Technology, Taiyuan, China. ✉e-mail: lijian02@tyut.edu.cn; zhangmingjiang@tyut.edu.cn

To address these challenges and increase the sensing distance, advanced SNR optimisation schemes, including algorithmic denoising schemes[21–24], neural network schemes[25,26] and pulse coding schemes[27–29], have been implemented in Raman distributed optical fibre sensing.

Algorithm denoising schemes aim to improve the SNR of the system by processing optical fibre scattering signals. For example, Yu et al. proposed a dynamic sampling-correction scheme achieving a sensing distance of 20.0 km[23]. Pradhan et al. proposed a Fourier wavelet regularised deconvolution scheme achieving a sensing distance of 30.0 km with 3.0 m spatial resolution[24]. Essentially derived from mathematical optimisation theory, such schemes only denoise signals from the perspective of signal processing without achieving breakthroughs in physical mechanisms. Furthermore, they have limitations, including parameter dependence and difficulty in preserving the true characteristics of signals[30].

In addition, as one of the current mainstream denoising methods, convolutional neural network scheme is a typical data-driven approaches. It optimises model parameters using large-scale training datasets, with the goal of achieving effective processing of test data. For example, Zhang et al. proposed a deep 1-D denoising convolutional neural network scheme, which realised a sensing distance of 10.0 km with a spatial resolution of 3.0 m[26]. However, the performance of this method is significantly constrained by the specificity of training data. When there is a distribution difference between test signals and the training set, its effect on enhancing sensing performance will be limited.

Pulse coding schemes have demonstrated significant advantages in long-distance Raman distributed optical fibre sensing. On one hand, by encoding the detection optical pulses, this technology can significantly increase the optical power injected into the sensing fibre while avoiding fibre nonlinear effects, thereby enhancing the intensity of effective sensing signals at the level of signal generation mechanism. On the other hand, at the data processing level, it leverages the separability between encoded signals and uncoded noise to effectively suppress noise and improve the SNR of system.

Currently, five typical pulse coding schemes are widely applied in Raman distributed optical fibre sensing, which specifically include the simplex coding scheme[31], low-repetition-rate cyclic pulse coding scheme[27], pre-shaped simplex coding scheme[32], genetic-optimised aperiodic coding scheme[29] and derived sequences coding scheme[33]. For example, Park et al. achieved 37.0 km sensing distance, 17.0 m spatial resolution, 3.0 °C temperature resolution and 2176 effective sensing points via simplex coding scheme[31]. Soto et al. achieved 26.0 km sensing distance, 1.0 m spatial resolution, 3.0 °C temperature resolution and 26,000 effective sensing points with a low-repetition-rate cyclic pulse coding scheme[27]. Rosolem et al. achieved 62.0 km sensing distance, 10.0 m spatial resolution, 8.4 °C temperature resolution and 6200 effective sensing points using a pre-shaped simplex coding scheme[32]. Sun et al. achieved 39.0 km sensing distance, 1.0 m spatial resolution, 3.9 °C temperature resolution and 39,000 effective sensing points based on a genetic-optimised aperiodic coding scheme[29]. Chai et al. achieved 44.0 km sensing distance, 5.0 m spatial resolution, 2.5 °C temperature resolution and 8800 effective sensing points with a derived sequences coding scheme[33].

However, existing pulse coding schemes in Raman distributed optical fibre sensing remain constrained by several prevalent bottlenecks: high hardware implementation complexity, excessive computational overhead in encoding and decoding, limited non-linear effect suppression capability, and inadequate resistance to signal distortion coupled with poor stability during long-distance transmission. Collectively, these factors constrain the system's application potential in high-performance scenarios, including low SNR environments and long-distance sensing applications.

To address the aforementioned limitations, this paper proposes an enhanced anti-distortion coding (EAD-coding) and waveform reconstruction scheme. Based on a tripartite synergistic mechanism, the proposed scheme effectively alleviates the conventional trade-offs among SNR, sensing distance and hardware cost. This mechanism integrates three key components: a pre-processing framework that compensates for transient effects in erbium-doped fibre amplifiers (EDFA), an encoding strategy that enhances the SNR of sensing signals through coding gain, and a refined residual noise suppression process based on the Haar wavelet transform. Experimental results demonstrate that the proposed scheme achieves a sensing distance of 70.0 km, a spatial resolution of 1.58 m, a temperature resolution of 5.39 °C, and a temperature measurement accuracy of 0.91 °C, with the number of effective sensing points reaching 44,303. To the best of our knowledge, this value represents the highest number of effective sensing points reported to date in the field of Raman distributed optical fibre sensing. Furthermore, while maintaining relatively low complexity, the EAD-coding scheme improves the system's anti-distortion capability and operational adaptability, offering a solution for long-distance Raman distributed fibre sensing system.

## Results

### Experimental setup

We built a Raman distributed optical fibre sensing experimental setup based on proposed EAD-coding and waveform reconstruction scheme, as shown in Fig. 1. Firstly, the laser device generates continuous laser with a central wavelength of 1550 nm, and then the encoded detection signals based on Golay complementary sequences are generated by using the digital delayed pulse generator (DDPG) to modulate the semiconductor optical amplifier (SOA). Each detection signal is amplified by EDFA and then enters a single-mode sensing fibre through a wavelength division multiplexer (WDM), and generates Raman backscattered signal (RBS) which enters an avalanche photodetector (APD) for amplification and optoelectronic conversion. Finally, all RBSs generated by detection signals are collected by a data acquisition card (DAC) and demodulated in a computer to obtain the distributed temperature information.

In this study, Golay complementary sequences are applied in Raman distributed optical fibre sensing, and an enhanced anti-distortion coding scheme is developed based on these sequences. Fig. 2 presents the flowchart of the EAD-coding scheme, detailing its encoding and decoding processes, demodulation and localisation principles and signal waveform reconstruction framework.

Figure 2a shows the encoding principle. Its main function is to produce four complementary optical detection signals after convolu-

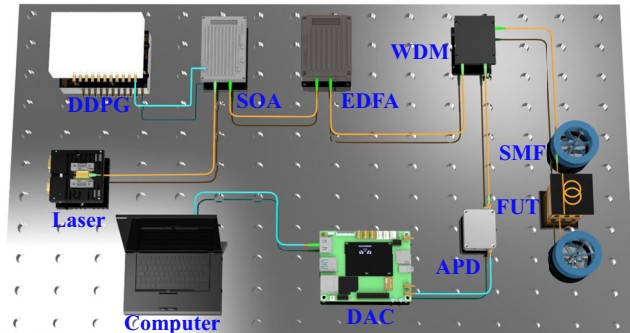

**Fig. 1 | Experimental setup based on EAD-coding and waveform reconstruction scheme for Raman distributed optical fibre sensing.** DDPG Digital delayed pulse generator, SOA Semiconductor optical amplifier, EDFA Erbium-doped fibre amplifier, WDM Wavelength division multiplexer, APD Avalanche photodetector, DAC Data acquisition card, SMF Single-mode fibre, FUT Fibre under test.

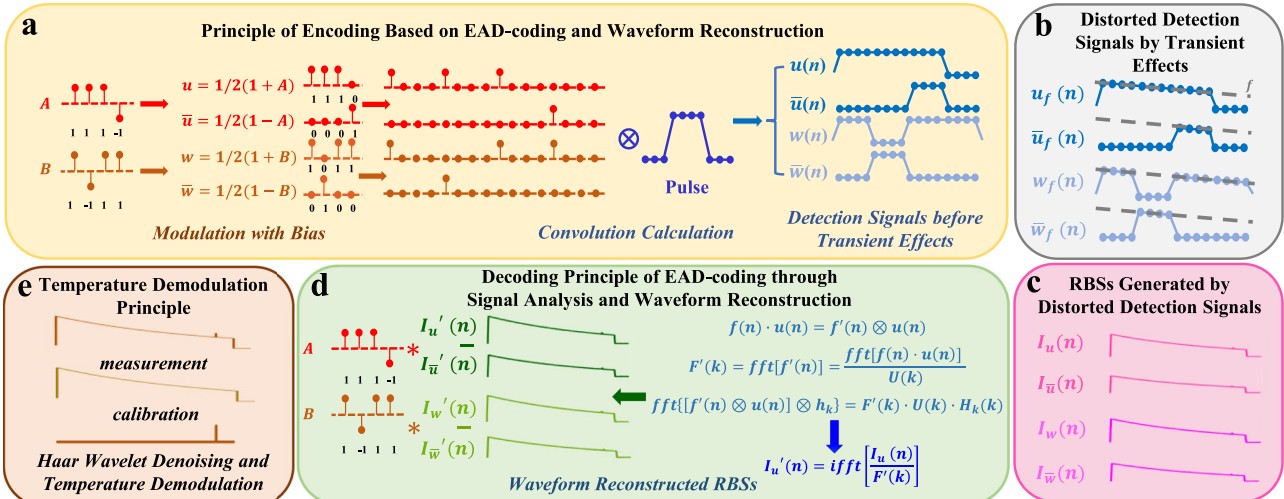

**Fig. 2 | Physics principle of EAD-coding and waveform reconstruction scheme for Raman distributed optical fibre sensing. a** Principle of encoding based on EAD-coding and waveform reconstruction. **b** Influence of transient effects on detection signals. **c** RBSs generated by four distorted detection signals in sensing fibre. **d** Decoding principle of EAD-coding through signal analysis and waveform reconstruction. **e** Temperature demodulation principle.

tion with a single-pulse signal. After being amplified by the EDFA, the waveforms of these four channels of detection signals will undergo distortion, as shown in Fig. 2b. Then these distorted detection signals enter the sensing fibre and generate RBSs, as shown in Fig. 2c. These RBSs will be analyzed and reconstructed to recover the autocorrelation in process of decoding, as shown in Fig. 2d. Finally, the distributed temperature information is demodulated, as shown in Fig. 2e.

The detailed principles of the pre-processing procedure are presented in the 'Method' section, and the specific principles of the EAD-coding and waveform reconstruction scheme are described in the Supplementary Information.

## Impact of transient effects on the detection signals and RBSs

In Raman distributed optical fibre sensing, the transient effects impact the sensing performance (SNR and temperature accuracy). Transient gain fluctuations in an EDFA arise because the relaxation process of the population inversion cannot equilibrate instantaneously with rapid changes in the input laser power. This phenomenon will lead to a non-uniform amplification that the detection signals will distort in timing when passing through the EDFA, thereby disrupting the autocorrelation of the detection signals and ultimately limiting the system's sensing performance.

The proposed scheme generates four sets of correlated detection signals ($u(n)$, $\bar{u}(n)$, $w(n)$, $\bar{w}(n)$, the $n$ denotes the discrete point) based on the EAD-coding approach. In the experiment, we first characterise the detection signals and evaluate the influence of transient effects. These detection signals are acquired before and after the EDFA for timing analysis, as shown in Fig. 3a–d and all curves are normalised to ensure consistency in comparative analysis. The yellow curves are the detection signals affected by transient effects, and the blue curves are the detection signals that are not affected by transient effects. It can be clearly seen that the transient effects cause the gain of EDFA to be non-uniform. These detection signals show a gradual decay trend in the timing after passing through EDFA, which results in a weakening of autocorrelation. Ultimately, the distortion in the detection signals adversely affects the overall sensing performance of the system.

To validate the impact of transient effects on autocorrelation characteristics, we conducted an analysis experiment of the auto-correlation curves before and after the transient effects. The blue curve in Fig. 3e is the autocorrelation function before transient effects. As clearly shown, the autocorrelation curve exhibits a standard unit impulse function (Dirac delta function) profile with no discernible

sidelobe noise adjacent to the main peak, demonstrating a better autocorrelation characteristic. As shown in Fig. 3f, pronounced side-lobe noise emerges on both sides of the autocorrelation main peak, evidencing degradation of autocorrelation characteristics under transient effects.

The degradation of autocorrelation characteristics induced by transient effects significantly compromises the system's sensing performance. To systematically investigate the impact of transient effects on sensing performance, we conducted a quantitative analysis experiment on both the RBSs generated from four detection signals after transient effects and their corresponding decoding results. We also conducted a comparative analysis experiment of the waveform reconstructed RBSs and their corresponding decoded results based on EAD-coding and the waveform reconstruction scheme. Quantitative analysis in Fig. 4 reveals that transient effects will lead to the weakening of the RBSs intensity generated by these distorted detection signals, as shown by the yellow curves in Fig. 4a–d, thereby decreasing SNR of the system. The waveform reconstruction (pre-processing) effectively suppresses transient effects, yielding an average SNR enhancement of 1.11 dB (the average SNR improvement of the RBSs generated by the aforementioned four detection signals) as depicted by the blue curves. Theoretically, the greater the influence of transient effects, the more pronounced the intensity enhancement of RBSs in the pre-processing within the EAD-coding scheme. In addition, all curves are normalised to ensure comparative analysis consistency.

Transient effects not only reduce the intensity of RBSs generated by these detection signals but also lead to the weakening of the decoded RBS, as shown in Fig. 4e. The decoded results prior to and following pre-processing are depicted by the yellow and blue curves, respectively. The result demonstrates that the decoded RBS intensity after pre-processing exhibits a higher magnitude compared to the intensity before pre-processing, thereby confirming a suppression of transient effects. To investigate the impact of transient effects on decoding performance, we systematically calculated the corresponding SNR. Due to the detrimental impact of transient effects on decoding fidelity, the reduced intensity of decoded RBSs consequently degrades the SNR. The EAD-coding and waveform reconstruction scheme effectively suppresses transient perturbations, restoring the decoded RBS intensity and thereby enhancing SNR by 1.45 dB, which experimentally validates the effectiveness of the EAD-coding and waveform reconstruction scheme.

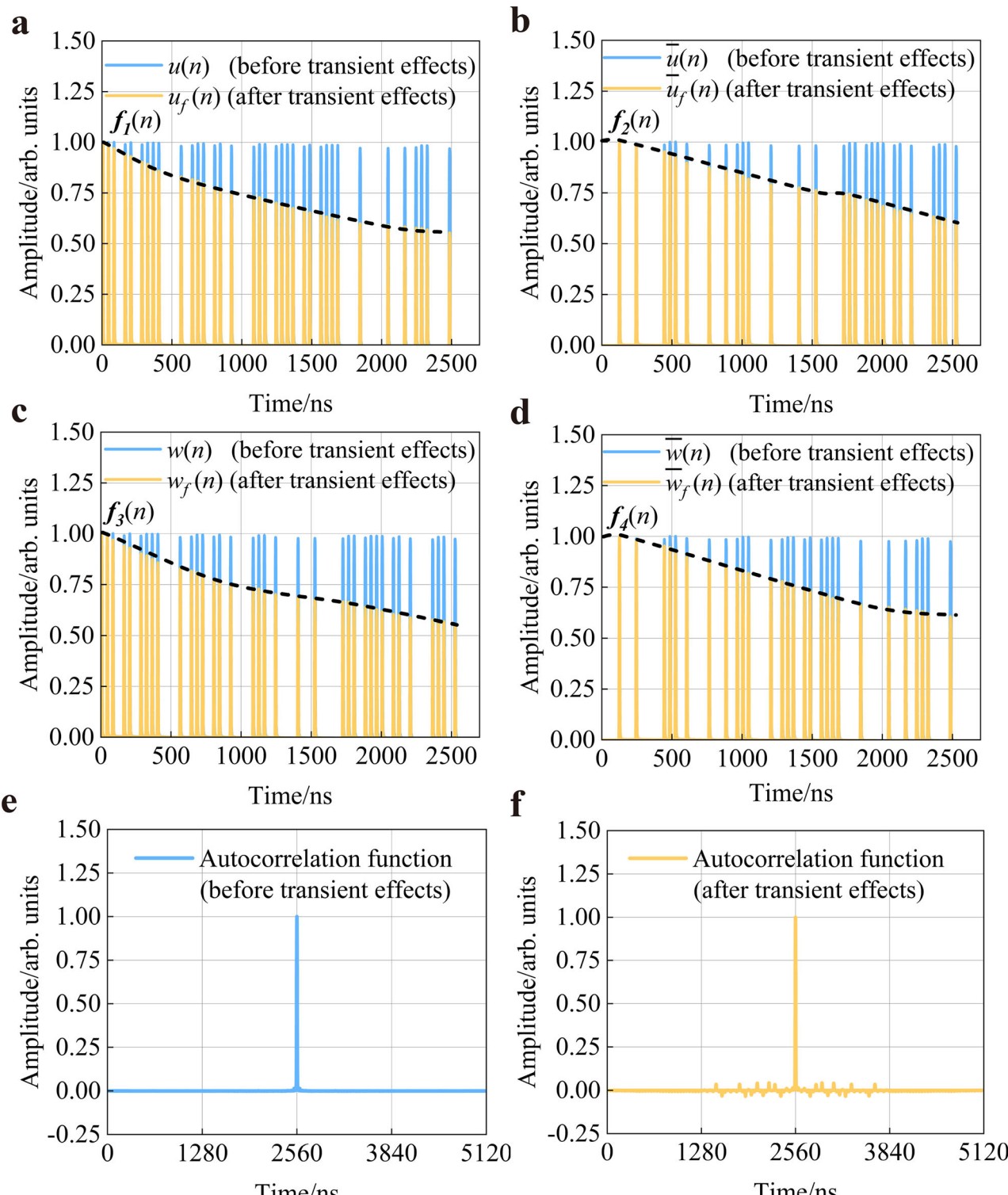

**Fig. 3 | Timing of detection signals and autocorrelation function curves.** Timing of four detection signals [(**a**) detection signal $u(n)$, **b** detection signal $\bar{u}(n)$, **c** detection signal $w(n)$ and **d** detection signal $\bar{w}(n)$] before and after transient effects. Autocorrelation function curve of the detection signals (**e**) before transient effects and **f** after transient effects. Among them, each detection signal ($u(n)$, $\bar{u}(n)$, $w(n)$, $\bar{w}(n)$), after the transient effects, is denoted as $u_f(n)$, $\bar{u}_f(n)$, $w_f(n)$, $\bar{w}_f(n)$. The attenuation envelopes of each detected signal is described by its corresponding envelope function, $f_1(n)$, $f_2(n)$, $f_3(n)$, $f_4(n)$.

Furthermore, an autocorrelation analysis is performed on the pre-processed detection signals coupled into the sensing fibre to confirm their efficacy in eliminating transient effects. The primary objective of this experiment is to verify that the waveform reconstructed RBSs are equivalent to the RBSs of the undistorted detection signals. Taking the detection signal $u(n)$ as an example, the detection signal coupled into the sensing fibre after waveform reconstruction can be calculated and is given by Eq. (1).

$$u'(n) = \text{ifft}\left(\frac{\text{fft}[u_f(n)]}{\text{fft}[f'_1(n)]}\right) \tag{1}$$

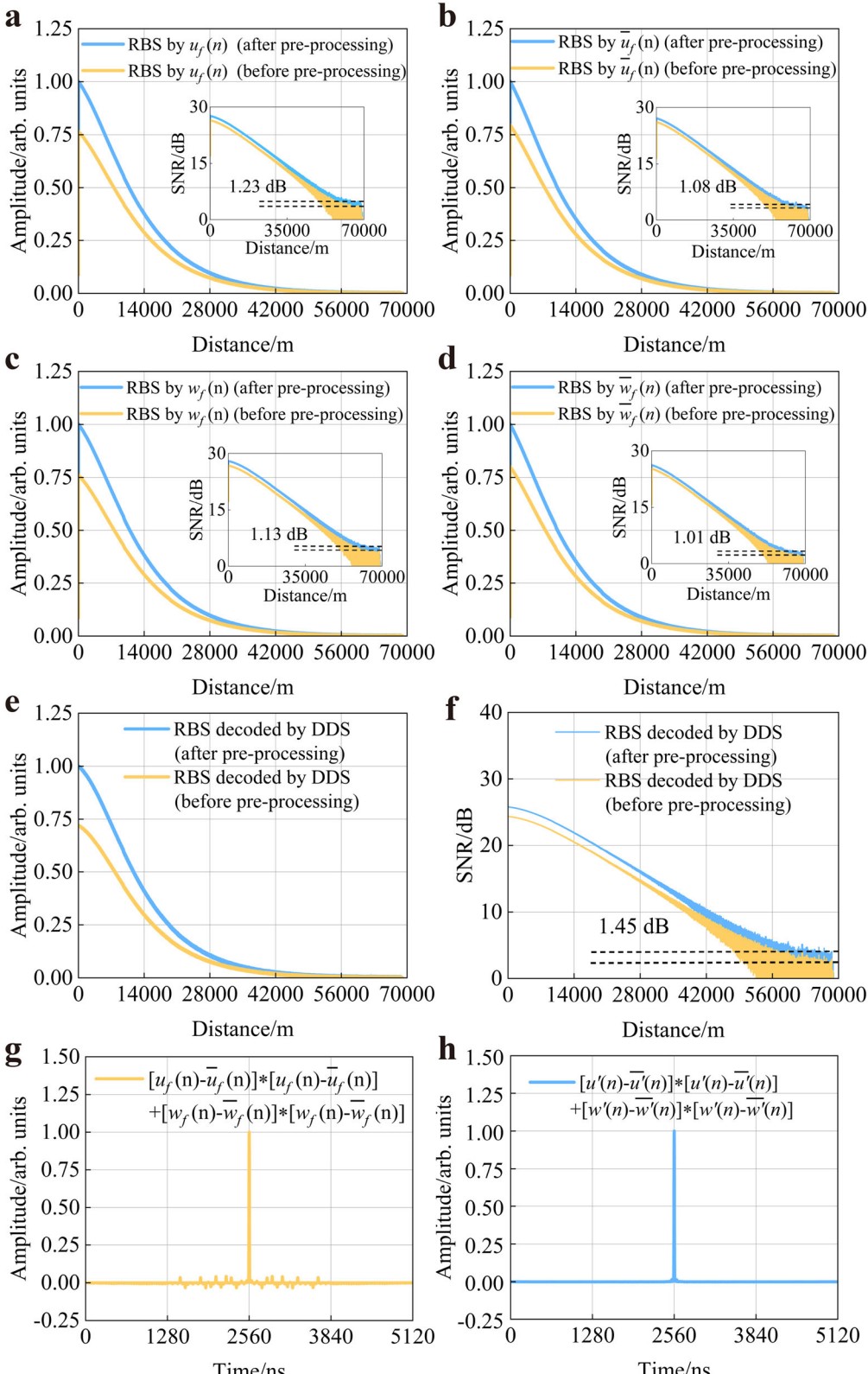

**Fig. 4 | Characteristics analysis of waveform reconstructed RBSs.** Intensity of RBSs (before and after pre-processing) generated by (**a**) detection signal $u(n)$, **b** detection signal $\bar{u}(n)$, **c** detection signal $w(n)$ and **d** detection signal $\bar{w}(n)$. The decoded RBS's (**e**) intensity and **f** SNR. Autocorrelation function curve of the detection signals (**g**) before pre-processing and **h** after pre-processing. DDS: distorted detection signals.

In Eq. (1), $u'(n)$ is the equivalent detection signal after reconstruction of $u(n)$. $f'_1(n)$ is the transient-modulated function of $u(n)$. The $u_f(n)$ represents the detection signal $u(n)$ after it has been subjected to transient effects. Where fft is the Fourier transform operation, ifft is the inverse Fourier transform operation. Then, we performed the same processing on the four detection signals and conducted autocorrelation operations on the processed detection signals. When Eq. (2) holds true, the results demonstrate that the autocorrelation characteristics

are restored after waveform reconstruction, accompanied by the suppression of transient effects. In Eq. (2), signals $u'(n)$, $\bar{u}'(n)$, $w'(n)$, $\bar{w}'(n)$ are the equivalent detection signals obtained from the waveform reconstruction of signals $u(n)$, $\bar{u}(n)$, $w(n)$, $\bar{w}(n)$. The * denotes the corresponding operation. $l$ is code length. $\delta(n)$ is a unit impulse function (Dirac delta function).

$$[u'(n) - \bar{u}'(n)]^*[u'(n) - \bar{u}'(n)]$$
$$+ [w'(n) - \bar{w}'(n)]^*[w'(n) - \bar{w}'(n)] = 2l\delta(n) \qquad (2)$$

The experimental results are shown in Fig. 4g, h. Figure 4g presents the autocorrelation function curve of the detection signals with waveform distortion caused by transient effects. And Fig. 4h shows the autocorrelation function of the equivalent detection signals coupled into the sensing fibre after pre-processing. It can be clearly observed that the sidelobe noise of the autocorrelation function curve is significantly reduced, which verifies that the autocorrelation characteristics are restored and the transient effects are suppressed.

### RBS denoising and SNR enhancement

In this study, a synergistic approach that integrates EAD-coding with Haar wavelet denoising is applied to suppress temperature fluctuation noise along the sensing optical fibre. Based on the EAD-coding scheme, the incident optical power can be substantially increased without inducing stimulated Raman scattering, leading to enhanced effective Raman signal intensity. Furthermore, the separability between encoded signals and uncoded noise enables effective noise suppression, resulting in an improved SNR. Among them, a denoising method based on the Haar wavelet is adopted to process distributed temperature sensing signals with abrupt change characteristics. Owing to its inherently step-like structure, the Haar wavelet is well-suited for extracting features from signal singularities, particularly in regions of abrupt temperature change along the sensing fibre. The Haar wavelet function generates a set of orthogonal basis functions through translation and scaling, enabling multi-scale decomposition of signals. Its inherent advantage in processing low-complexity signals with abrupt change characteristics makes it particularly suitable for the analysis of RBSs.

To improve the SNR performance, we adopted a 6-level Haar wavelet. In addition, in the post-processing stage of temperature demodulation, to further suppress noise and enhance signal smoothness, we additionally applied the db10 (Daubechies wavelet 10) wavelet function with 4-level decomposition. It should be noted that while this step improves signal quality (by reducing temperature fluctuation), it moderately degrades the spatial resolution of the system. Such parameter adjustment reflects the necessary trade-off between spatial resolution and temperature measurement accuracy.

To validate the performance improvement achieved by the synergistic approach that integrates EAD-coding with Haar wavelet denoising, we further compare the decoded RBS intensity based on the EAD-coding scheme with the conventional single-pulse scheme, as shown in Fig. 5a. The EAD-coding scheme exhibits a higher Raman gain, which results in a stronger signal intensity compared to the conventional single-pulse demodulation scheme. As shown in Fig. 5b, c the Haar wavelet is implemented for further denoising to enhance SNR of Raman distributed optical fibre sensing. It could be clearly seen that the EAD-coding scheme (with Haar wavelet) achieves a reduction in peak-to-peak ratio from 0.0081 to 0.0006 at the end of the sensing fibre, which will substantially extend the sensing distance of the system. Furthermore, Fig. 5b corroborates that the Haar wavelet provides consistently effective noise suppression in the single-pulse demodulation scheme. The conventional single-pulse scheme (with Haar wavelet) achieves a reduction in peak-to-peak ratio from 0.013 to 0.0011 at the end of the sensing fibre.

As shown in Fig. 5d, e, the Haar wavelet function significantly improves the SNR for both the EAD-coding scheme and the conventional single-pulse scheme. Experimental results indicate that after Haar wavelet denoising, the SNR of the EAD-coding scheme and the conventional single-pulse demodulation scheme is increased by 7.63 dB and 2.92 dB, respectively. It is worth noting that a relatively significant difference emerges between the two demodulation schemes at the end of the sensing fibre. For the single-pulse scheme, the effective Raman scattering signal is basically submerged by system noise beyond 42.0 km. After incorporating the Haar wavelet function, although the noise level is reduced, the RBS is still submerged by noise beyond 56.0 km. Therefore, the improvement in SNR at the end of the sensing fibre is relatively limited for the single-pulse scheme. This also confirms that the wavelet denoising algorithm has a limited capability to improve SNR in long-distance distributed optical fibre sensing demodulation schemes. Hence, we combine the EAD-coding scheme with Haar wavelet denoising. Only by avoiding the effective Raman scattering signal being submerged by system noise as much as possible can the wavelet denoising algorithm maximise the SNR of the sensing signal.

### Enhancement of temperature sensing performance

The temperature demodulation and SNR results are shown in Fig. 6a shows that the conventional single-pulse scheme exhibits substantial temperature fluctuations on the order of 300 °C. The synergistic approach that integrates EAD-coding with Haar wavelet denoising significantly reduces the system's temperature fluctuation to 22.51 °C while improving the SNR by 7.24 dB (compared to conventional single-pulse scheme), as shown in Fig. 6b, e. Furthermore, the pre-processing architecture within EAD-coding further reduces the system's temperature fluctuation to 20.86 °C by suppressing transient effects and reconstructing RBSs, as shown in Fig. 6c. Meanwhile, the SNR is further improved by an additional 1.48 dB, as shown in Fig. 6f. Temperature fluctuation is expressed as the difference between the measured temperature and the room temperature. Specifically, in Raman distributed optical fibre sensing system, there is a discrepancy between the demodulated temperature ($T'$) at the sampling points of the sensing fibre and the room reference temperature ($T_0$). This discrepancy is defined as temperature fluctuation ($\Delta T = T' - T_0$).

Temperature resolution refers to temperature uncertainty, which characterises the system's ability to distinguish temperature changes, and is generally defined as the standard deviation ($\sigma$) of measured temperatures under constant temperature conditions[34,35]. In this experiment, temperature resolution is characterised by the temperature standard deviation of each distributed temperature measurement point within a fixed length of the sensing fibre. Compared with the temperature resolution of 29.25 °C achieved by the conventional single-pulse scheme, synergistic approach that integrates EAD-coding with Haar wavelet denoising improves the system's temperature resolution to 5.82 °C, as shown in Fig. 6h. Meanwhile, the pre-processing architecture within EAD-coding further enhances the system's temperature resolution to 5.39 °C by suppressing transient effects and reconstructing RBSs, as shown in Fig. 6i. Experimental results show that at a sensing distance of 70.0 km, the EAD-coding and waveform reconstruction scheme achieves a temperature resolution that is 5.43 times better than the conventional method.

Meanwhile, we further compared the temperature measurement accuracy performance of the system, as shown in Fig. 6j–l. The transient effects adversely impact key performance, including the SNR, temperature fluctuation, temperature resolution, and also compromise the overall temperature measurement accuracy of the system. Under the experimental condition of a 90 °C FUT (fibre under test) temperature. The experimental results show that the demodulated temperature of the distorted signal before pre-processing is 104.98 °C, with a maximum measurement deviation of 14.98 °C. As the

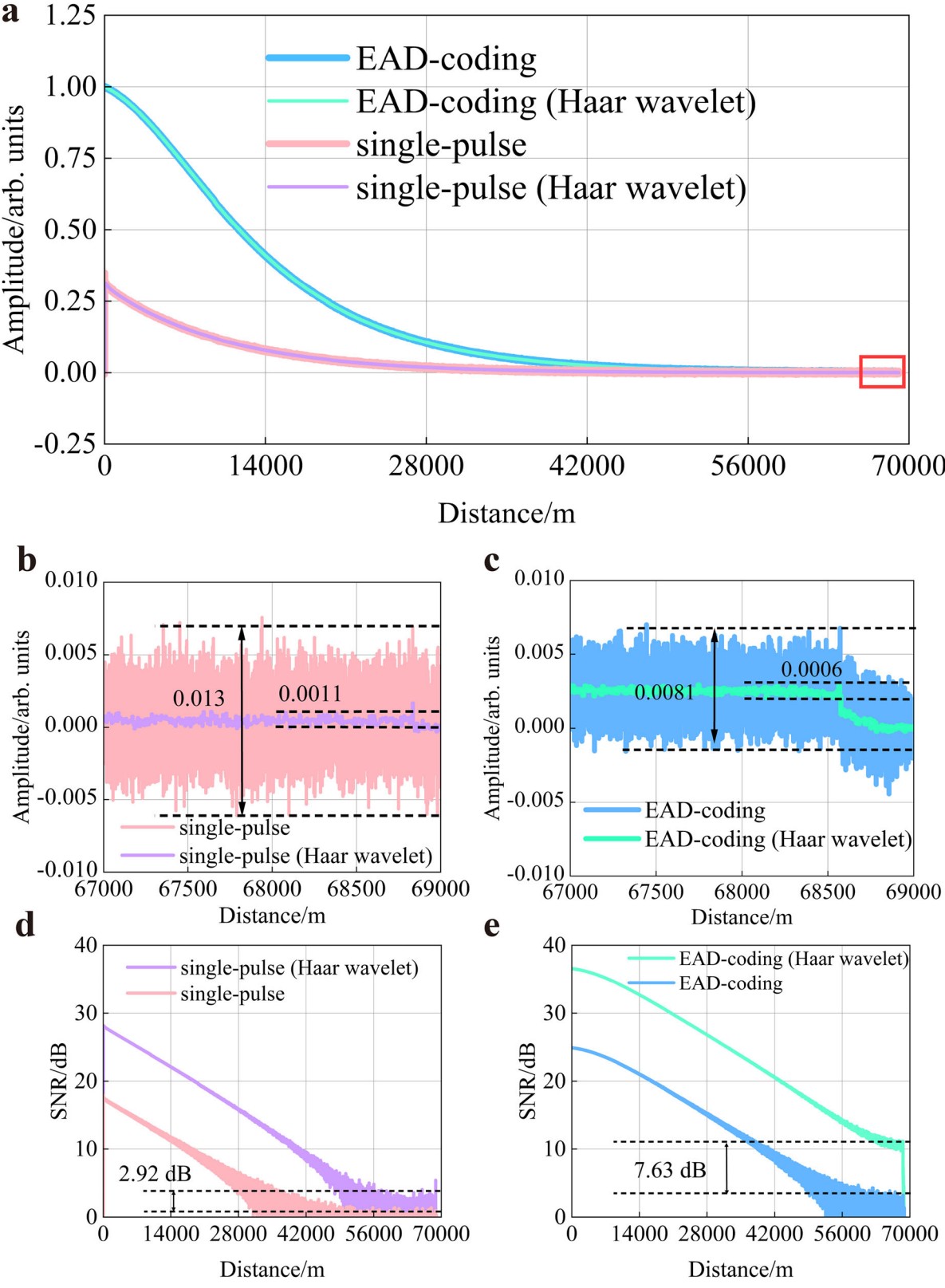

**Fig. 5 | RBSs intensity and SNR based on EAD-coding scheme and conventional single-pulse scheme. a** RBSs intensity in the absence and presence of Haar wavelet denoising. RBSs intensity at the end of sensing fibre based on (**b**) conventional single-pulse scheme and **c** EAD-coding scheme. RBSs SNR at the end of sensing fibre based on (**d**) conventional single-pulse scheme and **e** EAD-coding scheme.

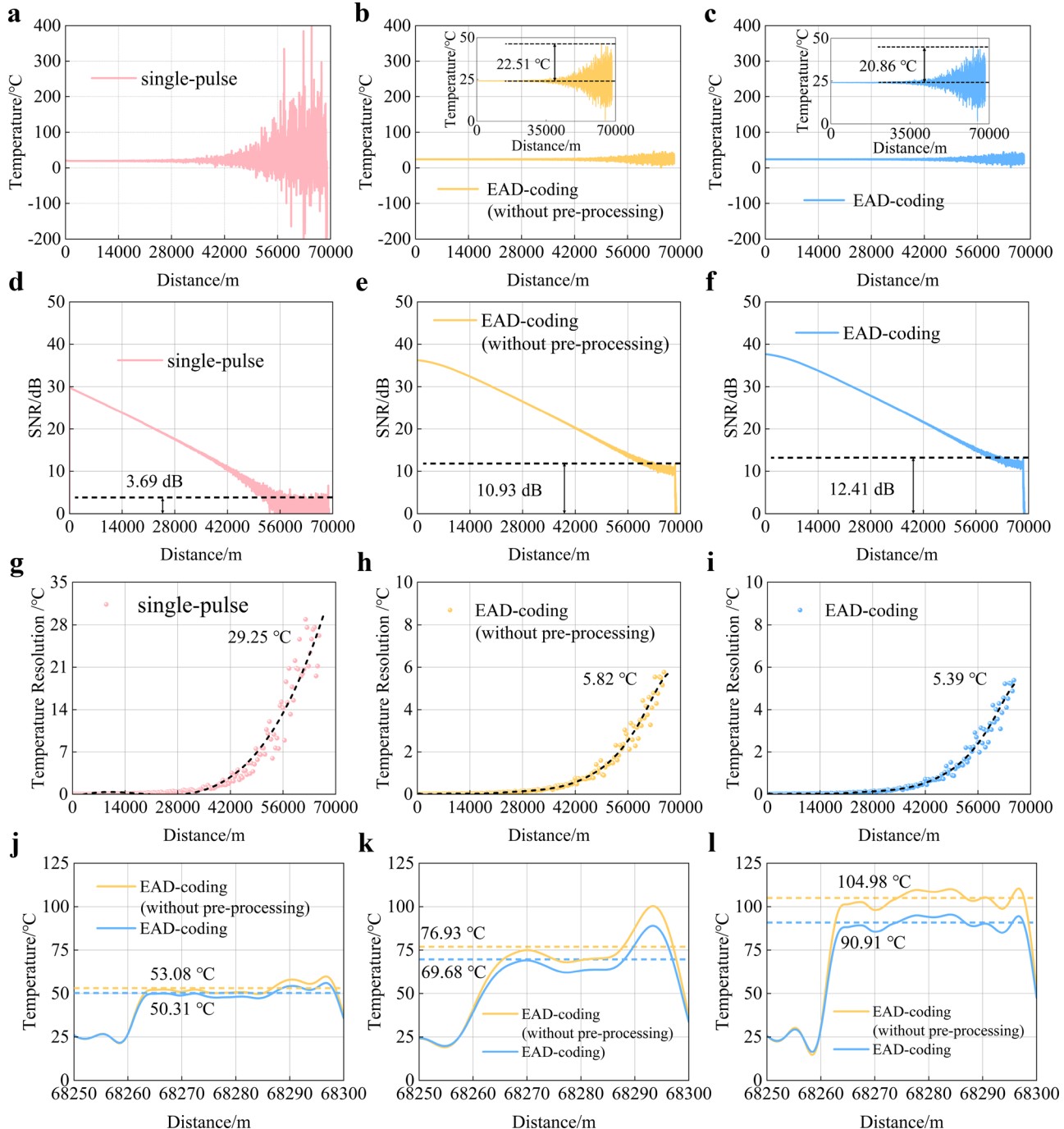

**Fig. 6 | Temperature demodulation comparison results.** Temperature demodulation results at room temperature based on (**a**) conventional single-pulse scheme, **b** EAD-coding scheme (without pre-processing) and **c** EAD-coding scheme (with pre-processing). SNR performance based on (**d**) conventional single-pulse scheme, **e** EAD-coding scheme (without pre-processing) and **f** EAD-coding scheme (with pre-processing). Temperature resolution performance based on (**g**) conventional single-pulse scheme, **h** EAD-coding scheme (without pre-processing) and **i** EAD-coding scheme (with pre-processing). Temperature demodulation results for the FUT at temperature setpoints of (**j**) 50 °C, **k** 70 °C and **l** 90 °C.

temperature rises, the elevation of the Raman backscattered curve in the FUT becomes more pronounced, and temperature measurement deviation gradually increases as well (from 3.08 °C in Fig. 6i to 14.98 °C in Fig. 6l). Consequently, the temperature measurement accuracy deteriorates accordingly. After pre-processing architecture, the temperature demodulation result is 90.91 °C, and the temperature measurement accuracy is 0.91 °C, representing an improvement of approximately 16.46 times. Therefore, the EAD-coding and waveform reconstruction scheme can effectively improve the temperature

measurement accuracy of the long-distance Raman distributed optical fibre sensing system (with a sensing distance of 70.0 km) to within 1.0 °C.

The spatial resolution measurement result is presented in Fig. 7, which demonstrates the precise identification of both the spatial position and length of FUT. This experiment achieves a spatial resolution of 1.58 m. It confirms the effectiveness of the demodulation principle of EAD-coding and the waveform reconstruction scheme. In this scheme, the pulse width of a single code element is set to 10 ns,

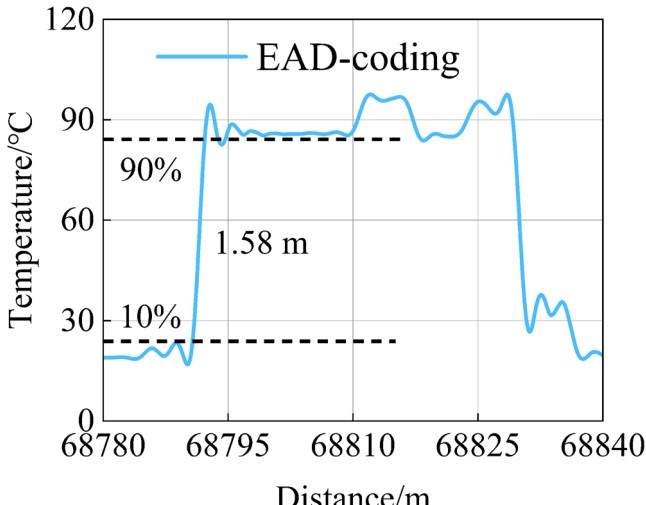

**Fig. 7 | Spatial resolution measurement result.** The spatial resolution is defined as the spatial length corresponding to 10–90% of the intensity of the FUT rising edge, in Raman distributed optical fibre sensing.

corresponding to a theoretical spatial resolution of 1.0 m. The application of wavelet denoising filters out the temperature envelope information in the FUT, resulting in an experimental spatial resolution that is lower than the theoretical value.

Above all, through the EAD-coding, Haar wavelet denoising and waveform reconstruction scheme, the theoretical trade-off bottleneck between SNR and sensing distance in traditional schemes is broken while maintaining relatively low complexity. Experimental results demonstrate that at a long distance of 70.0 km, the system achieves a temperature resolution of 5.39 °C. Moreover, the pre-processing architecture in EAD-coding (integrating autocorrelation characteristic analysis of RBSs and waveform analysis and reconstruction) enhances the temporal fidelity and transmission stability of encoded pulse sequences. This further improves the system's SNR and temperature measurement accuracy, enabling the temperature measurement accuracy to reach 0.91 °C. In this study, the improvements in SNR (accompanied by performance enhancements in sensing distance and temperature resolution) are primarily attributed to the proposed EAD-coding (for coding gain compensation) and Haar wavelet denoising schemes. In contrast, the pre-processing architecture within EAD-coding serves to improve the system's temperature measurement accuracy by suppressing transient effects and reconstructing RBSs.

## Discussion

### Physical performance limitations and solutions

In this proposed scheme, the physical limitations on the optimisation of sensing distance and SNR mainly stem from three aspects: the insufficient performance of experimental devices, the physical demodulation principle and the acquisition time cost.

In terms of experimental devices, the maximum code length of the DDPG and the sampling rate of the DAC are key factors restricting the improvement of the sensing distance and SNR of this scheme. The CIQTEK ASG8100 model DDPG used in the experiment has a maximum code length of 99 bits. Since the code length of Golay complementary sequences is typically $2^n$, a code length of 64 bits was selected in the experiment. The code length directly determines the coding gain of the system, which in turn affects the extent of improvement in sensing performance. Therefore, it is planned to adopt a DDPG with a higher maximum code length (e.g. ≥128 bits) in the future to increase the system's code length, thereby achieving better sensing distance performance. In addition, the currently used DAC has a sampling rate of

1.0 GSa/s and a maximum number of sampling points of 700,000, corresponding to a theoretical maximum sensing distance of approximately 70.0 km. Future plans involve upgrading to a DAC with a higher sampling rate (e.g. ≥2 GSa/s) and a larger number of sampling points (e.g. ≥1,000,000) to eliminate the limitations imposed by hardware devices on the sensing distance.

In terms of the physical demodulation principle, the stimulated scattering threshold and fibre dispersion effect also affect the improvement of sensing distance and SNR in this scheme. Firstly, the stimulated scattering threshold of single-mode fibres is relatively low, which limits the maximum optical power injected into the fibre. As a result, the intensity of the RBS excited in the fibre is relatively weak, leading to a low SNR. In the future, we will attempt to use few-mode fibres to increase the optical power injected into the fibre while minimising intermodal dispersion as much as possible. Finally, fibre dispersion introduced by long-distance transmission causes pulse broadening and intersymbol interference, which destroys the orthogonality of sequences. This results in the broadening of demodulated correlation peaks and a reduction in their amplitude, impairing both SNR and distance resolution. This issue can be compensated for by introducing dispersion-compensating fibres.

In terms of acquisition time cost, the current measurement time is 252 min, limited by a pulse period of 700 μs and a number of averages of three million, with a theoretical measurement time of 35 min × 4 = 140 min. Due to the massive data volume exceeding the on-board real-time processing limit of the DAC, the measurement time is prolonged. In the future, a high-performance DAC will be upgraded and the data processing pipeline optimised, thereby eliminating the delay degradation caused by hardware bottlenecks.

### Comparison of theoretical and practical SNR

For the coding technology with a code length of 64 bits (i.e. the EAD-coding scheme adopted in this paper). The maximum input average optical power for the experimental system in this study was set at 0.2 mW to avoid stimulated Raman scattering. Under this maximum input power achievable by a single-pulse demodulation system, the theoretical SNR improvement is $10\log_{10}[(\sqrt{64}/2)] = 6$ dB.)

However, the theoretical 6 dB SNR improvement is based on the strict premise that the input powers of the single-pulse demodulation scheme and the pulse coding scheme are the same. In this experiment, to explore the extreme performance of the EAD-coding scheme and maximise the system SNR, we set the input power to be close to the maximum value of the stimulated Raman scattering nonlinear effect threshold of the sensing fibre. Specifically, the average power input into the optical fibre is approximately 0.7 mW, and the calculated peak power is close to 1.4 W.

In contrast, the conventional single-pulse demodulation scheme, due to its higher peak power (under the same average power), is more likely to trigger the stimulated Raman scattering nonlinear effect, leading to signal distortion and performance degradation. Therefore, to ensure the stable operation of the single-pulse scheme below the nonlinear threshold, its average input power must be significantly reduced, which is set to approximately 0.2 mW in the experiment.

The different incident powers of the two schemes are incorporated into the calculation of the theoretical SNR. With all other factors remaining completely unchanged, the intensity of the RBS depends solely on the incident optical power. Thus, the modified theoretical SNR improvement is $10\log_{10}[(\sqrt{64}/2) \times (0.7/0.2)] = 11.46$ dB. Since the EDFA introduces a significant amount of amplified spontaneous emission noise while amplifying the effective signal, the final experimental improvement in SNR is 8.72 dB, which is lower than the theoretical value of 11.46 dB. In addition, based on the experimental results and theoretical analysis, the lower SNR improvement at the far end of the sensing fibre compared to the near end is primarily attributed to the cumulative effect of fibre attenuation, which significantly reduces

signal amplitude over distance and diminishes the relative advantage of the EAD-coding scheme near the system's detection limit.

In summary, it is precisely because the EAD-coding scheme allows a higher average input power without triggering significant non-linear effects, and combined with its inherent theoretical coding gain, the SNR improvement observed in the experiment finally reaches 8.72 dB, which is slightly lower than the theoretical SNR coefficient (11.46 dB) at this power setting (0.7 mW).

Additionally, the Supplementary Information includes four sections, consisting of: performance metrics and inherent limitations of other coding techniques, a detailed comparison with other coding techniques, the specific principles of the EAD-coding and waveform reconstruction scheme, and the specific experimental configuration for this approach.

## Methods

In Raman distributed optical fibre sensing system, higher incident optical power corresponds to an improved SNR. Consequently, amplification of the incident light is typically implemented using EDFA. However, the transient effects of EDFA can significantly degrade sensing performance. This phenomenon induces non-uniform amplification, leading to waveform distortion of the encoded pulse after passing through the EDFA. It manifests as a gradual attenuation profile in timing, which disrupts the autocorrelation properties of the encoded pulse. Consequently, the system experiences both a degraded SNR and a constrained maximum sensing distance. The functional representation of this distortion mechanism is formally defined in Eq. (3).

$$u_f(n) = u(n) \cdot f_1(n) \tag{3}$$

Where $u_f(n)$ is the encoded pulse $u(n)$ amplified by the EDFA, and $f_1(n)$ is the attenuation envelope induced by transient effects. The transient effects inherent to EDFA critically distort the waveform of encoded signals, compromising their integrity in correlation-based sensing systems. The intensity of the RBS generated by the signal $u_f(n)$ is governed by the relationship expressed in Eq. (4).

$$I_u(n) = [u(n) \cdot f_1(n)] \otimes h_k(n) \tag{4}$$

Where $I_u(n)$ is the RBS intensity produced by $u_f(n)$, $h_k(n)$ is the pulse response in Raman distributed fibre optical sensing system. The $\otimes$ denotes the convolution operation.

As demonstrated by Eq. (4), all four segments of the acquired RBSs are subjected to transient effects, resulting in attenuation of the backscattered intensity. To mitigate transient effects, a pre-processing scheme is implemented for RBSs. This involves analyzing the distorted signals and reconstructing their waveforms, thereby restoring the autocorrelation properties compromised by transient effects. The attenuation envelope induced by EDFA transient effects is reformulated from a multiplicative operation to a convolutional operation; the RBS is mathematically formulated as shown in the following equation:

$$I_u(n) = u(n) \otimes f_1'(n) \otimes h_k(n) \tag{5}$$

Where $f_1'(n)$ is the transient-modulated function. This function does not require an explicit analytical definition, as its functional form is fully determined by the attenuation envelope $f_1(n)$, as formalised in the following equation:

$$f_1'(n) = \text{ifft}\left\{ \frac{\text{fft}[u(n) \cdot f_1(n)]}{U(k)} \right\} \tag{6}$$

Where fft is the Fourier transform operation, ifft is the inverse Fourier transform operation, $U(k)$ is the Fourier transform result of $u(n)$, $k$ denotes the discrete frequency. It should be noted that $u(n) \cdot f_1(n)$

denotes the timing of the normalised detected signal collected at the output port of the EDFA, and $u(n)$ denotes the timing of the normalised detected signal collected at the output port of the SOA. The RBS in Eq. (7) is subjected to Fourier transform processing, as mathematically expressed in the following equation:

$$\text{fft}[I_u(n)] = U(k) \cdot F_1'(k) \cdot H_k(k) \tag{7}$$

Where the Fourier transform results of the encoded pulse $u(n)$, the transient-modulated function $f_1(n)$ and pulse response $h_k(n)$ are represented by $U(k)$, $F_1'(k)$ and $H_k(k)$. The waveform reconstructed RBS through inverse Fourier transform processing is mathematically formulated in Eq. (8). In Eq. (8), $I_u'(n)$ is the waveform reconstructed RBS intensity.

$$I_u'(n) = \text{ifft}\left( \frac{\text{fft}[I_u(n)]}{F_1'(k)} \right) = u(n) \otimes h_k(n) \tag{8}$$

For the encoded pulse $u(n)$, the RBS intensity (unaffected by transient effects) generated is mathematically expressed through Eq. (9). In Eq. (9), $I(n)$ is the RBS intensity produced by $u(n)$.

$$I(n) = u(n) \otimes h_k(n) \tag{9}$$

Comparison of Eq. (8) with Eq. (9) reveals that the transient effects in the RBS are fully eliminated after waveform reconstruction. The autocorrelation properties can be recovered by implementing the same pre-processing protocol on the remaining three RBSs.

The pre-processing architecture within the EAD-coding and waveform reconstruction scheme incorporates autocorrelation characteristic analysis, waveform analysis and reconstruction of RBSs. It can suppress the transient effects introduced by EDFA, thereby significantly improving the amplitude fidelity and temporal stability of the encoded pulse sequence. This innovative active reconstruction mechanism in the signal domain, which is based on analyzing the waveforms of RBSs via forward/inverse Fourier transform, directly eliminates the phenomenon of detection pulse distortion existing in traditional demodulation schemes. The scope of action of the pre-processing architecture encompasses two key aspects: specifically, one is addressing the fundamental limitation of detection signal distortion through physical-layer waveform reconstruction, and the other is reconstructing RBSs to restore the ideal autocorrelation characteristics of encoded sequences, and achieve a theoretical gain value, thereby achieving the coordinated optimisation of waveform correction, autocorrelation characteristic restoration and signal intensity enhancement.

Furthermore, the proposed EAD-coding and waveform reconstruction scheme features a synergistically enhanced two-stage noise suppression mechanism, demonstrating innovations in multi-level joint denoising.

First, this scheme enhances the SNR of sensing signals through its inherent coding gain. Based on the autocorrelation characteristics of its sequence design, this scheme provides an inherent gain, which effectively improves the system's baseline SNR. Furthermore, this mechanism not only significantly boosts optical power but also leverages the irrelevance between the encoded signals and uncoded noise to achieve preliminary suppression of SNR at the encoding-decoding level.

Second, aiming at the residual noise with specific time-frequency domain distribution characteristics in the decoded signals, this scheme further introduces the Haar wavelet transform for adaptive noise filtering. The multi-resolution analysis capability of the Haar wavelet enables accurate capture of the local time-frequency features of noise. By performing sparse representation of the signal at different scales (corresponding to different frequency bands) and time-domain

positions, it achieves effective separation between useful signals and non-stationary noise components. While preserving the key details of the distributed temperature signal, this method efficiently eliminates residual noise, significantly enhancing the clarity of the signal in the time-frequency domain and the overall measurement accuracy of the system.

This three-fold synergistic mechanism, consisting of the pre-processing framework compensating for suppressing transient effects, coding gain improves baseline SNR, and wavelet transform finely removing residual noise, breaks through the theoretical trade-off bottleneck between SNR and sensing distance in traditional schemes. This scheme provides a framework for long-distance Raman distributed optical fibre sensing. Experimental results indicate that, at a long distance of 70.0 km, this scheme can achieve a spatial resolution of 1.58 m, a temperature resolution of 5.39 °C, a measurement accuracy of 0.91 °C, and an effective number of sensing points reaching 44,303. To the best of our knowledge, the effective number of sensing points of the EAD-coding and waveform reconstruction scheme currently represents the world's highest level in the field of Raman distributed optical fibre sensing.

## Data availability

The Figs. 3–7 source data generated in this study are deposited in Zenodo (https://doi.org/10.5281/zenodo.17617324).

## Code availability

The code of EAD-coding is deposited in Zenodo (https://doi.org/10.5281/zenodo.17617408).

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

## Acknowledgements

This study is supported by the National Natural Science Foundation of China (62475183, 62205234, U23A20375); National Key Research and

Development Program of China (2023YFF0715700); Shanxi Provincial Key Research and Development Project (202302150101002, 202202030201004).

## Author contributions

M.J.Z. and J.L. led this work and supervised the project. J.L. and F.Z. conceived the idea and designed the experiment. F.Z., L.L.L. and K.Y.C. contributed to experiment implementation and data processing. All authors contributed to the writing of the paper.

## Competing interests

The authors declare no competing interests.
