## [Transparent Peer Review file · Nature Communications]

70 km Long-Range Raman Distributed Optical Fibre Sensing Through Enhanced Anti-distortion Coding and Waveform Reconstruction

Corresponding Author: Mr Jian Li

Version 0:

Reviewer comments:

Reviewer #1

(Remarks to the Author)

The authors present a novel manuscript discussing a long-range Raman distributed optical fiber sensing scheme through Golay-encoded autocorrelation and waveform reconstruction technology. The practical implementation of Raman-based distributed optical fiber sensing has been fundamentally constrained by the inherent low signal-to-noise ratio, particularly for operational ranges exceeding 30 km. The authors propose a groundbreaking paradigm integrating Golay-encoded autocorrelation processing with advanced Raman scattering waveform reconstruction to transcend this physical limitation. And the newly developed preprocessing framework simultaneously optimizes complementary sequence correlation and effectively mitigates disturbances of transient effects. The experimental results demonstrate a sensing performance of 70 km operational range with 1.58 m spatial resolution, while maintaining 0.88 °C measurement accuracy. To my knowledge, this represents the first work in achieving such a long sensing distance with 1 meter-level spatial resolution. The manuscript can be accepted if the following comments are addressed:

- (1) In Raman distributed temperature sensing systems, spatial resolution is intrinsically determined by the probing pulse duration, as the pulse width directly defines the minimum spatial discrimination through time-of-flight measurements. The paper explicitly states that the pulse width of a single code element is set to 10 ns and single-mode fiber is employed to avoid spatial resolution degradation due to intermodal dispersion. However, the experimental results show that the achieved spatial resolution failed to reach the theoretical spatial resolution of 1 m. What is the reason for the different between theoretical prediction and experimental outcome?
- (2) The method mentioned in this paper only explores the effect of the transient effect on anti-Stokes light in analyzing the transient effect. And only the anti-Stokes light is used in the temperature demodulation process, discarding the use of Stokes signals. How does this differ from the traditional dual-channel demodulation scheme that relies on both anti-Stokes and Stokes signals, and what are its significant advantages?
- (3) The experimental setup section lacks detailed descriptions of the specific types and parameters for each device, which may hinder readers from reproducing the experiment. It is advisable to supplement the detailed specifications of the optical components used, including their models, manufacturers, and key optical parameters.
- (4) In this paper, the Raman backscattered signals are analyzed by calculating the SNR in Fig. 4, whereas only peak-to-peak values are compared in the wavelet denoising results presented in Fig. 5. It is recommended to include a SNR analysis to ensure consistency in evaluation metrics across different experimental analyses.
- (5) This paper only compares the EAC-coding scheme with the traditional single pulse scheme in the experiment, which hinders the demonstration of its superiority. It is recommended to include a table in the discussion section detailing the performance metrics and comparative advantages/disadvantages of various mainstream techniques, thereby highlighting the superiority of this study.

(6) The paper clearly states that the acquisition time for each Raman scattering signal is 63 minutes, and a total of four Raman scattering signals need to be acquired during the decoding stage. Given the prolonged acquisition duration, how is the stability of the experimental temperature and optical instruments performance ensured?

(7) Discrepancies in typography consistency are observed in Fig. 3-Fig. 7, particularly regarding the font sizes of axis captions and intra-figure text. For example, the axis labels in Fig. 6-Fig. 7 and the text in Fig. 7(b) exhibit notably larger font sizes compared to other figures. It is recommended to standardize the typography across all figures to ensure visual coherence and readability.

(8) Some of the chart labels are not clear enough. For example, the text in Fig. 2 is undersized and the color contrast is insufficient, while the caption text in Fig. 3(b) appears blurred. It is recommended that the format and size of all figures be standardized to improve visual clarity and ensure that labels and annotations are clear and easy to read and consistent throughout the manuscript.

Reviewer #2

(Remarks to the Author)

The authors have demonstrated a record-long-distance distributed temperature sensing system based on Raman scattering. By combining Golay-encode autocorrelation processing, EDFA-transient-effect mitigation, and Haar wavelet denoising, the performance of the sensing system has been significantly enhanced, achieving 1.58m spatial resolution and 0.88 °C measurement accuracy at a maximum sensing range of 70km. This work represents a solid engineering breakthrough in long-distance distributed temperature sensing and has the potential to generate societal impacts by providing new capabilities to infrastructure monitoring. Therefore, this manuscript can be accepted after being further improved by addressing the specific comments below.

1. The introduction should be significantly optimized to clearly state the key innovation of this work. For example, the authors mentioned several existing denoising techniques, but their limitations and the corresponding advantages of Haar wavelet denoising is not mentioned at all.
2. As the core innovation of this work, the uniqueness of Golay-encode autocorrelation processing is insufficiently explained. How does it compare quantitatively to other well established coding methods such as Simplex coding and genetic-optimized aperiodic coding? For example, maybe the discussion can be made from the perspective of coding gain and measurement time. After all, the physical reason that the record-breaking performance can be achieved should be explicitly pointed out.
3. To let the reader to better evaluate the performance in this work, a quantitative and fair comparison with the state of the art should be made. If possible, a figure of merit should be given that normalizes the measurement time, considering that the performance in this work is obtained over one hour.
4. More in-depth discussions should be provided to talk about the physical limit of current performance and potential ways for further enhancement.
5. Details of Haar wavelet denoising should be provided. How is it implemented? Does it sacrifice spatial resolution?
6. It is confusing why the temperature-measurement uncertainty can be much larger than the resolution.
7. Some minor mistakes should be revised. For example, peak-to-peak ratio from 0.0067 to 0.0006 should be attributed to Haar wavelet denoising instead of to EAC-coding. Results for the single-pulse scheme are shown in Fig. 5(b1) instead of Fig. 5(b2).

Reviewer #3

(Remarks to the Author)

Please see attached file.

Version 1:

Reviewer comments:

Reviewer #1

(Remarks to the Author)

My comments have been weel addressed, and all the questions have been answered in detail. The current manuscript can be accepted for publication.

Reviewer #2

(Remarks to the Author)

I can see that the authors have been trying very hard to carefully address all the comments that the reviewers raised. While all the comments have been comprehensively discussed, the current manuscript is way too lengthy, especially for the introduction and the discussion parts. Much of the content should be much more concise or moved to the supplementary material.

Reviewer #3

(Remarks to the Author)

The response seems reasonable. Only remaining issue is the overhead in terms of the time taken for the processing. Perhaps the authors can include that in the manuscript.

Response to the reviewers' comments

Thank you very much for your consideration of our manuscript entitled “Overcoming the Longstanding Challenge of Long-Range Raman Distributed Optical Fiber Sensing Through Enhanced Anti-distortion Coding and Waveform Reconstruction” by Fan Zhang, Jian Li, Lulei Li, Kangyi Cao and Mingjiang Zhang.

We would like to thank the reviewers for their efforts to review our manuscript and their appreciation of the novelty and quality of our work. The constructive suggestions from the reviewers have been thoroughly considered and implemented in the revised manuscript, which now has been significantly improved.

In the next several pages, the changes made to the manuscript are detailed to fully address the concerns raised according to each of the reviewers' comments. We hope the manuscript is now appropriate for publication in *Nature Communications*.

In addition, we would like to declare:

- (1) All authors agree with the submission.
- (2) The work has not been published or submitted for publication elsewhere, either completely or in part, or in another form or language.
- (3) No materials are reproduced from another source.
- (4) The authors declare no conflicts of interest.

We look forward to hearing from you!

With best regards,

Jian Li, Mingjiang Zhang

All the reviewers' suggestions have been considered. The following are the changes made in the manuscript to fully address the reviewers' concerns:

Blue: Original comments from the reviewer.

Black: Our response including action taken.

Responds to the reviewers' comments:

Reviewer #1 (Comments to the Author):

Recommendations for the Author(s):

The authors present a novel manuscript discussing a long-range Raman distributed optical fiber sensing scheme through Golay-encoded autocorrelation and waveform reconstruction technology. The practical implementation of Raman-based distributed optical fiber sensing has been fundamentally constrained by the inherent low signal-to-noise ratio, particularly for operational ranges exceeding 30 km. The authors propose a ground breaking paradigm integrating Golay-encoded autocorrelation processing with advanced Raman scattering waveform reconstruction to transcend this physical limitation. And the newly developed preprocessing framework simultaneously optimizes complementary sequence correlation and effectively mitigates disturbances of transient effects. The experimental results demonstrate a sensing performance of 70 km operational range with 1.58m spatial resolution, while maintaining 0.88°C measurement accuracy. To my knowledge, this represents the first work in achieving such a long sensing distance with 1meter-level spatial resolution. The manuscript can be accepted if the following comments are addressed:

Reply: Thank you very much for your positive comments on the content of our research and constructive suggestions on the manuscript. Those comments are all valuable and very helpful for revising and improving our manuscript, as well as the important guiding significance to our research. We have studied the comments carefully and have made corrections which we hope meet with approval.

1. In Raman distributed temperature sensing systems, spatial resolution is intrinsically determined by the probing pulse duration, as the pulse width directly defines the minimum spatial discrimination through time-of-flight measurements. The paper explicitly states that the pulse width of a single code element is set to 10 ns and single-mode fiber is employed to avoid spatial resolution degradation due to intermodal dispersion. However, the experimental results show that the achieved

spatial resolution failed to reach the theoretical spatial resolution of 1m. What is the reason for the different between theoretical prediction and experimental outcome?

Reply: Thank you very much for your professional comments, and we fully agree with the viewpoints. The reason for the slight discrepancy between the theoretical spatial resolution and the experimental spatial resolution in this manuscript lies in the fact that the Haar and db10 (Daubechies wavelet 10) wavelet denoising methods used in the experiment will filter out the edge information of the temperature envelope in the sensing fiber region, which leads to the error between the experimental spatial resolution and the theoretical spatial resolution of the system. The specific reasons are as follows.

In this manuscript, the pulse width of a single code element is set to 10 ns, corresponding to a theoretical spatial resolution of 1.0 m. Furthermore, single-mode fiber is employed to avoid the degradation of spatial resolution caused by intermodal dispersion in the sensing fiber. However, the intensity of Raman scattering signals excited in single-mode fiber is relatively low. Therefore, in the experiment, to further improve the SNR of the sensing signals, Haar wavelet denoising processing is applied to the Raman anti-Stokes scattering signals after the decoding process. It is suitable for processing low-complexity signals with mutation characteristics, thus making it well-suited for handling fiber optic Raman scattering signals. Meanwhile, in this manuscript, after temperature demodulation, the db10 wavelet method is synchronously applied to the distributed temperature demodulation curve to further improve the SNR of the sensing signal. However, this denoising method will filter out the edge information of the temperature envelope in the sensing fiber region, resulting in a final experimentally measured spatial resolution of 1.58 m, which differs from the theoretical spatial resolution of 1.0 m.

Thank you for your suggestion. We have made the following modifications in our revised manuscript:

- 1) In the section of **Experimental results, page 17, right column, paragraph 2, line 11**, add “In the proposed scheme, the pulse width of a single code element is set to 10 ns, corresponding to a theoretical spatial resolution of 1.0 m. Furthermore, single-mode fiber is employed to avoid the degradation of spatial resolution caused by intermodal dispersion in the sensing fiber. However, the

intensity of Raman scattering signals excited in single-mode fiber is relatively low. Therefore, in the experiment, to further improve the SNR of the sensing signals, Haar wavelet denoising processing is applied to the Raman anti-Stokes scattering signals after the decoding process. It is suitable for processing low-complexity signals with mutation characteristics, thus making it well-suited for handling fiber optic Raman scattering signals. Meanwhile, in this paper, after temperature demodulation, the db10 wavelet method is synchronously applied to the distributed temperature demodulation curve to further improve the SNR of the sensing signal. However, this denoising method will filter out the edge information of the temperature envelope in the sensing fiber region, resulting in a final experimentally measured spatial resolution of 1.58 m, which differs from the theoretical spatial resolution of 1.0 m.”

2. The method mentioned in this paper only explores the effect of the transient effect on anti-Stokes light in analyzing the transient effect. And only the anti-Stokes light is used in the temperature demodulation process, discarding the use of Stokes signals. How does this differ from the traditional dual-channel demodulation scheme that relies on both anti-Stokes and Stokes signals, and what are its significant advantages?

Reply: We sincerely appreciate the reviewer’s professional comments. In the experimental process of this scheme, only the Raman anti-Stokes scattering signals were analyzed and processed, and the subsequent temperature demodulation was also performed solely based on the Raman anti-Stokes scattering signals. Compared with the dual-path demodulation scheme (where Raman Stokes scattering light is used to demodulate Raman anti-Stokes scattering signals), this scheme has the following two specific advantages.

(1) The Raman anti-Stokes scattering signal based on single-path demodulation exhibits higher sensitivity to the temperature distribution along the sensing fiber. Moreover, due to its simpler signal processing link and the elimination of the need for additional light splitting or switching mechanisms, the single-path demodulation scheme is more easily integrated with the Golay code modulation technology proposed in this manuscript at the hardware implementation level. Therefore, the characteristics of single-path demodulation, such as its low-complexity signal link, high hardware compatibility, and absence of multi-path signal conflicts, are highly

compatible with the requirements of pulse coding for signal integrity and hardware integration.

(2) The single-path demodulation scheme can enhance the system's measurement rate and reduce system costs. In the dual-path demodulation scheme, both the calibration process and the measurement process require independent acquisition of two channels of scattering signals, resulting in a doubling of the original data volume. This not only approximately doubles the overall measurement cycle of the system but also imposes higher requirements on the storage depth of the data acquisition system, thereby increasing the system cost.

Thank you for your suggestion. We have made the following modifications in our revised manuscript:

- 1) In the section of **Experimental setup, page 9, right column, paragraph 2, line 4, add** “In the experimental process of this scheme, only the Raman anti-Stokes scattering signals were analyzed and processed, and the subsequent temperature demodulation was also performed solely based on the Raman anti-Stokes scattering signals. Compared with the dual-path demodulation scheme (where Raman Stokes scattering light is used to demodulate Raman anti-Stokes scattering signals), this scheme has the following two specific advantages. The Raman anti-Stokes scattering signal based on single-path demodulation exhibits higher sensitivity to the temperature distribution along the sensing fiber; The single-path demodulation scheme can enhance the system's measurement rate and reduce system costs.”

3. The experimental setup section lacks detailed descriptions of the specific types and parameters for each device, which may hinder readers from reproducing the experiment. It is advisable to supplement the detailed specifications of the optical components used, including their models, manufacturers, and key optical parameters.

Reply: We sincerely appreciate the reviewer's professional suggestions. In the revised manuscript, we have supplemented the model numbers, manufacturers, and key optical parameters of the equipment used in the experiments. The detailed supplementary information is as follows.

- ① Laser (KEYANG PHOTONICS; DFB Continuous laser; Center Wavelength:1550 nm)

- ② SOA (OPEAK; OAM-SOA-PL-15-15-S; Operating Wavelength: 1520-1570 nm)
- ③ DDPG (CIQTEK; ASG8100; Maximum Coding bit: 99 bit)
- ④ EDFA (OPEAK; EDFA-C-PL-MB-100-S; Operating Wavelength: 1550 nm)
- ⑥ WDM (OPEAK; WDM-1*3-1550; Isolation: ≥ 60 dB)
- ⑦ APD (KEYANG PHOTONICS; KY-DTS-200M; Bandwidth:200 MHz)
- ⑧ DAC (CIQTEK; DAQ2100; Sampling Rate: 1 GSa/s)

Thank you for your suggestion. We have made the following modifications in our revised manuscript:

- 1) In the section of **Experimental setup, page 8, right column, paragraph 2, line 5**, add “The equipment used in the experiment, along with their corresponding model numbers, manufacturers, and key optical parameters, are listed as follows: Laser (KEYANG PHOTONICS; DFB Continuous laser; Center Wavelength:1550 nm);SOA (OPEAK; OAM-SOA-PL-15-15-S; Operating Wavelength: 1520-1570 nm); DDPG (CIQTEK; ASG8100; Maximum Coding bit: 99 bit); EDFA (OPEAK; EDFA-C-PL-MB-100-S; Operating Wavelength: 1550 nm); WDM (OPEAK; WDM-1*3-1550; Isolation: ≥ 60 dB); APD (KEYANG PHOTONICS; KY-DTS-200M; Bandwidth:200 MHz); DAC (CIQTEK; DAQ2100; Sampling Rate: 1 GSa/s). And all experimental equipment used in this experiment are conventional commercial products available on the market.”

4. In this paper, the Raman backscattered signals are analyzed by calculating the SNR in Fig.4, whereas only peak-to-peak values are compared in the wavelet denoising results presented in Fig.5. It is recommended to include a SNR analysis to ensure consistency in evaluation metrics across different experimental analyses.

Reply: Thank you very much for your professional comments, and we fully agree with the viewpoints. In Fig. 1, we have added an experimental analysis and comparison of the changes in SNR of Raman scattering signals before and after wavelet denoising. The experimental results are as follows.

As shown in Fig. 1(c1) and 1(c2), the Haar wavelet function significantly improves the SNR for both the EAC-coding scheme proposed in

this manuscript and the traditional single-pulse demodulation scheme. Experimental results indicate that after Haar wavelet denoising, the SNR of the EAC-coding scheme and the traditional single-pulse demodulation scheme is increased by 7.63 dB and 2.92 dB, respectively. It is worth noting that a relatively significant difference emerges between the two demodulation schemes at the end of the sensing fiber. For single-pulse scheme, the effective Raman scattering signal is basically submerged by system noise beyond 42.0 km. After incorporating the Haar wavelet function, although the noise level is reduced, the signal is still submerged by noise beyond 56.0 km. Therefore, the improvement in SNR at the end of the sensing fiber is relatively limited for the single-pulse scheme. This also confirms that the wavelet denoising algorithm has a limited capability to improve SNR in long-distance distributed optical fiber sensing demodulation schemes. Hence, we combine the EAC-coding scheme with Haar wavelet denoising. Only by avoiding the effective Raman scattering signal being submerged by system noise as much as possible can the wavelet denoising algorithm maximize the SNR of the sensing signal.

Fig. 1 (c1) SNR of Raman backscattered signals based on conventional single-pulse scheme at the end of sensing fiber. (c2) SNR of Raman backscattered signals based on EAC-coding scheme at the end of sensing fiber.

Thank you for your suggestion. We have made the following modifications in our revised manuscript:

- 1) In the section of **Experimental results**, page 12, right column, change “Fig. 5” to “Fig. 5”

Fig. 5. Intensity of Raman backscattered signals based on EAC-coding scheme and conventional single-pulse. (a) Intensity of Raman backscattered signals based on EAC-coding scheme and conventional single-pulse with and without Haar wavelet. (b1) Intensity of Raman backscattered signals based on conventional single-pulse at the end of sensing fiber. (b2) Intensity of Raman backscattered signals based on EAC-coding scheme at the end of sensing fiber. (c1) SNR of Raman backscattered signals based on conventional single-pulse scheme at the end of sensing fiber. (c2) SNR of Raman backscattered signals based on EAC-coding scheme at the end of sensing fiber.

2) In the section of **Experimental results**, page 13, right column, paragraph 2, line 11, add “As shown in Fig. 5(c1) and 5(c2), the Haar wavelet function significantly improves the SNR for both the EAC-coding scheme proposed in this paper and the traditional single-pulse demodulation scheme. Experimental results indicate that after Haar wavelet denoising, the SNR of the EAC-coding scheme and the traditional single-pulse demodulation scheme is increased by 7.63 dB and 2.92 dB, respectively. It is worth noting that a relatively significant difference emerges between the two demodulation schemes at the end of the sensing fiber. For single-pulse scheme, the effective Raman scattering signal is

basically submerged by system noise beyond 42.0 km. After incorporating the Haar wavelet function, although the noise level is reduced, the signal is still submerged by noise beyond 56.0 km. Therefore, the improvement in SNR at the end of the sensing fiber is relatively limited for the single-pulse scheme. This also confirms that the wavelet denoising algorithm has a limited capability to improve SNR in long-distance distributed optical fiber sensing demodulation schemes. Hence, we combine the EAC-coding scheme with Haar wavelet denoising. Only by avoiding the effective Raman scattering signal being submerged by system noise as much as possible can the wavelet denoising algorithm maximize the SNR of the sensing signal.”

5. This paper only compares the EAC-coding scheme with the traditional single pulse scheme in the experiment, which hinders the demonstration of its superiority. It is recommended to include a table in the discussion section detailing the performance metrics and comparative advantages/disadvantages of various mainstream techniques, thereby highlighting the superiority of this study.

Reply: We would like to express our sincere gratitude to the reviewer for professional suggestions. In response to these valuable comments, we have added a new table in the Discussion section, which provides a detailed comparison of the performance metrics, advantages, and disadvantages of various current pulse coding techniques, as shown in Table 1. The modified and newly added contents are as follows.

Pulse coding schemes have demonstrated significant advantages in long-distance Raman distributed optical fiber sensing system. On one hand, by encoding the detection optical pulses, this technology can significantly increase the optical power injected into the sensing fiber while avoiding fiber nonlinear effects, thereby enhancing the intensity of effective sensing signals at the level of signal generation mechanism. On the other hand, at the data processing level, it leverages the separability between coded signals and uncoded noise to effectively suppress noise and improve the SNR of system.

Currently, pulse coding scheme applied in Raman distributed optical fiber sensing system are primarily classified into six categories, which specifically include the Simplex coding scheme, Pre-Shaped Simplex Coding scheme, low-repetition-rate cyclic pulse coding scheme, Genetic-optimised aperiodic coding scheme, Derived Sequences coding scheme, and

EAC-coding scheme.

Simplex coding scheme, as a linear coding technique based on Hadamard matrix transformation, has its code length (in L bits) determining the number of detection signals required (L groups). For instance, in 2006, J. Park, *et al.* achieved a spatial resolution of 17.0 m, a temperature resolution of 3.0 °C, and a total of 2176 effective sensing points over a sensing distance of 37.0 km by utilizing this coding scheme and link optimization techniques [1].

Low-repetition-rate cyclic pulse coding scheme generates Simplex coding-based cyclic coding sequences using an acousto-optic modulator (AOM), enabling pulses to be injected into the optical fiber periodically at a low repetition rate. For example, in 2011, M. A. Soto, *et al.* combined the low-repetition-rate quasi-periodic cyclic pulse coding with a high-power fiber laser, achieving a spatial resolution of 1.0 m, a temperature resolution of 3.0 °C, and 26,000 effective sensing points over a sensing distance of 26.0 km [2].

Pre-Shaped Simplex Coding scheme is based on simplex coding technology and adopts a linearly increasing profile to adjust pulse amplitude, compensating for the amplitude variation caused by EDFA to suppress the transient effect of the EDFA and improve system performance. For example, in 2017, J. B. Rosolem, *et al.* achieved a spatial resolution of 10.0 m, a temperature resolution of 8.4 °C, and 6,200 effective sensing points over a sensing distance of 62.0 km by using the pre-shaped Simplex coding and a gain-controlled erbium-doped fiber amplifier [3].

Genetic-optimised aperiodic coding scheme utilizes a distributed genetic algorithm (DGA) to generate a single-sequence aperiodic code (GO-code), which is then converted into an optical pulse sequence and injected into the optical fiber. For example, in 2020, X. Z. Sun, *et al.* achieved a spatial resolution of 1.0 m, a temperature resolution of 3.9 °C, and 39,000 effective sensing points over a sensing distance of 39.0 km based on the Genetic-optimised aperiodic coding scheme [4].

Derived Sequences coding scheme addresses the transient effect of EDFA through a derived sequence decoding method, thereby enhancing the performance of long-distance sensing. For instance, in 2022, D. D. Chai, *et al.* achieved a spatial resolution of 5.0 m, a temperature resolution of 2.5 °C, and 8,800 effective sensing points over a sensing distance of 44.0 km based on the Derived Sequences coding scheme [5].

EAC-coding scheme proposed in this manuscript incorporates a novel preprocessing framework for autocorrelation characteristic analysis, waveform analysis and reconstruction of Raman backscattered signals. This framework effectively suppresses the transient effect caused by EDFA and significantly enhances the fidelity and stability of the coded pulse sequence. Specifically, the intrinsic gain provided by the pulse coding itself effectively compensates for signal attenuation induced by long-distance transmission, thereby notably improving the base SNR. On this basis, a Haar wavelet denoising algorithm is further integrated. For the residual noise in the signal after encoding and decoding—noise with specific time-frequency domain characteristics (e.g., broadband noise or interference in specific frequency bands), the wavelet denoising algorithm can adaptively filter it out. This three-fold synergistic mechanism, consisting of the preprocessing framework compensating for EDFA fluctuation, coding gain suppressing transmission losses, and wavelet transform finely removing residual noise, breaks through the bottlenecks of traditional schemes in terms of SNR and effective sensing distance. Over a sensing distance of 70.0 km, this technology achieves a spatial resolution of 1.58 m, a temperature resolution of 5.39 °C, and 44,303 effective sensing points. To the best of our knowledge, the number of effective sensing points achieved by this scheme ranks the highest in the field of Raman distributed optical fiber sensing systems.

Table 1. Sensing performance of various schemes.

Scheme	Sensing Distance	Spatial Resolution	Sensing Points	Temperature Resolution	Measurement Time
Simplex coding ^[1]	37.0 km	17.0 m	2176	3.0 °C	/
Low-repetition-rate cyclic pulse coding ^[2]	26.0 km	1.0 m	26000	3.0 °C	30 s
Pre-Shaped Simplex Coding ^[3]	62.0 km	10.0 m	6200	8.4 °C	/
Genetic-optimised aperiodic coding ^[4]	39.0 km	1.0 m	39000	3.9 °C	13.6 min
Derived Sequences coding ^[5]	44.0 km	5.0 m	8800	2.5 °C	/
EAC-coding	70.0 km	1.58 m	44303	5.39 °C	252 min

Sensing Points refers to the number of independent spatial sampling points in the fiber optic sensing link where the system can achieve effective measurements. The specific calculation method is the ratio of the sensing distance to the spatial resolution.
Temperature resolution is defined as the minimum temperature change that a system can reliably distinguish, and its calculation method is based on the standard deviation under steady-state conditions.

[1] J. Park, *et al.* Raman-based distributed temperature sensor with simplex coding and link optimization. *IEEE Photonics Technol. Lett.* **18**(17): 1879-1881 (2006).

[2] M. A. Soto, *et al.* Raman-based distributed temperature sensor with 1 m spatial resolution over 26 km SMF using low-repetition-rate cyclic pulse coding. *Opt. Lett.* **36**(13): 2557-2559 (2011).

- [3] J. B. Rosolem, *et al.* Raman DTS based on OTDR improved by using gain-controlled EDFA and pre-shaped simplex code. *IEEE Sens. J.* **17**(11): 3346-3353 (2017).
- [4] X. Z. Sun, *et al.* Genetic-optimised aperiodic code for distributed optical fibre sensors. *Nat. Commun.* **11**(1), 5774 (2020).
- [5] D. D. Chai, *et al.* Derived sequences decoding approach for long-range distributed temperature sensors. *IEEE Sens. J.* **23**(3): 2204-2210 (2022).

Thank you for your suggestion. We have made the following modifications in our revised manuscript:

- 1) In the **(unmodified manuscript)** section of **Introduction, page 2, left column, paragraph 3, line 35, delete “The pulse coding scheme employs specific functional equations to encode detection signals, and then decodes the collected Raman backscattered signals through a decoding program ^[21-22]. It increases the intensity of the detection signals without the stimulated Raman generating nonlinear effects and thus improve the SNR of system. For example, Sun *et al.* proposed a genetic-optimised aperiodic coding scheme achieving a sensing spatial resolution of 1.0 m at the sensing distance of 39.0 km ^[23]. With the development of digital signal processing, various denoising schemes are used in Raman distributed optical fiber sensing, including wavelet denoising algorithm and dynamic sampling-correction method ^[24-25]. For example, M. Yu, *et al.* proposed a dynamic sampling-correction scheme achieving a sensing distance of 20.0 km ^[26]. H. S. Pradhan, *et al.* proposed a Fourier wavelet regularised deconvolution scheme achieving a sensing distance of 30.0 km with a spatial resolution of 3.0 m ^[27].**

Furthermore, the neural network scheme can learn the characteristic patterns of Raman scattering signals, perform convolutional operations and subsequent processing on the input signals to effectively identify and remove noise components, thereby improving the SNR ^[28]. For example, Zhang *et al.* proposed a deep 1-D denoising convolutional neural network scheme, which realized a sensing distance of 10 km with a spatial resolution of 3 m ^[29].

However, in large-scale engineering infrastructures, fiber deployment typically adopts U-shaped or Hilbert-curve configurations, necessitating length of sensing fiber several times

infrastructures. Despite employing advanced denoising schemes in Raman distributed fiber sensing [30-32], achieving a further breakthrough in sensing distance remains challenging under meter-scale spatial resolution constraints.”

- 2) In the section of **Introduction, page 2, right column, paragraph 4, line 34**, add “Pulse coding schemes have demonstrated significant advantages in long-distance Raman distributed optical fiber sensing system. On one hand, by encoding the detection optical pulses, this technology can significantly increase the optical power injected into the sensing fiber while avoiding fiber nonlinear effects, thereby enhancing the intensity of effective sensing signals at the level of signal generation mechanism. On the other hand, at the data processing level, it leverages the separability between coded signals and uncoded noise to effectively suppress noise and improve the SNR of system.

Currently, pulse coding schemes applied in Raman distributed optical fiber sensing system are primarily classified into five categories, which specifically include the Simplex coding scheme, Pre-Shaped Simplex Coding scheme, low-repetition-rate cyclic pulse coding scheme, Genetic-optimised aperiodic coding scheme, Derived Sequences coding scheme.

Simplex coding scheme, as a linear coding technique based on Hadamard matrix transformation, has its code length (in L bits) determining the number of detection signals required (L groups). For instance, in 2006, J. Park, *et al.* achieved a spatial resolution of 17.0 m, a temperature resolution of 3.0 °C, and a total of 2176 effective sensing points over a sensing distance of 37.0 km by utilizing this coding scheme and link optimization techniques [31].

Low-repetition-rate cyclic pulse coding scheme generates Simplex coding-based cyclic coding sequences using an acousto-optic modulator (AOM), enabling pulses to be injected into the optical fiber periodically at a low repetition rate. For example, in 2011, M. A. Soto, *et al.* combined the low-repetition-rate quasi-periodic cyclic pulse coding with a high-power fiber laser, achieving a spatial resolution of 1.0 m, a temperature resolution of 3.0 °C, and 26,000 effective sensing points over a sensing distance of 26.0 km [27].

Pre-Shaped Simplex Coding scheme is based on simplex coding technology and adopts a linearly increasing profile to adjust pulse amplitude, compensating for the amplitude variation caused by EDFA to suppress the transient effect of the EDFA and improve system performance. For example, in 2017, J. B. Rosolem, *et al.* achieved a spatial resolution of 10.0 m, a temperature resolution of 8.4 °C, and 6,200 effective sensing points over a sensing distance of 62.0 km by using the pre-shaped Simplex coding and a gain-controlled erbium-doped fiber amplifier [32].

Genetic-optimised aperiodic coding scheme utilizes a distributed genetic algorithm (DGA) to generate a single-sequence aperiodic code (GO-code), which is then converted into an optical pulse sequence and injected into the optical fiber. For example, in 2020, X. Z. Sun, *et al.* achieved a spatial resolution of 1.0 m, a temperature resolution of 3.9 °C, and 39,000 effective sensing points over a sensing distance of 39.0 km based on the Genetic-optimised aperiodic coding scheme [29].

Derived Sequences coding scheme addresses the transient effect of EDFA through a derived sequence decoding method, thereby enhancing the performance of long-distance sensing. For instance, in 2022, D. D. Chai, *et al.* achieved a spatial resolution of 5.0 m, a temperature resolution of 2.5 °C, and 8,800 effective sensing points over a sensing distance of 44.0 km based on the Derived Sequences coding scheme [33].”

6. The paper clearly states that the acquisition time for each Raman scattering signal is 63 minutes, and a total of four Raman scattering signals need to be acquired during the decoding stage. Given the prolonged acquisition duration, how is the stability of the experimental temperature and optical instruments performance ensured?

Reply: We would like to express our sincere gratitude to the reviewers for their professional comments. In the proposed scheme of this manuscript, the acquisition time for each Raman backscattered signal is 63 minutes. Therefore, in the experiment, the measurement time required for demodulating the entire distributed temperature curve lasts for several hours. To ensure the environmental temperature stability of the sensing fiber during the long-term experiment, the entire 70.0 km sensing fiber is placed in an ultra-clean constant-temperature laboratory, where the ambient temperature

is maintained at 24.0 ± 1.0 °C. The FUT is placed in a high-precision constant-temperature water bath with a temperature fluctuation of less than 0.10 °C. Thus, the above experimental setup can guarantee the temperature stability of the testing environment for the sensing fiber during long-term measurements. In addition, the experimental equipment used in this manuscript, such as the detection light source, APD, and acquisition card, can operate continuously for up to 2000 hours, which is sufficient to ensure long-term continuous measurements.

Thank you for your suggestion. We have made the following modifications in our revised manuscript:

- 1) In the section of **Experimental setup**, page 9, right column, paragraph 3, line 20, add “To ensure the environmental temperature stability of the sensing fiber during the long-term experiment, the entire 70.0 km sensing fiber is placed in an ultra-clean constant-temperature laboratory, where the ambient temperature is maintained at 24.0 ± 1.0 °C. The FUT is placed in a high-precision constant-temperature water bath with a temperature fluctuation of less than 0.10 °C. Thus, the above experimental setup can guarantee the temperature stability of the testing environment for the sensing fiber during long-term measurements.”

7. Discrepancies in typography consistency are observed in Fig.3-Fig.7, particularly regarding the font sizes of axis captions and intra-figure text. For example, the axis labels in Fig.6-Fig.7 and the text in Fig.7(b) exhibit notably larger font sizes compared to other figures. It is recommended to standardize the typography across all figures to ensure visual coherence and readability.

Reply: We would like to express our sincere gratitude to the reviewers for their valuable suggestions. Based on the reviewer’s comments, we have revised the image formats of Fig. 6 and Fig. 7, and merged them into a single figure.

Thank you for your suggestion. We have made the following modifications in our revised manuscript:

- 1) In the section of **Experimental results**, page 14, change “Fig .6, Fig .7” to “Fig .6”

Fig.6 Experimental comparison results of temperature sensing. (a1) Temperature demodulation results at room temperature based on single-pulse scheme. **(a2)** Temperature demodulation results at room temperature based on distorted detection signals. **(a3)** Temperature demodulation results at room temperature based on proposed EAC-coding scheme. **(b1)** SNR based on single-pulse scheme. **(b2)** SNR based on EAC-coding scheme (without pre-processing). **(b3)** SNR based on proposed EAC-coding scheme. **(c1)** Temperature resolution based on single-pulse scheme. **(c2)** Temperature resolution based on EAC-coding scheme (without pre-processing). **(c3)** Temperature resolution based on proposed EAC-coding scheme. **(d1)** Temperature demodulation results at 50 °C based on EAC-coding scheme and EAC-coding scheme (without pre-processing). **(d2)** Temperature demodulation results at 70 °C based on EAC-coding scheme and EAC-coding scheme (without pre-processing). **(d3)** Temperature demodulation results at 90 °C based on EAC-coding scheme and EAC-coding scheme (without pre-processing).

8. Some of the chart labels are not clear enough. For example, the text in Fig.2 is undersized and the color contrast is insufficient, while the caption text in Fig.3(b) appears blurred. It is recommended that the format and size of all figures be standardized to improve visual clarity and ensure that labels and annotations are clear

and easy to read and consistent throughout the manuscript.

Reply: We would like to sincerely thank the reviewer for the valuable suggestions. In accordance with the reviewers' comments, we have adjusted the text size and color in Fig. 2, and modified the text size in Fig. 3 (b).

Thank you for your suggestion. We have made the following modifications in our revised manuscript:

- 1) In the section of **Experimental setup**, page 8, right column, change “Fig .2” to “Fig.2”

Fig. 2. Experimental setup based on proposed EAC-coding scheme for Raman distributed optical fiber sensing.

- 2) In the section of **Experimental setup**, page 10, left column, change “Fig .3” to “Fig.3”

Fig. 3. Timing of detection signals and autocorrelation function. (a1) Timing of the detection signal u before and after transient effects. (a2) Timing of the

detection signal \bar{u} before and after transient effects. (a3) Timing of the detection signal w before and after transient effects. (a4) Timing of the detection signal \bar{w} before and after transient effects. (b1) Autocorrelation function of the detection signals before transient effects. (b2) Autocorrelation function of the detection signals after transient effects.

Responds to the reviewers' comments:

Reviewer #2 (Comments to the Author):

Recommendations for the Author(s):

The authors have demonstrated a record-long-distance distributed temperature sensing system based on Raman scattering. By combining Golay-encode autocorrelation processing, EDFA-transient-effect mitigation, and Haar wavelet denoising, the performance of the sensing system has been significantly enhanced, achieving 1.58m spatial resolution and 0.88°C measurement accuracy at a maximum sensing range of 70km. This work represents a solid engineering breakthrough in long-distance distributed temperature sensing and has the potential to generate societal impacts by providing new capabilities to infrastructure monitoring. Therefore, this manuscript can be accepted after being further improved by addressing the specific comments below.

Reply: Thank you very much for your evaluation of the content of this research and your constructive suggestions on the manuscript. We are also very grateful to the reviewers for their recognition of the innovation of the proposal presented in this manuscript. These comments are of great value and will be very helpful for revising and improving the manuscript as well as guiding the research work. We have carefully studied your comments and made revisions accordingly, hoping that the revised version will meet your requirements.

1. The introduction should be significantly optimized to clearly state the key innovation of this work. For example, the authors mentioned several existing denoising techniques, but their limitations and the corresponding advantages of Haar wavelet denoising is not mentioned at all.

Reply: Thank you very much for the valuable and professional comments from the reviewers, we fully concur with the reviewer's comments. Based on the reviewer's professional suggestions, we have made substantial revisions to the introduction section, and three parts of revisions have been made in the revised manuscript. First, we have supplemented the discussion on the limitations of existing technologies in the Introduction and Discussion section. Second, we have added content regarding the innovativeness of the EAC-coding technique in the last paragraph of the Introduction. Finally, we have included the advantages of Haar wavelets in the Experimental Results

section. The specific supplementary content in the revised manuscript is as follows.

(1) Based on the professional suggestions of the reviewer, we have supplemented the discussion on the limitations of existing technologies in the Introduction and Discussion section. The specific supplementary content is as follows.

Algorithm denoising schemes aim to improve the SNR of the system by processing optical fiber scattering signals. Essentially derived from mathematical optimization theory, such schemes only denoise signals from the perspective of signal processing without achieving breakthroughs in physical mechanisms. Furthermore, they have limitations including parameter dependence and difficulty in preserving the true characteristics of signals ^[1]. In addition, as one of the current mainstream denoising methods, convolutional neural networks (CNNs) scheme is a typical data-driven approach. It optimizes model parameters using large-scale training datasets, with the goal of achieving effective processing of test data. However, the performance of this method is significantly constrained by the specificity of training data; when there is a notable distribution difference between test signals and the training set, its effect on enhancing sensing performance will be limited.

In contrast, pulse coding technology exhibits unique advantages in long-distance Raman distributed optical fiber sensing, with its strengths lying in two aspects: physical mechanisms and information processing. At the physical mechanism level, by encoding and modulating the detection optical pulses, this technology significantly increases the optical power injected into the sensing fiber without inducing fiber nonlinear effects, thereby enhancing the intensity of the excited Raman scattering signals at the physical source ^[2]. At the signal processing level, pulse coding technology fully leverages the separability between coded signals and background noise in the time/frequency domain. Through matched filtering or demodulation algorithms, it achieves effective noise suppression, thus improving the SNR at the signal processing level ^[3].

Currently, the pulse coding techniques applied in Raman distributed optical fiber sensing system mainly include the following: Simplex coding scheme, Pre-Shaped Simplex coding scheme, Low-repetition-rate cyclic pulse coding scheme, Genetic-optimised aperiodic coding scheme, and Derived Sequences coding scheme. Their key performance indicators are

presented in Table 2. **Simplex coding scheme** is a linear coding technique based on Hadamard matrix transformation, where its code length (L bits) determines the number of detection signal groups required (L groups) ^[4]. The main limitation of this scheme lies in the linear growth of data acquisition volume and computational complexity with code length, which restricts its practical efficiency in applications requiring long code lengths and high-speed sensing. **Low-repetition-rate cyclic pulse coding scheme** utilizes an acousto-optic modulator (AOM) to generate Simplex coding-based cyclic pulse sequences, enabling pulse injection at a low repetition rate. This facilitates the injection of more energy into the sensing fiber, thereby enhancing the intensity of backscattered signals ^[2]. However, higher pulse energy also makes it more likely to induce fiber nonlinear effects, and when approaching the stimulated scattering threshold, it restricts the system's sensing distance and SNR performance. **Pre-Shaped Simplex coding scheme** introduces linear profile shaping of pulse amplitude based on the Simplex code to compensate for the amplitude fluctuation of pulses caused by EDFA, thereby improving system stability ^[5]. Although this scheme effectively suppresses the transient response of the EDFA at the hardware level, its hardware structure is more complex, leading to a significant increase in implementation cost. **Genetic-optimised aperiodic coding scheme** generates aperiodic sequences (GO-code) through optimization via a distributed genetic algorithm (DGA), which are then converted into optical pulse sequences and injected into the sensing fiber ^[3]. The core challenge of this scheme lies in its high dependence on optimization algorithms to search for the optimal code, which typically requires substantial computational resources and iterative testing. **Derived Sequences coding scheme** suppresses the impact of EDFA transient effects on long-distance sensing by introducing a specific derived decoding strategy ^[6]. However, its decoding process imposes a heavy computational burden, and the core algorithm involves high-complexity matrix operations, making it difficult to meet real-time requirements.

Currently, existing pulse coding techniques in Raman distributed optical fiber sensing remain constrained by several prevalent bottlenecks: high hardware implementation complexity, excessive computational overhead in encoding and decoding, limited nonlinear effect suppression capability, and inadequate resistance to signal distortion coupled with poor stability during long-distance transmission. Collectively, these factors constrain the system's

application potential in high-performance scenarios—including low SNR environments and long-distance sensing applications.

To address the aforementioned issues, an Enhanced anti-distortion coding (EAC-coding) scheme is proposed in this manuscript. This scheme effectively overcomes the trade-off dilemma among SNR, sensing distance, and hardware cost in traditional methods through a three-fold synergistic mechanism: compensating for EDFA fluctuation via a preprocessing framework, suppressing transmission losses through coding gain, and finely removing residual noise using wavelet transform. While maintaining relatively low complexity, the EAC-coding scheme significantly enhances the anti-distortion capability and adaptability of the system, providing a new solution for long-distance, high-performance Raman distributed optical fiber sensing.

Table 2. Sensing performance of various coding schemes.

Scheme	Sensing Distance	Spatial Resolution	Sensing Points	Temperature Resolution	Measurement Time
Simplex coding ^[4]	37.0 km	17.0 m	2176	3.0 °C	/
Low-repetition-rate cyclic pulse coding ^[2]	26.0 km	1.0 m	26000	3.0 °C	30 s
Pre-Shaped Simplex Coding ^[5]	62.0 km	10.0 m	6200	8.4 °C	/
Genetic-optimised aperiodic coding ^[3]	39.0 km	1.0 m	39000	3.9 °C	13.6 min
Derived Sequences coding ^[6]	44.0 km	5.0 m	8800	2.5 °C	/
EAC-coding	70.0 km	1.58 m	44303	5.39 °C	252 min

Sensing Points refers to the number of independent spatial sampling points in the fiber optic sensing link where the system can achieve effective measurements. The specific calculation method is the ratio of the sensing distance to the spatial resolution.
Temperature resolution is defined as the minimum temperature change that a system can reliably distinguish, and its calculation method is based on the standard deviation under steady-state conditions.

(2) Based on the professional suggestions of the reviewer, we have added content regarding the innovativeness of the EAC-coding technique in the Introduction section. The specific supplementary content is as follows.

EAC-coding scheme proposed in this manuscript incorporates a novel preprocessing framework for autocorrelation characteristic analysis, waveform analysis and reconstruction of Raman backscattered signals. The preprocessing architecture for autocorrelation characteristic analysis, waveform analysis and reconstruction in this scheme can effectively suppress the transient effects introduced by EDFA, thereby significantly improving the amplitude fidelity and temporal stability of the encoded pulse sequence. This innovative active reconstruction mechanism in the signal domain, which is based on analyzing the waveforms of Raman backscattered signals via forward/inverse Fourier transforms, directly eliminates the phenomenon of

detection pulse distortion existing in traditional demodulation schemes. The scope of action of the EAC-coding scheme encompasses two key aspects: specifically, one is addressing the fundamental limitation of detection signal distortion through physical-layer waveform reconstruction, and the other is performing correlation operations between the proposed coding sequence and the reconstructed Raman signal to restore the ideal cross-correlation characteristics of Golay sequences, and achieve a theoretical gain value, thereby breaking through the SNR-distance trade-off limit. It achieves the coordinated optimization of “waveform correction, correlation characteristic restoration, signal intensity enhancement”, laying a physical layer foundation for long-distance sensing.

Furthermore, the proposed EAC-coding scheme features a synergistically enhanced two-stage noise suppression mechanism, demonstrating innovations in multi-level joint denoising.

First, EAC-coding scheme enhances the SNR of sensing signals through its inherent coding gain. Based on the cross-correlation characteristics of its sequence design, this coding scheme provides a significant inherent gain, which compensates for signal attenuation caused by long-distance transmission and effectively improves the system's baseline SNR. Furthermore, this mechanism not only significantly boosts optical power but also leverages the irrelevance between the decoding algorithm and incoherent noise to achieve preliminary suppression of transmission link losses at the encoding-decoding level.

Second, aiming at the residual noise with specific time-frequency domain distribution characteristics in the decoded signal, this scheme further introduces Haar wavelet transform for adaptive noise filtering. The multi-resolution analysis capability of the Haar wavelet enables accurate capture of the local time-frequency features of noise. By performing sparse representation of the signal at different scales (corresponding to different frequency bands) and time-domain positions, it achieves effective separation between useful signals and non-stationary noise components. While preserving the key details of the distributed temperature signal, this method efficiently eliminates residual noise, significantly enhancing the clarity of the signal in the time-frequency domain and the overall measurement accuracy of the system.

This three-fold synergistic mechanism, consisting of the preprocessing framework compensating for EDFA fluctuation, coding gain suppressing

transmission losses, and wavelet transform finely removing residual noise, systematically breaks through the bottlenecks of traditional schemes in terms of SNR and sensing distance, providing a novel decoding framework for long-distance and high-precision Raman distributed optical fiber sensing.

(3) Based on the professional suggestions of the reviewer, we have included the advantages of Haar wavelets in the Experimental Results section. The specific supplementary content is as follows.

In this study, a denoising algorithm based on the Haar wavelet is adopted to process distributed temperature sensing signals with abrupt change characteristics. The inherent step-like form of the Haar wavelet basis function endows it with excellent feature extraction capability for singular points in the signal, especially for the abrupt temperature field change regions along the sensing fiber. Theoretical analysis indicates that the Haar wavelet transform tends to represent abrupt signals as prominent wavelet coefficients, while its inverse transform process can effectively reconstruct signals with step characteristics. This property is crucial in the denoising process of Raman distributed optical fiber sensing technology: through targeted threshold processing, it can not only effectively suppress noise but also maximize the retention of the signal structure that characterizes abrupt temperature changes, thereby significantly alleviating the deterioration of the system's spatial resolution performance caused by traditional denoising methods.

- [1] L. Xu, *et al.* RDTs noise reduction method based on ICEEMDAN-FE-WSTD. *IEEE Sens. J.* **22**(18): 17854-17863 (2022).
- [2] M. A. Soto, *et al.* Raman-based distributed temperature sensor with 1 m spatial resolution over 26 km SMF using low-repetition-rate cyclic pulse coding. *Opt. Lett.* **36**(13): 2557-2559 (2011).
- [3] X. Z. Sun, *et al.* Genetic-optimised aperiodic code for distributed optical fibre sensors. *Nat. Commun.* **11**(1), 5774 (2020).
- [4] J. Park, *et al.* Raman-based distributed temperature sensor with simplex coding and link optimization. *IEEE Photonics Technol. Lett.* **18**(17): 1879-1881 (2006).
- [5] J. B. Rosolem, *et al.* Raman DTS based on OTDR improved by using gain-controlled EDFA and pre-shaped simplex code. *IEEE Sens. J.* **17**(11): 3346-3353 (2017).
- [6] D. D. Chai, *et al.* Derived sequences decoding approach for long-range distributed temperature sensors. *IEEE Sens. J.* **23**(3): 2204-2210 (2022).

Thank you for your suggestion. We have made the following modifications in our revised manuscript:

- 1) In the section of **Introduction, page 2, right column, paragraph 2, line 3, add** “Algorithm denoising schemes aim to improve the SNR of the system by processing optical fiber scattering signals. For example, M. Yu, *et al.* proposed a dynamic sampling-correction scheme achieving a sensing distance of 20.0 km [23]. H. S. Pradhan, *et al.* proposed a Fourier wavelet regularised deconvolution scheme achieving a sensing distance of 30.0 km with a spatial resolution of 3.0 m [24]. Essentially derived from mathematical optimization theory, such schemes only denoise signals from the perspective of signal processing without achieving breakthroughs in physical mechanisms. Furthermore, they have limitations including parameter dependence and difficulty in preserving the true characteristics of signals [30].

In addition, as one of the current mainstream denoising methods, convolutional neural networks (CNNs) scheme is a typical data-driven approach. It optimizes model parameters using large-scale training datasets, with the goal of achieving effective processing of test data. For example, Z. Zhang, *et al.* proposed a deep 1-D denoising convolutional neural network scheme, which realized a sensing distance of 10 km with a spatial resolution of 3 m [26]. However, the performance of this method is significantly constrained by the specificity of training data; when there is a notable distribution difference between test signals and the training set, its effect on enhancing sensing performance will be limited.”

- 2) In the **(unmodified manuscript) section of Introduction, page 3, left column, paragraph 2, line 3, delete**“ In this work, an enhanced autocorrelation pulse coding (EAC-coding) scheme is proposed, which introduces Golay complementary sequences into Raman distributed optical fiber sensing, and at the same time restores the autocorrelation which is destroyed by the transient effects of erbium-doped fiber amplifier (EDFA) by preprocessing the Raman backscattered signals for analysis and waveform-reconstruction. The proposed scheme further incorporates Haar wavelet denoising algorithm to enhance system performance. In the experiment, a Raman distributed optical fiber sensing with sensing distance of 70.0

km, spatial resolution of 1.58 m and temperature resolution of 5.39 °C is achieved, which provides a new solution for the research of long-distance Raman distributed optical fiber sensing.”

- 3) In the section of **Introduction, page 4, left column, paragraph 3, line 22, add** “Currently, existing pulse coding techniques in Raman distributed optical fiber sensing remain constrained by several prevalent bottlenecks: high hardware implementation complexity, excessive computational overhead in encoding and decoding, limited nonlinear effect suppression capability, and inadequate resistance to signal distortion coupled with poor stability during long-distance transmission. Collectively, these factors constrain the system’s application potential in high-performance scenarios—including low SNR environments and long-distance sensing applications.

To address the aforementioned issues, an Enhanced anti-distortion coding (EAC-coding) scheme is proposed in this paper. This scheme effectively overcomes the trade-off dilemma among SNR, sensing distance, and hardware cost in traditional methods through a three-fold synergistic mechanism: compensating for EDFA fluctuation via a preprocessing framework, suppressing transmission losses through coding gain, and finely removing residual noise using wavelet transform. While maintaining relatively low complexity, the EAC-coding scheme significantly enhances the anti-distortion capability and adaptability of the system, providing a new solution for long-distance, high-performance Raman distributed optical fiber sensing. Experimental results demonstrate that at an ultra-long distance of 70.0 km, the system achieves a spatial resolution of 1.58 m and a temperature resolution of 5.39 °C, with the number of effective sensing points reaching 44,303. To the best of our knowledge, this number of effective sensing points represents the highest level in the field of Raman distributed optical fiber sensing. In addition, the preprocessing architecture within EAC-coding (autocorrelation characteristic analysis, waveform analysis and reconstruction of Raman backscattered signals) significantly enhances the temporal fidelity and transmission stability of encoded sequences by dynamically compensating for EDFA transient gain fluctuation. This further improves the SNR and temperature measurement accuracy performance, enabling the

temperature measurement accuracy to reach 0.91 °C, verifying the engineering feasibility of the scheme.”

- 4) In the **(unmodified manuscript)** section of **Experimental results, page 10, right column, paragraph 2, line 29, delete** “Raman distributed optical fiber sensing has emerged as a pivotal technology for continuous temperature monitoring in long-range infrastructure and environmental applications. However, its practical implementation over distances exceeding 30 km has been fundamentally hindered by the inherently low signal-to-noise ratio (SNR) of Raman scattering signals, which creates a critical trade-off between sensing distance and spatial resolution. In this study, we present a novel Raman distributed optical fiber sensing system utilizing the proposed EAC-coding scheme, which effectively suppresses transient effects and significantly enhances the SNR of system while incorporating Haar wavelet. This system enables long-distance Raman distributed temperature measurement with high spatial resolution, achieving precise temperature demodulation and accurate identification of FUT position and length. The experimental results demonstrate that the proposed scheme achieves a spatial resolution of 1.58 m, temperature resolution of 5.39 °C, and temperature measurement accuracy of 0.88 °C at the sensing distance of 70.0 km, establishing a new approach for long-distance Raman distributed optical fiber sensing system. Future research will integrate more advanced denoising techniques to extend the system's sensing distance while preserving its spatial resolution capabilities.”
- 5) In the section of **Experimental results, page 18, left column, paragraph 2, line 10, add** “EAC-coding scheme proposed in this paper incorporates a novel preprocessing framework for autocorrelation characteristic analysis, waveform analysis and reconstruction of Raman backscattered signals. The preprocessing architecture for autocorrelation characteristic analysis, waveform analysis and reconstruction in this scheme can effectively suppress the transient effects introduced by EDFA, thereby significantly improving the amplitude fidelity and temporal stability of the encoded pulse sequence. This innovative active reconstruction mechanism in the signal domain, which is based on analyzing the waveforms of Raman backscattered signals via forward/inverse

Fourier transforms, directly eliminates the phenomenon of detection pulse distortion existing in traditional demodulation schemes. The scope of action of the EAC-coding scheme encompasses two key aspects: specifically, one is addressing the fundamental limitation of detection signal distortion through physical-layer waveform reconstruction, and the other is performing correlation operations between the proposed coding sequence and the reconstructed Raman signal to restore the ideal cross-correlation characteristics of Golay sequences, and achieve a theoretical gain value, thereby breaking through the SNR-distance trade-off limit. It achieves the coordinated optimization of “waveform correction, correlation characteristic restoration, signal intensity enhancement”, laying a physical layer foundation for long-distance sensing.

Furthermore, the proposed EAC-coding scheme features a synergistically enhanced two-stage noise suppression mechanism, demonstrating innovations in multi-level joint denoising.

First, EAC-coding scheme enhances the SNR of sensing signals through its inherent coding gain. Based on the cross-correlation characteristics of its sequence design, this coding scheme provides a significant inherent gain, which compensates for signal attenuation caused by long-distance transmission and effectively improves the system's baseline SNR. Furthermore, this mechanism not only significantly boosts optical power but also leverages the irrelevance between the decoding algorithm and incoherent noise to achieve preliminary suppression of transmission link losses at the encoding-decoding level.

Second, aiming at the residual noise with specific time-frequency domain distribution characteristics in the decoded signal, this scheme further introduces Haar wavelet transform for adaptive noise filtering. The multi-resolution analysis capability of the Haar wavelet enables accurate capture of the local time-frequency features of noise. By performing sparse representation of the signal at different scales (corresponding to different frequency bands) and time-domain positions, it achieves effective separation between useful signals and non-stationary noise components. While preserving the key details of the distributed temperature signal, this method efficiently eliminates residual noise,

significantly enhancing the clarity of the signal in the time-frequency domain and the overall measurement accuracy of the system.

This three-fold synergistic mechanism, consisting of the preprocessing framework compensating for EDFA fluctuation, coding gain suppressing transmission losses, and wavelet transform finely removing residual noise, systematically breaks through the bottlenecks of traditional schemes in terms of SNR and sensing distance, providing a novel decoding framework for long-distance and high-precision Raman distributed optical fiber sensing.”

- 6) In the section of **Experimental results, page 13, left column, paragraph 2, line 18**, add “In this study, a denoising algorithm based on the Haar wavelet is adopted to process distributed temperature sensing signals with abrupt change characteristics. The inherent step-like form of the Haar wavelet basis function endows it with excellent feature extraction capability for singular points in the signal, especially for the abrupt temperature field change regions along the sensing fiber.”
- 7) In the section of **Introduction, page 3, left column, paragraph 3, line 27**, add “The main limitation of this scheme lies in the linear growth of data acquisition volume and computational complexity with code length, which restricts its practical efficiency in applications requiring long code lengths and high-speed sensing.”
- 8) In the section of **Introduction, page 3, right column, paragraph 1, line 9**, add “However, higher pulse energy also makes it more likely to induce fiber nonlinear effects, and when approaching the stimulated scattering threshold, it restricts the system’s sensing distance and SNR performance.”
- 9) In the section of **Introduction, page 3, right column, paragraph 2, line 25**, add “Although this scheme effectively suppresses the transient response of the EDFA at the hardware level, its hardware structure is more complex, leading to a significant increase in implementation cost.”
- 10) In the section of **Introduction, page 4, left column, paragraph 1, line 3**, add “The core challenge of this scheme lies in its high dependence on optimization algorithms to search for the optimal code, which typically requires substantial computational resources

and iterative testing.”

- 11) In the section of **Introduction, page 4, left column, paragraph 2, line 17**, add “However, this scheme’s decoding process imposes a heavy computational burden, and the core algorithm involves high-complexity matrix operations, making it difficult to meet real-time requirements.”

2. As the core innovation of this work, the uniqueness of Golay-encode autocorrelation processing is insufficiently explained. How does it compare quantitatively to other well established coding methods such as Simplex coding and genetic-optimized aperiodic coding? For example, maybe the discussion can be made from the perspective of coding gain and measurement time. After all, the physical reason that the record-breaking performance can be achieved should be explicitly pointed out.

Reply: We sincerely appreciate the reviewers' professional and valuable suggestions. We believe that the innovation of this manuscript has been significantly enhanced after revisions based on your comments. In accordance with the reviewers' professional advice, we have compared our proposed scheme with other coding methods such as simplex coding and Genetic-optimised aperiodic coding in terms of coding gain, measurement time, and system complexity. The specific revisions are as follows.

In terms of coding gain, the EAC-coding scheme proposed in this manuscript exhibits significant advantages in SNR improvement. The core of this scheme lies in the analysis and accurate reconstruction of the waveform of Raman backscattered signals, which effectively compensates for the distortion of detection signals caused by transient effects. This mechanism enables EAC-coding to overcome the performance bottleneck of traditional derived sequences coding schemes induced by signal distortion, thereby achieving a coding gain that is closer to the theoretical limit. Specifically, existing derived sequences coding schemes construct coding sequences based on Golay complementary sequences and perform correlation decoding using their derived sequences. The upper limit of their theoretical coding gain is $\sqrt{L}/2$ (where L denotes the length of the coding sequence) ^[1]. However, due to the unavoidable distortion (especially transient distortion) in actual detection signals, the results of their correlation operations are degraded, leading to a significant gap between the actual coding gain obtained and this theoretical value. In contrast, the EAC-coding scheme, through the

autocorrelation characteristic analysis, waveform analysis and reconstruction of Raman backscattered signals, significantly suppresses the impact of transient distortion on the detected waveform, allowing its coding gain to approach the theoretical upper limit of $\sqrt{L}/2$. Benefiting from this, the proposed system ultimately achieves the capability of temperature sensing in a long-distance Raman distributed optical fiber sensing system with a sensing distance of 70.0 km.

In terms of measurement time, the core advantage of the EAC-coding scheme stems from its extremely low decoding computational complexity. This scheme achieves signal reconstruction solely through Fourier transform, which significantly reduces the computational cost and time delay of real-time data processing, thereby effectively improving the overall response speed of the system. Details are as follows: For a Simplex code with a code length of L , it is necessary to inject L groups of detection signals into the optical fiber and collect L groups of Raman backscattered signals; its theoretical minimum acquisition time is given by $T_{Simplex}=L \times t$ (where t denotes the acquisition time of a single group of Raman backscattered curves) [2]. Although Genetic-optimised aperiodic coding only requires injecting one group of detection signals into the optical fiber, it necessitates additional time to design coding sequences based on the distributed genetic algorithm. Derived Sequences coding requires injecting 4 groups of detection signals into the optical fiber (theoretical acquisition time of $4t$), and during the decoding stage, it involves complex procedures such as extracting the attenuation envelope, establishing a compensation model, and solving the compensation envelope via matrix operations—all of which increase the time cost of the decoding process. The EAC-coding scheme proposed in this manuscript performs coding based on Golay complementary sequences, with a theoretical acquisition time of $4t$, and only needs Fourier transform to complete the analysis and reconstruction of the Raman backscattered signal waveform. This significant efficiency improvement at the algorithm level greatly reduces the delay in real-time data processing, effectively enhances the system response speed, and provides crucial real-time performance guarantees for the final realization of a 70.0 km long-distance Raman distributed optical fiber sensing system.

In terms of system complexity, the core innovation of the EAC-coding scheme lies in achieving a high degree of simplification in the decoding process through algorithm optimization, while keeping the hardware

architecture unchanged. This scheme only needs to perform efficient Fourier transform operations to complete signal reconstruction, which significantly reduces the algorithmic complexity and computational burden of real-time data processing. The specific reasons are as follows. Simplex coding scheme requires the injection of L groups of detection signals, resulting in a long acquisition time and a large volume of raw data. This not only imposes high requirements on hardware performance (such as data acquisition rate and storage capacity) and long-term stability, but also the large-scale data processing involved in its decoding process significantly increases the computational complexity. Genetic-optimised aperiodic coding scheme greatly shortens the acquisition time without increasing hardware costs. However, due to its unique design, it is necessary to design coding sequences based on a distributed genetic algorithm and conduct extensive tests to find the optimal coding sequence. Derived sequences coding scheme also does not require additional hardware overhead. However, this scheme introduces high computational complexity in the decoding stage, as it relies on computationally expensive matrix operations to generate and process derived sequences.

[1] D. D. Chai, *et al.* Derived sequences decoding approach for long-range distributed temperature sensors. *IEEE Sens. J.* **23**(3): 2204-2210 (2022).

[2] J. Park, *et al.* Raman-based distributed temperature sensor with simplex coding and link optimization. *IEEE Photonics Technol. Lett.* **18**(17): 1879-1881 (2006).

Thank you for your suggestion. We have made the following modifications in our revised manuscript:

- 1) In the section of **Discussion**, **page 19**, **right column**, **paragraph 4**, **line 7**, **add** “Currently, the pulse coding techniques applied in Raman distributed optical fiber sensing system mainly include the following: Simplex coding scheme, Pre-Shaped Simplex coding scheme, Low-repetition-rate cyclic pulse coding scheme, Genetic-optimised aperiodic coding scheme, and Derived Sequences coding scheme. Their key performance indicators are presented in Table 2.”

Table 2. Sensing performance of various coding schemes.

Scheme	Sensing Distance	Spatial Resolution	Sensing Points	Temperature Resolution	Measurement Time
Simplex coding ^[31]	37.0 km	17.0 m	2176	3.0 °C	/
Low-repetition-rate cyclic pulse coding ^[27]	26.0 km	1.0 m	26000	3.0 °C	30 s
Pre-Shaped Simplex Coding ^[32]	62.0 km	10.0 m	6200	8.4 °C	/
Genetic-optimised aperiodic coding ^[29]	39.0 km	1.0 m	39000	3.9 °C	13.6 min
Derived Sequences coding ^[33]	44.0 km	5.0 m	8800	2.5 °C	/
EAC-coding	70.0 km	1.58 m	44303	5.39 °C	252 min

Sensing Points refers to the number of independent spatial sampling points in the fiber optic sensing link where the system can achieve effective measurements. The specific calculation method is the ratio of the sensing distance to the spatial resolution.
Temperature resolution is defined as the minimum temperature change that a system can reliably distinguish, and its calculation method is based on the standard deviation under steady-state conditions.

We have compared our proposed scheme with other coding methods such as simplex coding and Genetic-optimised aperiodic coding in terms of coding gain, measurement time, and system complexity. The specific revisions are as follows.

In terms of coding gain, the EAC-coding scheme proposed in this paper exhibits significant advantages in SNR improvement. The core of this scheme lies in the analysis and accurate reconstruction of the waveform of Raman backscattered signals, which effectively compensates for the detection signals distortion caused by the transient effects of the system. This mechanism enables EAC-coding to overcome the performance bottleneck of traditional derived sequences coding schemes induced by signals distortion, achieving a coding gain closer to the theoretical limit. Specifically, existing derived sequence coding scheme construct coding sequences based on Golay complementary sequences and perform correlation decoding using their derived sequences. The upper limit of their theoretical coding gain is $\sqrt{L}/2$ (where L denotes the length of the coding sequence). However, due to the unavoidable distortion (especially transient distortion) in actual detection signals, the results of their correlation operations deteriorate, leading to a significant gap between the actually achieved coding gain and this theoretical value. In contrast, the EAC-coding scheme, through the aforementioned signals analysis and reconstruction mechanism, significantly suppresses the impact of transient distortion on the detection signals waveform, thereby enabling its coding gain to approach the theoretical upper limit of $\sqrt{L}/2$. Benefiting from this, the proposed system ultimately achieves temperature sensing capability in a long-distance Raman distributed optical fiber sensing system with a sensing distance of 70.0 km.

In terms of measurement time, the core advantage of the EAC-coding scheme stems from its extremely low decoding computational complexity. This scheme achieves signals reconstruction solely through Fourier transform, which significantly reduces the computational cost and time delay of real-time data processing, thereby effectively improving the overall response speed of the system. Specifically, for a Simplex code with length L , it is necessary to inject L groups of detection signals into the optical fiber and collect L groups of Raman backscattered signals, with its theoretical minimum acquisition time being $T_{Simplex}=L \times t$ (where t denotes the acquisition time of one group of Raman backscattered curves). Although the Genetic-optimised aperiodic coding scheme only requires injecting one group of detection signals into the optical fiber, it needs additional time to design the coding sequence based on the distributed genetic algorithm. The Derived sequences coding scheme requires injecting 4 groups of detection signals into the optical fiber, with a theoretical acquisition time of $4t$. However, in the decoding stage, it involves complex steps such as extracting the attenuation envelope, establishing a compensation model, and solving the compensation envelope via matrix operations, which increases the time cost of the decoding process. The EAC-coding scheme proposed in this paper performs encoding based on Golay complementary sequences, with a theoretical acquisition time of $4t$, and only needs Fourier transform to complete the analysis and reconstruction of the Raman backscattered signals waveform. This significant efficiency improvement at the algorithm level greatly reduces the real-time data processing delay and effectively enhances the system response speed, providing crucial real-time guarantee for the final realization of temperature sensing in a 70.0 km long-distance Raman distributed optical fiber sensing system.

In terms of system complexity, the core innovation of the EAC-coding scheme lies in achieving a high degree of simplification in the decoding process through algorithm optimization, while keeping the hardware architecture unchanged. This scheme only needs to perform efficient Fourier transform operations to complete signal reconstruction, which significantly reduces the algorithmic complexity and computational burden of

real-time data processing. The specific reasons are as follows. Simplex coding scheme requires the injection of L groups of detection signals, resulting in a long acquisition time and a large volume of raw data. This not only imposes high requirements on hardware performance (such as data acquisition rate and storage capacity) and long-term stability, but also the large-scale data processing involved in its decoding process significantly increases the computational complexity. Genetic-optimised aperiodic coding scheme greatly shortens the acquisition time without increasing hardware costs. However, due to its unique design, it is necessary to design coding sequences based on a distributed genetic algorithm and conduct extensive tests to find the optimal coding sequence. Derived sequences coding scheme also does not require additional hardware overhead. However, this scheme introduces high computational complexity in the decoding stage, as it relies on computationally expensive matrix operations to generate and process derived sequences.

EAC-coding scheme proposed in this paper incorporates a novel preprocessing framework for autocorrelation characteristic analysis, waveform analysis and reconstruction of Raman backscattered signals. This framework significantly improves the fidelity and transmission stability of the encoded pulse sequence by dynamically compensating for the transient gain fluctuation of EDFA. This technology leverages the inherent gain of EAC-coding pulse coding to effectively offset fiber scattering attenuation during 70.0 km long-distance transmission, achieving a significant improvement in the baseline SNR. On this basis, a two-stage noise suppression method is further constructed. The first-stage processing involves compensating for signal attenuation using the theoretical coding gain of EAC-coding. For the second-stage processing, a Haar wavelet denoising algorithm is adopted. Its step-matching characteristic enables accurate extraction of the abrupt temperature change features of the sensing fiber and adaptive filtering of the residual time/frequency domain coupled noise in the decoded signal. This three-fold synergistic mechanism, consisting of the preprocessing framework compensating for EDFA fluctuation, coding gain suppressing transmission losses, and wavelet transform

finely removing residual noise, breaks through the theoretical trade-off bottleneck between SNR and sensing distance in traditional schemes. Experiment results indicate that, at an ultra-long distance of 70.0 km, this scheme can achieve a spatial resolution of 1.58 m, a temperature resolution of 5.39 °C, and an effective number of sensing points reaching 44,303, verifying the engineering feasibility of the scheme. To the best of our knowledge, the number of effective sensing points achieved by this scheme ranks the highest in the field of Raman distributed optical fiber sensing systems.”

3. To let the reader to better evaluate the performance in this work, a quantitative and fair comparison with the state of the art should be made. If possible, a figure of merit should be given that normalizes the measurement time, considering that the performance in this work is obtained over one hour.

Reply: Thank you for your valuable suggestions, which are of great help in enhancing the innovation of this manuscript and guiding the research work to be carried out in the future. Based on the reviewer's professional suggestions, we have quantitatively analyzed the measurement time in the discussion section. Furthermore, we have compared and elaborated on the key performance indicators of our proposed scheme with those of existing coding techniques. The revised content is as follows.

Currently, in the field of Raman distributed optical fiber sensing, a performance comparison between conventional coding schemes and the proposed EAC-coding scheme is presented in Table 3.

Simplex coding scheme, as a linear coding technique based on Hadamard matrix transformation, has its code length (in L bits) determining the number of detection signals required (L groups). By leveraging the Simplex coding and link optimization technology, a spatial resolution of 17.0 m, a temperature resolution of 3.0 °C, and an effective number of sensing points of 2176 have been achieved at a sensing distance of 37.0 km ^[1].

Low-repetition-rate cyclic pulse coding scheme generates Simplex coding-based cyclic coding sequences using an acousto-optic modulator (AOM), enabling pulses to be injected into the optical fiber periodically at a low repetition rate. The combination of low-repetition-rate cyclic pulse coding and a high-power fiber laser achieves a spatial resolution of 1.0 m, a temperature resolution of 3.0 °C, an effective number of sensing points of 26,000, and a measurement time of 30 s at a sensing distance of 26.0 km ^[2].

Pre-Shaped Simplex Coding scheme is based on simplex coding technology and adopts a linearly increasing profile to adjust pulse amplitude, compensating for the amplitude variation caused by EDFA to suppress the transient effect of EDFA and improve system performance. Based on pre-shaped Simplex coding and a gain-controlled EDFA, a spatial resolution of 10.0 m, a temperature resolution of 8.4 °C, and an effective number of sensing points of 6,200 have been achieved at a sensing distance of 62.0 km [3].

Genetic-optimised aperiodic coding scheme utilizes a distributed genetic algorithm (DGA) to generate a single-sequence aperiodic code (GO-code), which is then converted into an optical pulse sequence and injected into the optical fiber. This scheme achieved a spatial resolution of 1.0 m, a temperature resolution of 3.9 °C, an effective number of sensing points of 39,000, and a measurement time of 13.6 min at a sensing distance of 39.0 km [4].

Derived Sequences coding scheme addresses the transient effect of EDFA through a derived sequence decoding method, thereby enhancing the performance of long-distance sensing. Based on the Derived Sequences coding scheme, a spatial resolution of 5.0 m, a temperature resolution of 2.5 °C, and an effective number of sensing points of 8,800 have been achieved at a sensing distance of 44.0 km [5].

EAC-coding scheme proposed in this manuscript incorporates a novel preprocessing framework for autocorrelation characteristic analysis, waveform analysis and reconstruction of Raman backscattered signals. This framework significantly improves the fidelity and transmission stability of the encoded pulse sequence by dynamically compensating for the transient gain fluctuation of EDFA. This technology leverages the inherent gain of EAC-coding pulse coding to effectively offset fiber scattering attenuation during 70.0 km long-distance transmission, achieving a significant improvement in the baseline SNR. On this basis, a two-stage noise suppression method is further constructed. The first-stage processing involves compensating for signal attenuation using the theoretical coding gain of EAC-coding. For the second-stage processing, a Haar wavelet denoising algorithm is adopted. Its step-matching characteristic enables accurate extraction of the abrupt temperature change features of the sensing fiber and adaptive filtering of the residual time/frequency domain coupled noise in the decoded signal. This three-fold synergistic mechanism,

consisting of the preprocessing framework compensating for EDFA fluctuation, coding gain suppressing transmission losses, and wavelet transform finely removing residual noise, breaks through the theoretical trade-off bottleneck between SNR and sensing distance in traditional schemes. Experiment results indicate that, at an ultra-long distance of 70.0 km, this scheme can achieve a spatial resolution of 1.58 m, a temperature resolution of 5.39 °C, and an effective number of sensing points reaching 44,303, verifying the engineering feasibility of the scheme.

Furthermore, the measurement time of the scheme proposed in this manuscript is mainly limited by the massive data volume (with the effective number of sensing points reaching 44,303) and the hardware accumulation processing time of the acquisition card, which are caused by its long sensing distance and high spatial resolution—rather than the complexity of its processing architecture. The current measurement time (252 min) is constrained by the processing capability of the data acquisition card (acquisition rate: 1.0 GSa/s; storage depth: 512 Mpts). The massive data volume (approximately 700 k points per frame corresponding to a 70.0 km sensing distance) exceeds the on-board real-time processing limit. In the future, high-performance acquisition cards will be upgraded to optimize the data processing pipeline, thereby eliminating the delay deterioration caused by hardware bottlenecks.

Table 3. Sensing performance of various coding schemes.

Scheme	Sensing Distance	Spatial Resolution	Sensing Points	Temperature Resolution	Measurement Time
Simplex coding ^[1]	37.0 km	17.0 m	2176	3.0 °C	/
Low-repetition-rate cyclic pulse coding ^[2]	26.0 km	1.0 m	26000	3.0 °C	30 s
Pre-Shaped Simplex Coding ^[3]	62.0 km	10.0 m	6200	8.4 °C	/
Genetic-optimised aperiodic coding ^[4]	39.0 km	1.0 m	39000	3.9 °C	13.6 min
Derived Sequences coding ^[5]	44.0 km	5.0 m	8800	2.5 °C	/
EAC-coding	70.0 km	1.58 m	44303	5.39 °C	252 min

Sensing Points refers to the number of independent spatial sampling points in the fiber optic sensing link where the system can achieve effective measurements. The specific calculation method is the ratio of the sensing distance to the spatial resolution.
Temperature resolution is defined as the minimum temperature change that a system can reliably distinguish, and its calculation method is based on the standard deviation under steady-state conditions.

[1] J. Park, *et al.* Raman-based distributed temperature sensor with simplex coding and link optimization. *IEEE Photonics Technol. Lett.* **18**(17): 1879-1881 (2006).

[2] M. A. Soto, *et al.* Raman-based distributed temperature sensor with 1 m spatial resolution over 26 km SMF using low-repetition-rate cyclic pulse coding. *Opt. Lett.* **36**(13): 2557-2559 (2011).

- [3] J. B. Rosolem, *et al.* Raman DTS based on OTDR improved by using gain-controlled EDFA and pre-shaped simplex code. *IEEE Sens. J.* **17**(11): 3346-3353 (2017).
- [4] X. Z. Sun, *et al.* Genetic-optimised aperiodic code for distributed optical fibre sensors. *Nat. Commun.* **11**(1), 5774 (2020).
- [5] D. D. Chai, *et al.* Derived sequences decoding approach for long-range distributed temperature sensors. *IEEE Sens. J.* **23**(3): 2204-2210 (2022).

Thank you for your suggestion. We have made the following modifications in our revised manuscript:

- 1) In the section of **Experimental setup, page 9, left column, paragraph 2, line 21**, add “Furthermore, the measurement time of the scheme proposed in this paper is mainly limited by the massive data volume (with the effective number of sensing points reaching 44,303) and the hardware accumulation processing time of the acquisition card, which are caused by its long sensing distance and high spatial resolution—rather than the complexity of its processing architecture. The current measurement time (252 min) is constrained by the processing capability of the data acquisition card (acquisition rate: 1.0 GSa/s; storage depth: 512 Mpts). The massive data volume (approximately 700 k points per frame corresponding to a 70.0 km sensing distance) exceeds the on-board real-time processing limit. In the future, high-performance acquisition cards will be upgraded to optimize the data processing pipeline, thereby eliminating the delay deterioration caused by hardware bottlenecks.”
- 2) In the section of **Introduction, page 3, left column, paragraph 2, line 10**, add “Currently, pulse coding schemes applied in Raman distributed optical fiber sensing system are primarily classified into five categories, which specifically include the Simplex coding scheme, Pre-Shaped Simplex Coding scheme, low-repetition-rate cyclic pulse coding scheme, Genetic-optimised aperiodic coding scheme, Derived Sequences coding scheme.

Simplex coding scheme, as a linear coding technique based on Hadamard matrix transformation, has its code length (in L bits) determining the number of detection signals required (L groups). For instance, in 2006, J. Park, *et al.* achieved a spatial resolution of 17.0 m, a temperature resolution of 3.0 °C, and a total of 2176 effective

sensing points over a sensing distance of 37.0 km by utilizing this coding scheme and link optimization techniques.

Low-repetition-rate cyclic pulse coding scheme generates Simplex coding-based cyclic coding sequences using an acousto-optic modulator (AOM), enabling pulses to be injected into the optical fiber periodically at a low repetition rate. For example, in 2011, M. A. Soto, *et al.* combined the low-repetition-rate quasi-periodic cyclic pulse coding with a high-power fiber laser, achieving a spatial resolution of 1.0 m, a temperature resolution of 3.0 °C, and 26,000 effective sensing points over a sensing distance of 26.0 km.

Pre-Shaped Simplex Coding scheme is based on simplex coding technology and adopts a linearly increasing profile to adjust pulse amplitude, compensating for the amplitude variation caused by EDFA to suppress the transient effect of the EDFA and improve system performance. For example, in 2017, J. B. Rosolem, *et al.* achieved a spatial resolution of 10.0 m, a temperature resolution of 8.4 °C, and 6,200 effective sensing points over a sensing distance of 62.0 km by using the pre-shaped Simplex coding and a gain-controlled erbium-doped fiber amplifier.

Genetic-optimised aperiodic coding scheme utilizes a distributed genetic algorithm (DGA) to generate a single-sequence aperiodic code (GO-code), which is then converted into an optical pulse sequence and injected into the optical fiber. For example, in 2020, X. Z. Sun, *et al.* achieved a spatial resolution of 1.0 m, a temperature resolution of 3.9 °C, and 39,000 effective sensing points over a sensing distance of 39.0 km based on the Genetic-optimised aperiodic coding scheme.

Derived Sequences coding scheme addresses the transient effect of EDFA through a derived sequence decoding method, thereby enhancing the performance of long-distance sensing. For instance, in 2022, D. D. Chai, *et al.* achieved a spatial resolution of 5.0 m, a temperature resolution of 2.5 °C, and 8,800 effective sensing points over a sensing distance of 44.0 km based on the Derived Sequences coding scheme.”

4. More in-depth discussions should be provided to talk about the physical limit of

current performance and potential ways for further enhancement.

Reply: We gratefully thanks for the precious time you spent making constructive remarks, we fully agree with the reviewers. In the Discussion section of the revised version of this manuscript, we have added content regarding the physical limit of the current performance of the proposed scheme as well as potential approaches for further enhancement. The specific added parts are as follows.

In this proposed scheme, the physical limitations on the optimization of sensing distance and SNR mainly stem from two aspects: the insufficient performance of experimental devices and the physical demodulation principle.

In terms of experimental devices, the maximum coding bit length of the digital delayed pulse generator (DDPG) and the sampling rate of the data acquisition card are key factors restricting the improvement of the sensing distance and SNR of this scheme. The CIQTEK ASG8100 model DDPG used in the experiment has a maximum coding bit length of 99 bits. Since the bit length of Golay complementary sequences is typically 2^n , a coding length of 64 bits was selected in the experiment. The coding length directly determines the coding gain of the system, which in turn affects the extent of improvement in sensing performance. Therefore, it is planned to adopt a DDPG with a higher maximum coding bit length (e.g., ≥ 128 bits) in the future to increase the system's coding length, thereby achieving better sensing distance performance. In addition, the currently used data acquisition card has a sampling rate of 1.0 GSa/s and a maximum number of sampling points of 700,000, corresponding to a theoretical maximum sensing distance of approximately 70.0 km. Future plans involve upgrading to a data acquisition card with a higher sampling rate (e.g., ≥ 2 GSa/s) and a larger number of sampling points (e.g., $\geq 1,000,000$) to eliminate the limitations imposed by hardware devices on the sensing distance.

In terms of the physical demodulation principle, the stimulated scattering threshold and fiber dispersion effect also affect the improvement of sensing distance and SNR in this scheme. Firstly, the stimulated scattering threshold of single-mode fibers is relatively low, which limits the maximum optical power injected into the fiber. As a result, the intensity of the Raman backscattered signal excited in the fiber is relatively weak, leading to a low SNR. In the future, we will attempt to use few-mode fibers to increase the optical power injected into the fiber while minimizing intermodal dispersion as much as possible. Finally, fiber dispersion introduced by long-distance transmission

causes pulse broadening and intersymbol interference, which destroys the orthogonality of sequences. This results in the broadening of demodulated correlation peaks and a reduction in their amplitude, impairing both SNR and distance resolution. This issue can be compensated for by introducing dispersion-compensating fibers.

Thank you for your suggestion. We have made the following modifications in our revised manuscript:

- 1) In the section of **Discussion, page 21, right column, paragraph 3, line 35**, add “In this proposed scheme, the physical limitations on the optimization of sensing distance and SNR mainly stem from two aspects: the insufficient performance of experimental devices and the physical demodulation principle.

In terms of experimental devices, the maximum coding bit length of the digital delayed pulse generator (DDPG) and the sampling rate of the data acquisition card are key factors restricting the improvement of the sensing distance and SNR of this scheme. The CIQTEK ASG8100 model DDPG used in the experiment has a maximum coding bit length of 99 bits. Since the bit length of Golay complementary sequences is typically 2^n , a coding length of 64 bits was selected in the experiment. The coding length directly determines the coding gain of the system, which in turn affects the extent of improvement in sensing performance. Therefore, it is planned to adopt a DDPG with a higher maximum coding bit length (e.g., ≥ 128 bits) in the future to increase the system's coding length, thereby achieving better sensing distance performance. In addition, the currently used data acquisition card has a sampling rate of 1.0 GSa/s and a maximum number of sampling points of 700,000, corresponding to a theoretical maximum sensing distance of approximately 70.0 km. Future plans involve upgrading to a data acquisition card with a higher sampling rate (e.g., ≥ 2 GSa/s) and a larger number of sampling points (e.g., $\geq 1,000,000$) to eliminate the limitations imposed by hardware devices on the sensing distance.

In terms of the physical demodulation principle, the stimulated scattering threshold and fiber dispersion effect also affect the improvement of sensing distance and SNR in this scheme. Firstly, the stimulated scattering threshold of single-mode fibers is relatively

low, which limits the maximum optical power injected into the fiber. As a result, the intensity of the Raman backscattered signal excited in the fiber is relatively weak, leading to a low SNR. In the future, we will attempt to use few-mode fibers to increase the optical power injected into the fiber while minimizing intermodal dispersion as much as possible. Finally, fiber dispersion introduced by long-distance transmission causes pulse broadening and intersymbol interference, which destroys the orthogonality of sequences. This results in the broadening of demodulated correlation peaks and a reduction in their amplitude, impairing both SNR and distance resolution. This issue can be compensated for by introducing dispersion-compensating fibers.”

5. Details of Haar wavelet denoising should be provided. How is it implemented? Does it sacrifice spatial resolution?

Reply: We gratefully thanks for the precious time you spent making constructive remarks, we fully agree with the reviewers. In the revised manuscript, we have added content regarding the implementation method of Haar wavelet denoising and its limitations (specifically, the impact on the spatial resolution of the sensing system). The specific added information is as follows.

The Haar wavelet function generates a set of orthogonal basis functions through translation and scaling, enabling multi-scale decomposition of signals. Its inherent advantage in processing low-complexity signals with abrupt change characteristics makes it particularly suitable for the analysis of Raman backscattered signals.

In the experiment, to further improve the SNR of Raman backscattered signals, we adopted a 6-level decomposition Haar wavelet. An important reason for selecting the Haar wavelet is that the processed signals typically exhibit step-like characteristics, which helps to retain the abrupt change structure of Raman backscattered signals to the greatest extent, thereby slightly mitigating the impact of denoising processes using other wavelet types on the abrupt temperature signals of the system.

Furthermore, in the post-processing stage of temperature demodulation, to further suppress noise and enhance signal smoothness, we additionally applied the db10 (Daubechies wavelet 10) wavelet function with 4-level decomposition. It should be noted that while this step improves signal quality (by reducing temperature fluctuation), it moderately degrades the spatial resolution of the

system. Such parameter adjustment reflects the necessary trade-off between spatial resolution and temperature measurement accuracy.

Thank you for your suggestion. We have made the following modifications in our revised manuscript:

- 1) In the section of **Experimental results, page 13, left column, paragraph 2, line 26, add** “The Haar wavelet function generates a set of orthogonal basis functions through translation and scaling, enabling multi-scale decomposition of signals. Its inherent advantage in processing low-complexity signals with abrupt change characteristics makes it particularly suitable for the analysis of Raman backscattered signals.

In the experiment, to further improve the SNR of Raman backscattered signals, we adopted a 6-level decomposition Haar wavelet. Furthermore, in the post-processing stage of temperature demodulation, to further suppress noise and enhance signal smoothness, we additionally applied the db10 (Daubechies wavelet 10) wavelet function with 4-level decomposition. It should be noted that while this step improves signal quality (by reducing temperature fluctuation), it moderately degrades the spatial resolution of the system. Such parameter adjustment reflects the necessary trade-off between spatial resolution and temperature measurement accuracy.”

6. It is confusing why the temperature-measurement uncertainty can be much larger than the resolution.

Reply: We apologize for the ambiguous expression. In the revised manuscript, we have supplemented two parts of content, namely the definitions of temperature fluctuation and temperature resolution as well as their respective calculation methods. The supplementary content is specified as follows.

(1) **Temperature fluctuation** is expressed as the difference between the measured temperature and the ambient temperature. Specifically, in a Raman distributed fiber optic sensing system, there exists a discrepancy between the demodulated temperature (T') at the sampling points of the sensing fiber and the ambient reference temperature (T_0). This discrepancy is defined as temperature fluctuation ($\Delta T = T' - T_0$). In the experiments of this manuscript, the maximum temperature difference (i.e., the maximum temperature

fluctuation) observed at the end of the 70.0 km sensing fiber was 20.86 °C.

Temperature resolution refers to temperature uncertainty, which characterizes the system's ability to distinguish temperature changes, is generally defined as the standard deviation (σ) of measured temperatures under constant temperature conditions (e.g., a section of reference fiber) ^{[1][2]}. In this experiment, by calculating the standard deviation of temperatures at the reference fiber, the temperature resolution at a sensing distance of 70.0 km was determined to be 5.39 °C.

(2) Theoretically, the reason why the value of temperature fluctuation is greater than that of temperature resolution mainly lies in the difference in the types of errors they characterize:

Temperature fluctuation (ΔT) primarily reflects systematic errors, which stem from inherent biases or calibration errors of the system (such as light source power drift, deviation in the estimation of fiber link loss, errors in the selection of ambient temperature reference points, etc.). Such errors manifest as a relatively fixed offset in measurements (e.g., 20.86 °C in this case), and their magnitude is closely related to the sensing distance (the longer the distance, the more significant the accumulation effect of systematic errors). Moreover, they cannot be eliminated by averaging multiple measurements.

Temperature resolution (σ) mainly characterizes random errors or noise levels (such as shot noise, thermal noise, amplifier noise, optical path noise, etc.). It determines the minimum temperature change that the system can resolve. Random errors follow a statistical distribution in multiple measurements and can be suppressed to a certain extent through techniques such as signal averaging or filtering (e.g., in this case, wavelet denoising is employed to improve the signal-to-noise ratio, thereby enhancing the resolution to 5.39 °C).

[1] X. Liu, *et al.* High-speed and high-resolution YAG fiber based distributed high temperature sensing system empowered by a 2D image restoration algorithm. *Opt. Express*, **31**(4): 6170-6183 (2023).

[2] J. Gasser, *et al.* Distributed temperature sensor combining centimeter resolution with hundreds of meters sensing range. *Opt. Express*, **30**(5): 6768-6777 (2022).

Thank you for your suggestion. We have made the following modifications in our revised manuscript:

1) In the section of **Experimental results**, page 15, left column,

paragraph 2, line 28, add “Temperature fluctuation is expressed as the difference between the measured temperature and the ambient temperature. Specifically, in a Raman distributed fiber optic sensing system, there exists a discrepancy between the demodulated temperature (T') at the sampling points of the sensing fiber and the ambient reference temperature (T_0). This discrepancy is defined as temperature fluctuation ($\Delta T = T' - T_0$).”

- 2) In the section of **Experimental results, page 16, left column, paragraph 3, line 29, add** “Temperature resolution refers to temperature uncertainty, which characterizes the system's ability to distinguish temperature changes, is generally defined as the standard deviation (σ) of measured temperatures under constant temperature conditions (e.g., a section of reference fiber) ^[34-35].”

7. Some minor mistakes should be revised. For example, peak-to-peak ratio from 0.0067 to 0.0006 should be attributed to Haar wavelet denoising instead of to EAC-coding. Results for the single-pulse scheme are shown in Fig.5(b1) instead of Fig.5(b2).

Reply: We sincerely apologize for any misunderstandings caused to the reviewers due to some ambiguous expressions in our manuscript. We have revised all the relevant content in accordance with your comments.

In the experiment, the improvement in the peak-to-peak ratio (from 0.0067 to 0.0006) is jointly determined by the Haar wavelet denoising and the EAC-coding scheme, as shown in Fig. 2. We applied the Haar wavelet denoising algorithm to both the traditional single-pulse scheme and the EAC-coding scheme, with all denoising settings being completely consistent. Fig. 2(b1) presents the Raman backscattered signals of the traditional single-pulse scheme before and after Haar wavelet denoising, where the peak-to-peak ratio is reduced from 0.0118 to 0.0011. Fig. 2(b2) shows the Raman backscattered signals of the EAC-coding scheme before and after Haar wavelet denoising, with the peak-to-peak ratio decreasing from 0.0067 to 0.0006. Therefore, the combination of the EAC-coding and Haar wavelet denoising scheme results in a peak-to-peak ratio of 0.0006. We have revised the relevant sentences in the manuscript to avoid misunderstandings.

Fig.2. (b1) Intensity of Raman backscattered signals based on EAC-coding scheme at the end of sensing fiber. (b2) Intensity of Raman backscattered signals based on conventional single pulse at the end of sensing fiber.

Thank you for your suggestion. We have made the following modifications in our revised manuscript:

- 1) In the section of **Experimental results**, page 13, left column, paragraph 1, line 10, add “(with Haar wavelet).”

Responds to the reviewers' comments:

Reviewer #3 (Comments to the Author):

It is well known that Distributed Anti-Stokes Raman Thermometry (aka Raman OTDR) is an attractive approach for distributed temperature sensing without much cross-sensitivity due to strain (unlike Brillouin distributed sensors). A key challenge for such a Raman OTDR is its operation in the spontaneous Raman scattering regime, which severely limits the peak power that can be used thereby limiting the sensing range to 20 km.

The use of a coded sequence of pulses is a well-established approach in Raman OTDR to increase the backscattered energy without losing spatial resolution. Specifically, Golay encoded sequences are attractive since they use a complementary pair of sequences that theoretically eliminates side lobes leading to excellent spatial resolution and low temperature uncertainty. In this work, the authors present their efforts on utilizing this idea to extend the measurement range to a distance of 70 km which seem impressive at first sight.

However, the above manuscript is not recommended to be published in Nature Communications in its current form due to the following reasons:

Reply: We gratefully thanks for the valuable time the reviewer spent making constructive remarks. These comments are all very professional and helpful for revising and improving our manuscript, as well as the meaningful guidance to our research.

We have read through comments carefully and made corrections. We hope meet with approval. If there are still some misleading statements in the reply, we hope you can give us your opinions again and give us another opportunity to revise the manuscript. We will do our best to improve this work, thanks again to the professional reviewers.

1. There is a fundamental question on the improvement achieved in this work compared to previous work. The authors claim that their work “introduces Golay complementary sequences into Raman distributed optical fiber sensing”, which is not true. In fact, it is surprising that they are not aware of previous work on this (10.1109/JSEN.2022.3229417) from the same University. In general, the authors need to present a thorough literature survey of previous work in this area and provide a clear justification of how their work is different from the previous approaches/results.

Reply: We would like to express our sincere gratitude to the reviewer for their insightful comments. We fully concur with their suggestions and have revised the manuscript accordingly to address all the points raised. Furthermore, we are sorry that we used some imperfect words, which caused ambiguity about the innovation of this manuscript. The innovation of this manuscript lies in proposing a novel Enhanced anti-distortion coding (EAC-coding) demodulation scheme that combines the autocorrelation characteristic analysis with waveform analysis and reconstruction of Raman backscattered signals. This scheme is different from the conventional Golay complementary sequences. Based on the reviewer's professional suggestions, we have made revisions to this manuscript in the following three aspects: First, we have redefined the innovation of this manuscript. Second, we have conducted a more comprehensive investigation of the technologies in this field and performed a detailed comparative analysis between the EAC-coding scheme and the scheme described in the reference for comparison (10.1109/JSEN.2022.3229417). Finally, we have compared the technical characteristics of the proposed EAC-coding scheme with those of the current mainstream coded pulse schemes. The detailed revision contents are as follows.

(1) We have first revised the innovative aspects of this manuscript. EAC-coding scheme proposed in this manuscript incorporates a novel preprocessing framework for autocorrelation characteristic analysis, waveform analysis and reconstruction of Raman backscattered signals. The preprocessing architecture for autocorrelation characteristic analysis, waveform analysis and reconstruction in this scheme can effectively suppress the transient effects introduced by EDFA, thereby significantly improving the amplitude fidelity and temporal stability of the encoded pulse sequence. This innovative active reconstruction mechanism in the signal domain, which is based on analyzing the waveforms of Raman backscattered signals via forward/inverse Fourier transforms, directly eliminates the phenomenon of detection pulse distortion existing in traditional demodulation schemes. The scope of action of the EAC-coding scheme encompasses two key aspects: specifically, one is addressing the fundamental limitation of detection signal distortion through physical-layer waveform reconstruction, and the other is performing correlation operations between the proposed coding sequence and the reconstructed Raman signal to restore the ideal cross-correlation characteristics of Golay sequences, and achieve a theoretical gain value,

thereby breaking through the SNR-distance trade-off limit. It achieves the coordinated optimization of “waveform correction, correlation characteristic restoration, signal intensity enhancement”, laying a physical layer foundation for long-distance sensing.

Furthermore, the proposed EAC-coding scheme features a synergistically enhanced two-stage noise suppression mechanism, demonstrating innovations in multi-level joint denoising.

First, EAC-coding scheme enhances the SNR of sensing signals through its inherent coding gain. Based on the cross-correlation characteristics of its sequence design, this coding scheme provides a significant inherent gain, which compensates for signal attenuation caused by long-distance transmission and effectively improves the system's baseline SNR. Furthermore, this mechanism not only significantly boosts optical power but also leverages the irrelevance between the decoding algorithm and incoherent noise to achieve preliminary suppression of transmission link losses at the encoding-decoding level.

Second, aiming at the residual noise with specific time-frequency domain distribution characteristics in the decoded signal, this scheme further introduces Haar wavelet transform for adaptive noise filtering. The multi-resolution analysis capability of the Haar wavelet enables accurate capture of the local time-frequency features of noise. By performing sparse representation of the signal at different scales (corresponding to different frequency bands) and time-domain positions, it achieves effective separation between useful signals and non-stationary noise components. While preserving the key details of the distributed temperature signal, this method efficiently eliminates residual noise, significantly enhancing the clarity of the signal in the time-frequency domain and the overall measurement accuracy of the system.

This three-fold synergistic mechanism, consisting of the preprocessing framework compensating for EDFA fluctuation, coding gain suppressing transmission losses, and wavelet transform finely removing residual noise, breaks through the theoretical trade-off bottleneck between SNR and sensing distance in traditional schemes. This scheme provides a novel decoding framework for long-distance and high-precision Raman distributed optical fiber sensing. Experiment results indicate that, at an ultra-long distance of 70.0 km, this scheme can achieve a spatial resolution of 1.58 m, a temperature resolution of 5.39 °C, and an effective number of sensing points

reaching 44,303. To the best of our knowledge, the effective number of sensing points of EAC-coding scheme currently represents the world's highest level in the field of Raman distributed optical fiber sensing.

(2) Based on the reviewer's professional suggestions, we have conducted a more comprehensive investigation and comparative analysis of the technologies in this field. We have carried out a detailed comparison between the EAC-coding scheme proposed in the manuscript and the scheme described in the reference for comparison from four dimensions: preprocessing architecture, coding gain, decoding steps, and system complexity.

① **In terms of the preprocessing architecture**, the proposed EAC-coding scheme is fundamentally different from the Derived sequences coding scheme in the references for comparison (which corrects the decoding sequence through attenuation envelope modeling). The Derived sequences coding scheme constructs decoded derived sequences that maintain good cross-correlation with distorted detection signals through steps including extracting the attenuation envelope, establishing a compensation model, and solving the compensation envelope via matrix operations (as shown in Eq. (8) - (14) of the original paper). This technology modifies the coding sequence (i.e., the derived sequence), and the modified sequence maintains good correlation characteristics with the detection signals. However, the detection signals remain in a distorted state essentially, and the reduction in the intensity of Raman backscattered signals caused by the distorted detection signal is not resolved. EAC-coding scheme proposed in this paper innovatively constructs an active reconstruction mechanism in the signal domain. Based on the forward/inverse Fourier transform to analyze the waveform of Raman backscattered signals (Eq. (5) - (10) of the manuscript), it directly eliminates the detection pulse distortion caused by the transient effect of EDFA. This method not only restores the ideal cross-correlation characteristics between the Golay complementary sequence and the detection signals but also achieves the coordinated optimization of “waveform correction, correlation characteristic restoration, signal intensity enhancement”, laying a physical layer foundation for long-distance sensing.

② **In terms of the coding gain**, the scheme proposed in the reference for comparison performs encoding based on Golay complementary sequences and conducts correlation operations for decoding using derived sequences generated from the Golay complementary sequences. However, the detection

signals remain in a distorted state (as stated in the first paragraph of the left column on Page 6 of the original paper). The EAC-coding scheme in this manuscript addresses the performance degradation issue of the reference caused by detection signals distortion through physical-layer waveform reconstruction. The reference for comparison only maintains partial cross-correlation through sequence modification, resulting in an actual gain lower than the theoretical limit of $\sqrt{L}/2$. In contrast, EAC-coding scheme performs Fourier-domain analytical reconstruction on Raman backscattered signals to directly eliminate transient distortion, restore the ideal cross-correlation characteristics of Golay sequences, and achieve a theoretical gain value. Furthermore, this paper integrates Haar wavelet denoising to suppress time/frequency domain coupled noise, synergistically improving the SNR by 8.72 dB. Ultimately, this breaks through the SNR bottleneck for long-distance sensing up to 70.0 km, while the reference for comparison only achieves a sensing distance of 44.0 km.

③ **In terms of the decoding principle**, the breakthrough of EAC-coding lies in addressing the fundamental limitation of detection signal distortion in the reference literature through physical-layer waveform reconstruction. The reference for comparison adopts derived sequences to perform correlation decoding with distorted detection signals (as stated in the first paragraph of the left column on Page 6 of the original paper), but fails to compensate for the Raman backscattered signals attenuation caused by signal distortion. This results in limited Raman backscattered signals intensity and a sensing distance capped at 44.0 km. In contrast, the proposed scheme performs correlation operations between the Golay sequence and the reconstructed Raman signal (with transient distortion eliminated), thereby breaking through the SNR-distance trade-off limit. Ultimately, it achieves ultra-long-distance sensing of 70.0 km, representing a 59% sensing distance extension compared to the scheme in the reference for comparison.

④ **In terms of the system complexity**, EAC-coding achieves an order-of-magnitude simplification through innovations in the algorithm architecture. The reference for comparison requires implementing a sequence of steps (including attenuation envelope extraction, nonlinear compensation modeling, and matrix inversion computation) to construct derived sequences, which significantly increases the difficulty of real-time demodulation. In contrast, the proposed scheme only needs a single forward/inverse Fourier transform to reconstruct the Raman backscattered signal. While ensuring

performance, it reduces the system complexity to a level that facilitates engineering deployment.

(3) Furthermore, we have systematically compared the technical characteristics of the EAC-coding scheme proposed in this manuscript with those of the current mainstream pulse coding schemes.

Currently, the performance comparison between conventional pulse coding schemes and the EAC-coding scheme proposed in this manuscript is presented in Table 4.

① **In terms of coding gain**, the EAC-coding scheme proposed in this manuscript exhibits significant advantages in SNR improvement. The core of this scheme lies in the analysis and accurate reconstruction of the waveform of Raman backscattered signals, which effectively compensates for the detection signals distortion caused by the transient effects of the system. This mechanism enables EAC-coding to overcome the performance bottleneck of traditional derived sequences coding schemes induced by signals distortion, achieving a coding gain closer to the theoretical limit. Specifically, existing derived sequence coding scheme construct coding sequences based on Golay complementary sequences and perform correlation decoding using their derived sequences. The upper limit of their theoretical coding gain is $\sqrt{L}/2$ (where L denotes the length of the coding sequence)^[1]. However, due to the unavoidable distortion (especially transient distortion) in actual detection signals, the results of their correlation operations deteriorate, leading to a significant gap between the actually achieved coding gain and this theoretical value. In contrast, the EAC-coding scheme, through the aforementioned signals analysis and reconstruction mechanism, significantly suppresses the impact of transient distortion on the detection signals waveform, thereby enabling its coding gain to approach the theoretical upper limit of $\sqrt{L}/2$. Benefiting from this, the proposed system ultimately achieves temperature sensing capability in a long-distance Raman distributed optical fiber sensing system with a sensing distance of 70.0 km.

② **In terms of measurement time**, the core advantage of the EAC-coding scheme stems from its extremely low decoding computational complexity. This scheme achieves signals reconstruction solely through Fourier transform, which significantly reduces the computational cost and time delay of real-time data processing, thereby effectively improving the overall response speed of the system. Specifically, for a Simplex code with length L ^[2], it is necessary to inject L groups of detection signals into the

optical fiber and collect L groups of Raman backscattered signals, with its theoretical minimum acquisition time being $T_{Simplex}=L \times t$ (where t denotes the acquisition time of one group of Raman backscattered curves). Although the Genetic-optimised aperiodic coding scheme [3] only requires injecting one group of detection signals into the optical fiber, it needs additional time to design the coding sequence based on the distributed genetic algorithm. The Derived sequences coding scheme requires injecting 4 groups of detection signals into the optical fiber, with a theoretical acquisition time of $4t$. However, in the decoding stage, it involves complex steps such as extracting the attenuation envelope, establishing a compensation model, and solving the compensation envelope via matrix operations, which increases the time cost of the decoding process. The EAC-coding scheme proposed in this manuscript performs encoding based on Golay complementary sequences, with a theoretical acquisition time of $4t$, and only needs Fourier transform to complete the analysis and reconstruction of the Raman backscattered signals waveform. This significant efficiency improvement at the algorithm level greatly reduces the real-time data processing delay and effectively enhances the system response speed, providing crucial real-time guarantee for the final realization of temperature sensing in a 70.0 km long-distance Raman distributed optical fiber sensing system.

③ **In terms of system complexity**, the core innovation of the EAC-coding scheme lies in achieving a high degree of simplification in the decoding process through algorithm optimization, while keeping the hardware architecture unchanged. This scheme only needs to perform efficient Fourier transform operations to complete signal reconstruction, which significantly reduces the algorithmic complexity and computational burden of real-time data processing. The specific reasons are as follows. Simplex coding scheme requires the injection of L groups of detection signals, resulting in a long acquisition time and a large volume of raw data. This not only imposes high requirements on hardware performance (such as data acquisition rate and storage capacity) and long-term stability, but also the large-scale data processing involved in its decoding process significantly increases the computational complexity. Genetic-optimised aperiodic coding scheme greatly shortens the acquisition time without increasing hardware costs. However, due to its unique design, it is necessary to design coding sequences based on a distributed genetic algorithm and conduct extensive tests to find the optimal coding sequence. Derived sequences coding scheme

also does not require additional hardware overhead. However, this scheme introduces high computational complexity in the decoding stage, as it relies on computationally expensive matrix operations to generate and process derived sequences.

EAC-coding scheme proposed in this manuscript incorporates a novel preprocessing framework for autocorrelation characteristic analysis, waveform analysis and reconstruction of Raman backscattered signals. This framework significantly improves the fidelity and transmission stability of the encoded pulse sequence by dynamically compensating for the transient gain fluctuation of EDFA. This technology leverages the inherent gain of EAC-coding pulse coding to effectively offset fiber scattering attenuation during 70.0 km long-distance transmission, achieving a significant improvement in the baseline SNR. On this basis, a two-stage noise suppression method is further constructed. The first-stage processing involves compensating for signal attenuation using the theoretical coding gain of EAC-coding. For the second-stage processing, a Haar wavelet denoising algorithm is adopted. Its step-matching characteristic enables accurate extraction of the abrupt temperature change features of the sensing fiber and adaptive filtering of the residual time/frequency domain coupled noise in the decoded signal. This three-fold synergistic mechanism, consisting of the preprocessing framework compensating for EDFA fluctuation, coding gain suppressing transmission losses, and wavelet transform finely removing residual noise, breaks through the theoretical trade-off bottleneck between SNR and sensing distance in traditional schemes. Experiment results indicate that, at an ultra-long distance of 70.0 km, this scheme can achieve a spatial resolution of 1.58 m, a temperature resolution of 5.39 °C, and an effective number of sensing points reaching 44,303, verifying the engineering feasibility of the scheme.

Table 4. Sensing performance of various coding schemes.

Scheme	Sensing Distance	Spatial Resolution	Sensing Points	Temperature Resolution	Measurement Time
Simplex coding ^[2]	37.0 km	17.0 m	2176	3.0 °C	/
Low-repetition-rate cyclic pulse coding ^[4]	26.0 km	1.0 m	26000	3.0 °C	30 s
Pre-Shaped Simplex Coding ^[5]	62.0 km	10.0 m	6200	8.4 °C	/
Genetic-optimised aperiodic coding ^[3]	39.0 km	1.0 m	39000	3.9 °C	13.6 min
Derived Sequences coding ^[1]	44.0 km	5.0 m	8800	2.5 °C	/
EAC-coding	70.0 km	1.58 m	44303	5.39 °C	252 min

Sensing Points refers to the number of independent spatial sampling points in the fiber optic sensing link where the system can achieve effective measurements. The specific calculation method is the ratio of the sensing distance to the spatial resolution.
Temperature resolution is defined as the minimum temperature change that a system can reliably distinguish, and its calculation method is based on the standard deviation under steady-state conditions.

[1] D. D. Chai, *et al.* Derived sequences decoding approach for long-range distributed temperature sensors. *IEEE Sens. J.* **23**(3): 2204-2210 (2022).

[2] J. Park, *et al.* Raman-based distributed temperature sensor with simplex coding and link optimization. *IEEE Photonics Technol. Lett.* **18**(17): 1879-1881 (2006).

[3] X. Z. Sun, *et al.* Genetic-optimised aperiodic code for distributed optical fibre sensors. *Nat. Commun.* **11**(1), 5774 (2020).

[4] M. A. Soto, *et al.* Raman-based distributed temperature sensor with 1 m spatial resolution over 26 km SMF using low-repetition-rate cyclic pulse coding. *Opt. Lett.* **36**(13): 2557-2559 (2011).

[5] J. B. Rosolem, *et al.* Raman DTS based on OTDR improved by using gain-controlled EDFA and pre-shaped simplex code. *IEEE Sens. J.* **17**(11): 3346-3353 (2017).

Thank you for your suggestion. We have made the following modifications in our revised manuscript:

- 1) In the **(unmodified manuscript)** section of **Experimental results**, **page 10, right column, paragraph 2, line 29**, delete “Raman distributed optical fiber sensing has emerged as a pivotal technology for continuous temperature monitoring in long-range infrastructure and environmental applications. However, its practical implementation over distances exceeding 30 km has been fundamentally hindered by the inherently low signal-to-noise ratio (SNR) of Raman scattering signals, which creates a critical trade-off between sensing distance and spatial resolution. In this study, we present a novel Raman distributed optical fiber sensing system utilizing the proposed EAC-coding scheme, which effectively suppresses transient effects and significantly enhances the SNR of system while incorporating Haar wavelet. This system enables

long-distance Raman distributed temperature measurement with high spatial resolution, achieving precise temperature demodulation and accurate identification of FUT position and length. The experimental results demonstrate that the proposed scheme achieves a spatial resolution of 1.58 m, temperature resolution of 5.39 °C, and temperature measurement accuracy of 0.88 °C at the sensing distance of 70 km, establishing a new approach for long-distance Raman distributed optical fiber sensing system. Future research will integrate more advanced denoising techniques to extend the system's sensing distance while preserving its spatial resolution capabilities.”

- 2) In the section of **Experimental results, page 18, left column, paragraph 2, line 10**, add “EAC-coding scheme proposed in this paper incorporates a novel preprocessing framework for autocorrelation characteristic analysis, waveform analysis and reconstruction of Raman backscattered signals. The preprocessing architecture for autocorrelation characteristic analysis, waveform analysis and reconstruction in this scheme can effectively suppress the transient effects introduced by EDFA, thereby significantly improving the amplitude fidelity and temporal stability of the encoded pulse sequence. This innovative active reconstruction mechanism in the signal domain, which is based on analyzing the waveforms of Raman backscattered signals via forward/inverse Fourier transforms, directly eliminates the phenomenon of detection pulse distortion existing in traditional demodulation schemes. The scope of action of the EAC-coding scheme encompasses two key aspects: specifically, one is addressing the fundamental limitation of detection signal distortion through physical-layer waveform reconstruction, and the other is performing correlation operations between the proposed coding sequence and the reconstructed Raman signal to restore the ideal cross-correlation characteristics of Golay sequences, and achieve a theoretical gain value, thereby breaking through the SNR-distance trade-off limit. It achieves the coordinated optimization of “waveform correction, correlation characteristic restoration, signal intensity enhancement”, laying a physical layer foundation for long-distance sensing.

Furthermore, the proposed EAC-coding scheme features a synergistically enhanced two-stage noise suppression mechanism,

demonstrating innovations in multi-level joint denoising.

First, EAC-coding scheme enhances the SNR of sensing signals through its inherent coding gain. Based on the cross-correlation characteristics of its sequence design, this coding scheme provides a significant inherent gain, which compensates for signal att

enuation caused by long-distance transmission and effectively improves the system's baseline SNR. This mechanism not only significantly boosts optical power but also leverages the irrelevance between the decoding algorithm and incoherent noise to achieve preliminary suppression of transmission link losses at the encoding-decoding level.

Second, aiming at the residual noise with specific time-frequency domain distribution characteristics in the decoded signal, this scheme further introduces Haar wavelet transform for adaptive noise filtering. The multi-resolution analysis capability of the Haar wavelet enables accurate capture of the local time-frequency features of noise. By performing sparse representation of the signal at different scales (corresponding to different frequency bands) and time-domain positions, it achieves effective separation between useful signals and non-stationary noise components. While preserving the key details of the distributed temperature signal, this method efficiently eliminates residual noise, significantly enhancing the clarity of the signal in the time-frequency domain and the overall measurement accuracy of the system.

This three-fold synergistic mechanism, consisting of the preprocessing framework compensating for EDFA fluctuation, coding gain suppressing transmission losses, and wavelet transform finely removing residual noise, breaks through the theoretical trade-off bottleneck between SNR and sensing distance in traditional schemes. This scheme provides a novel decoding framework for long-distance and high-precision Raman distributed optical fiber sensing. Experiment results indicate that, at an ultra-long distance of 70.0 km, this scheme can achieve a spatial resolution of 1.58 m, a temperature resolution of 5.39 °C, and an effective number of sensing points reaching 44,303. To the best of our knowledge, the effective number of sensing points of EAC-coding scheme currently

represents the world's highest level in the field of Raman distributed optical fiber sensing.”

- 3) In the **(unmodified manuscript)** section of **Introduction, page 3, left column, paragraph 2, line 3, delete** “ In this work, an enhanced autocorrelation pulse coding (EAC-coding) scheme is proposed, which introduces Golay complementary sequences into Raman distributed optical fiber sensing, and at the same time restores the autocorrelation which is destroyed by the transient effects of erbium-doped fiber amplifier (EDFA) by preprocessing the Raman backscattered signals for analysis and waveform-reconstruction. The proposed scheme further incorporates Haar wavelet denoising algorithm to enhance system performance. In the experiment, a Raman distributed optical fiber sensing with sensing distance of 70.0 km, spatial resolution of 1.58 m and temperature resolution of 5.39 °C is achieved, which provides a new solution for the research of long-distance Raman distributed optical fiber sensing.”
- 4) In the section of **Introduction, page 4, left column, paragraph 4, line 35, add** “To address the aforementioned issues, an Enhanced anti-distortion coding (EAC-coding) scheme is proposed in this paper. This scheme effectively overcomes the trade-off dilemma among SNR, sensing distance, and hardware cost in traditional methods through a three-fold synergistic mechanism: compensating for EDFA fluctuation via a preprocessing framework, suppressing transmission losses through coding gain, and finely removing residual noise using wavelet transform. While maintaining relatively low complexity, the EAC-coding scheme significantly enhances the anti-distortion capability and adaptability of the system, providing a new solution for long-distance, high-performance Raman distributed optical fiber sensing. Experimental results demonstrate that at an ultra-long distance of 70.0 km, the system achieves a spatial resolution of 1.58 m and a temperature resolution of 5.39 °C, with the number of effective sensing points reaching 44,303. To the best of our knowledge, this number of effective sensing points represents the highest level in the field of Raman distributed optical fiber sensing. In addition, the preprocessing architecture within EAC-coding (autocorrelation characteristic analysis, waveform

analysis and reconstruction of Raman backscattered signals) significantly enhances the temporal fidelity and transmission stability of encoded sequences by dynamically compensating for EDFA transient gain fluctuation. This further improves the SNR and temperature measurement accuracy performance, enabling the temperature measurement accuracy to reach 0.91 °C, verifying the engineering feasibility of the scheme.”

- 5) In the section of **Discussion, page 19, right column, paragraph 4, line 7**, add “Currently, the pulse coding techniques applied in Raman distributed optical fiber sensing system mainly include the following: Simplex coding scheme, Pre-Shaped Simplex coding scheme, Low-repetition-rate cyclic pulse coding scheme, Genetic-optimised aperiodic coding scheme, and Derived Sequences coding scheme. Their key performance indicators are presented in Table 2.

Table 2. Sensing performance of various coding schemes.

Scheme	Sensing Distance	Spatial Resolution	Sensing Points	Temperature Resolution	Measurement Time
Simplex coding ^[31]	37.0 km	17.0 m	2176	3.0 °C	/
Low-repetition-rate cyclic pulse coding ^[27]	26.0 km	1.0 m	26000	3.0 °C	30 s
Pre-Shaped Simplex Coding ^[32]	62.0 km	10.0 m	6200	8.4 °C	/
Genetic-optimised aperiodic coding ^[29]	39.0 km	1.0 m	39000	3.9 °C	13.6 min
Derived Sequences coding ^[33]	44.0 km	5.0 m	8800	2.5 °C	/
EAC-coding	70.0 km	1.58 m	44303	5.39 °C	252 min

Sensing Points refers to the number of independent spatial sampling points in the fiber optic sensing link where the system can achieve effective measurements. The specific calculation method is the ratio of the sensing distance to the spatial resolution.
Temperature resolution is defined as the minimum temperature change that a system can reliably distinguish, and its calculation method is based on the standard deviation under steady-state conditions.

We have compared our proposed scheme with other coding methods such as simplex coding and Genetic-optimised aperiodic coding in terms of coding gain, measurement time, and system complexity. The specific revisions are as follows.

In terms of coding gain, the EAC-coding scheme proposed in this paper exhibits significant advantages in SNR improvement. The core of this scheme lies in the analysis and accurate reconstruction of the waveform of Raman backscattered signals, which effectively compensates for the detection signals distortion caused by the transient effects of the system. This mechanism enables EAC-coding to overcome the performance bottleneck of traditional derived

sequences coding schemes induced by signals distortion, achieving a coding gain closer to the theoretical limit. Specifically, existing derived sequence coding scheme construct coding sequences based on Golay complementary sequences and perform correlation decoding using their derived sequences. The upper limit of their theoretical coding gain is $\sqrt{L}/2$ (where L denotes the length of the coding sequence). However, due to the unavoidable distortion (especially transient distortion) in actual detection signals, the results of their correlation operations deteriorate, leading to a significant gap between the actually achieved coding gain and this theoretical value. In contrast, the EAC-coding scheme, through the aforementioned signals analysis and reconstruction mechanism, significantly suppresses the impact of transient distortion on the detection signals waveform, thereby enabling its coding gain to approach the theoretical upper limit of $\sqrt{L}/2$. Benefiting from this, the proposed system ultimately achieves temperature sensing capability in a long-distance Raman distributed optical fiber sensing system with a sensing distance of 70.0 km.

In terms of measurement time, the core advantage of the EAC-coding scheme stems from its extremely low decoding computational complexity. This scheme achieves signals reconstruction solely through Fourier transform, which significantly reduces the computational cost and time delay of real-time data processing, thereby effectively improving the overall response speed of the system. Specifically, for a Simplex code with length L , it is necessary to inject L groups of detection signals into the optical fiber and collect L groups of Raman backscattered signals, with its theoretical minimum acquisition time being $T_{Simplex} = L \times t$ (where t denotes the acquisition time of one group of Raman backscattered curves). Although the Genetic-optimised aperiodic coding scheme only requires injecting one group of detection signals into the optical fiber, it needs additional time to design the coding sequence based on the distributed genetic algorithm. The Derived sequences coding scheme requires injecting 4 groups of detection signals into the optical fiber, with a theoretical acquisition time of $4t$. However, in the decoding stage, it involves complex steps such as extracting the attenuation envelope, establishing a compensation model, and

solving the compensation envelope via matrix operations, which increases the time cost of the decoding process. The EAC-coding scheme proposed in this paper performs encoding based on Golay complementary sequences, with a theoretical acquisition time of $4t$, and only needs Fourier transform to complete the analysis and reconstruction of the Raman backscattered signals waveform. This significant efficiency improvement at the algorithm level greatly reduces the real-time data processing delay and effectively enhances the system response speed, providing crucial real-time guarantee for the final realization of temperature sensing in a 70.0 km long-distance Raman distributed optical fiber sensing system.

In terms of system complexity, the core innovation of the EAC-coding scheme lies in achieving a high degree of simplification in the decoding process through algorithm optimization, while keeping the hardware architecture unchanged. This scheme only needs to perform efficient Fourier transform operations to complete signal reconstruction, which significantly reduces the algorithmic complexity and computational burden of real-time data processing. The specific reasons are as follows. Simplex coding scheme requires the injection of L groups of detection signals, resulting in a long acquisition time and a large volume of raw data. This not only imposes high requirements on hardware performance (such as data acquisition rate and storage capacity) and long-term stability, but also the large-scale data processing involved in its decoding process significantly increases the computational complexity. Genetic-optimised aperiodic coding scheme greatly shortens the acquisition time without increasing hardware costs. However, due to its unique design, it is necessary to design coding sequences based on a distributed genetic algorithm and conduct extensive tests to find the optimal coding sequence. Derived sequences coding scheme also does not require additional hardware overhead. However, this scheme introduces high computational complexity in the decoding stage, as it relies on computationally expensive matrix operations to generate and process derived sequences.

EAC-coding scheme proposed in this paper incorporates a novel preprocessing framework for autocorrelation characteristic

analysis, waveform analysis and reconstruction of Raman backscattered signals. This framework significantly improves the fidelity and transmission stability of the encoded pulse sequence by dynamically compensating for the transient gain fluctuation of EDFA. This technology leverages the inherent gain of EAC-coding pulse coding to effectively offset fiber scattering attenuation during 70.0 km long-distance transmission, achieving a significant improvement in the baseline SNR. On this basis, a two-stage noise suppression method is further constructed. The first-stage processing involves compensating for signal attenuation using the theoretical coding gain of EAC-coding. For the second-stage processing, a Haar wavelet denoising algorithm is adopted. Its step-matching characteristic enables accurate extraction of the abrupt temperature change features of the sensing fiber and adaptive filtering of the residual time/frequency domain coupled noise in the decoded signal. This three-fold synergistic mechanism, consisting of the preprocessing framework compensating for EDFA fluctuation, coding gain suppressing transmission losses, and wavelet transform finely removing residual noise, breaks through the theoretical trade-off bottleneck between SNR and sensing distance in traditional schemes. Experiment results indicate that, at an ultra-long distance of 70.0 km, this scheme can achieve a spatial resolution of 1.58 m, a temperature resolution of 5.39 °C, and an effective number of sensing points reaching 44,303, verifying the engineering feasibility of the scheme. To the best of our knowledge, the number of effective sensing points achieved by this scheme ranks the highest in the field of Raman distributed optical fiber sensing systems.

2. The authors explain the process of reconstructing the Raman backscattered signals to eliminate transient effects in Sec. 2, but it would be better to demonstrate experimental proof of their approach. The results shown in Fig. 4 are alluding to this, but not very convincing.

Reply: We thank the reviewer for their valuable comments. We fully concur with their assessment and have revised the manuscript to incorporate all of their suggestions. Based on the reviewer's professional suggestions, we have added an autocorrelation processing experiment on the detection signals

coupled into the sensing fiber after reconstruction, aiming to further verify that it eliminates the transient effects of the system. The details of the newly added experiment are as follows.

The primary objective of the newly added experiment is to verify that the reconstructed Raman backscattered signals is equivalent to the Raman backscattered signal of the undistorted detection signals.

Taking the detection signal u as an example, its attenuation envelope function can be expressed by Eq. (1).

$$f' = \text{iffi} \left[\frac{\text{fft}(u \cdot f)}{U} \right] \quad (1)$$

Based on Eq. (1), the detection signal coupled into the sensing fiber after reconstruction can be calculated and is given by Eq. (2).

$$u' = \text{iffi} \left(\frac{\text{fft}[u(f)]}{\text{fft}(f')} \right) \quad (2)$$

Then, we performed the same processing on the four detection signals and conducted autocorrelation operations on the processed detection signals. If Eq. (3) is satisfied, it can be verified that their autocorrelation characteristics are restored after reconstruction, with the transient effects effectively suppressed.

$$(u' - \bar{u}') * (u' - \bar{u}') + (w' - \bar{w}') * (w' - \bar{w}') = 2l\delta \quad (3)$$

The experimental results are shown in Fig. 3. Fig. 3 (c1) presents the autocorrelation function of the detection signals with waveform distortion caused by transient effects, while Fig. 3(c2) shows the autocorrelation function of the equivalent detection signals coupled into the sensing fiber after reconstruction. It can be clearly observed from Fig. 3(c2) that the sidelobe noise of its autocorrelation function is significantly reduced, which verifies that its autocorrelation characteristics are restored and the transient effects are effectively suppressed.

Fig.3. (c1) Autocorrelation function of the detection signals before reconstruction. (c2) Autocorrelation function of the detection signals after reconstruction.

Thank you for your suggestion. We have made the following modifications in our revised manuscript:

- 1) In the section of Experimental results, page 11, left column, change “Fig .4” to “Fig. 4”.

Fig. 4. Intensity of Raman backscattered signals and SNR. (a1) Intensity of

Raman backscattered signal generated by detection signal u before and after pre-processing. (a2) Intensity of Raman backscattered signal generated by detection signal \bar{u} before and after pre-processing. (a3) Intensity of Raman backscattered signal generated by detection signal w before and after pre-processing. (a4) Intensity of Raman backscattered signal generated by detection signal \bar{w} before and after pre-processing. (b1) Intensity of Raman backscattered signals decoded by distorted detection signals before and after pre-processing. (b2) SNR of Raman backscattered signals decoded by detection signals before and after pre-processing. (c1) Autocorrelation function of the detection signals before pre-processing. (c2) Autocorrelation function of the distorted detection signals after pre-processing.

2) In the section of **Experimental results, page 12, left column, paragraph 2, line 3**, add “We have added an autocorrelation processing experiment on the detection signals coupled into the sensing fiber after reconstruction, aiming to further verify that it eliminates the transient effects of the system. The details of the newly added experiment are as follows. The primary objective of the newly added experiment is to verify that the reconstructed Raman backscattered signals is equivalent to the Raman backscattered signal of the undistorted detection signals.

Taking the detection signal u as an example, its attenuation envelope function can be expressed by Eq. (8). Thus, the detection signal coupled into the sensing fiber after reconstruction can be calculated and is given by Eq. (17).

$$u' = \text{ifft}\left(\frac{\text{fft}[u(f)]}{\text{fft}(f')}\right) \quad (17)$$

Then, we performed the same processing on the four detection signals and conducted autocorrelation operations on the processed detection signals. If Eq. (18) is satisfied, it can be verified that their autocorrelation characteristics are restored after reconstruction, with the transient effects effectively suppressed.

$$(u' - \bar{u}') * (u' - \bar{u}') + (w' - \bar{w}') * (w' - \bar{w}') = 2l\delta \quad (18)$$

The experimental results are shown in Fig. 4. Fig. 4 (c1) presents the autocorrelation function of the detection signals with waveform distortion caused by transient effects, while Fig. 4(c2)

shows the autocorrelation function of the equivalent detection signals coupled into the sensing fiber after reconstruction. It can be clearly observed from Fig. 4(c2) that the sidelobe noise of its autocorrelation function is significantly reduced, which verifies that its autocorrelation characteristics are restored and the transient effects are effectively suppressed.”

3. The improvement claimed in Fig. 4 is with respect to RBS decoded using “destroyed detection” signals, but the comparison presented for the Raman OTDR (Figs. 5-7) is with respect to single pulse Raman OTDR. Why is that? How about comparing with the uncompensated RBS?

Reply: We would like to express our sincere gratitude for the reviewer’s professional comments pointed out this issue. Based on the reviewer’s professional suggestions, we have added correlation demodulation experiments corresponding to Fig. 5-7, which compare three Raman distributed optical fiber sensing temperature schemes: the traditional demodulation scheme, the Raman backscattered sensing scheme before compensation, and the Raman backscattered sensing scheme after compensation. In this study, the improvements in SNR (accompanied by performance enhancements in sensing distance and temperature resolution) are primarily attributed to the proposed EAC-coding (for coding gain compensation) and Haar wavelet denoising schemes. In contrast, the preprocessing architecture within EAC-coding (characterized by autocorrelation characteristic analysis, waveform analysis and reconstruction of Raman backscattered signals) serves to improve the system's temperature measurement accuracy by dynamically compensating for EDFA transient gain fluctuation and reconstructing Raman scattered signals. The detailed supplementary experiment content is as follow.

In this manuscript, through the synergistic demodulation mechanism of EAC-coding (which utilizes coding gain to compensate for attenuation) combined with the refined processing of Haar wavelet denoising, the theoretical trade-off bottleneck between SNR and sensing distance in traditional schemes is broken through. Experimental results demonstrate that at an ultra-long distance of 70.0 km, the system achieves a spatial resolution of 1.58 m and a temperature resolution of 5.39 °C. Moreover, the preprocessing architecture in EAC-coding (integrating autocorrelation

characteristic analysis of Raman backscattered signals and waveform analysis and reconstruction) significantly enhances the temporal fidelity and transmission stability of coded pulse sequences by dynamically compensating for EDFA transient gain fluctuation. This further improves the system's SNR and temperature measurement accuracy, enabling the temperature measurement accuracy to reach 0.91 °C.

The results of temperature demodulation and SNR are presented in Fig. 4. EAC-coding, by integrating the coded pulse gain with the preprocessing architecture (which incorporates autocorrelation characteristic analysis of Raman backscattered signals and waveform analysis and reconstruction), effective suppression of temperature fluctuation noise along the optical fiber is achieved. This optimizes the temperature fluctuation range from 300 °C (in the traditional single-pulse scheme) to 20.86 °C, as illustrated in Fig. 4(a1) - (a3). The SNR measurement results are shown in Fig. 4(b1) - (b3). Compared with the traditional single-pulse scheme, the SNR of the Raman backscattered signal after decoding based on EAC-coding is improved by 8.72 dB. This verifies that the EAC-coding scheme proposed in this manuscript effectively enhances the SNR and the sensing distance through coding gain. Furthermore, we conducted additional quantitative analysis by evaluating the system's temperature resolution. Temperature resolution refers to the minimum distinguishable temperature range of the system. It is one of the core parameters for evaluating the performance of Raman distributed optical fiber sensing systems. The temperature resolution of the conventional single-pulse scheme is shown in Fig. 4(c1). At a sensing distance of 70.0 km, the traditional single-pulse scheme achieves a temperature resolution of 29.25 °C. In contrast, the EAC-coding scheme proposed in this paper optimizes the temperature resolution to 5.39 °C at the same sensing distance.

Meanwhile, we further compared the temperature measurement accuracy performance of the system, as shown in Fig. 4(d1) - (d3). It can be observed that the transient effects not only affect the SNR, temperature fluctuation, and temperature resolution but also exert a significant impact on the system's temperature measurement accuracy. From Fig. 6(d1) - (d3), it can be observed that as the temperature of the fiber under test (FUT) gradually increases, the temperature measurement error of the distorted signal demodulation results becomes increasingly large. This confirms that

the influence of the transient effect also intensifies with rising temperature—with the temperature measurement accuracy deteriorating to a maximum of 14.98 °C at 90 °C (where the demodulation result is 104.98 °C). This is because with increasing temperature, the elevation of the Raman backscattered curve at the FUT becomes more pronounced, while transient effects exert an increasingly significant influence on signal intensity. After the preprocessing architecture in the EAC-coding scheme (autocorrelation characteristic analysis, waveform analysis and reconstruction of Raman backscattered signals), the temperature demodulation result is 90.91 °C, and the temperature measurement accuracy is 0.91 °C, representing an improvement of approximately 16.46 times. Therefore, the EAC-coding scheme proposed in this manuscript can effectively improve the system's temperature measurement accuracy, enhancing the temperature measurement accuracy of the long-distance Raman distributed optical fiber sensing system (with a sensing distance of 70.0 km) to within 1.0 °C.

Fig.4. Experimental comparison results of temperature sensing. (a1) Temperature demodulation results at room temperature based on single-pulse scheme. (a2) Temperature demodulation results at room temperature based on distorted detection signals. (a3) Temperature demodulation results at room temperature based on proposed EAC-coding scheme. (b1) SNR based on single-pulse scheme. (b2) SNR based on EAC-coding scheme (without pre-processing). (b3) SNR based on proposed EAC-coding scheme. (c1) Temperature resolution based on single-pulse scheme. (c2) Temperature resolution based on EAC-coding scheme (without pre-processing). (c3) Temperature resolution based on proposed EAC-coding scheme. (d1) Temperature demodulation results at 50 °C based on EAC-coding scheme and EAC-coding scheme (without pre-processing). (d2) Temperature demodulation results at 70 °C based on EAC-coding scheme and EAC-coding scheme (without pre-processing). (d3) Temperature demodulation results at 90 °C based on EAC-coding scheme and EAC-coding scheme (without pre-processing).

Thank you for your suggestion. We have made the following modifications in our revised manuscript:

- 1) In the section of **Experimental results**, page 14, change “Fig. 6, Fig. 7” to “Fig. 6”.

Fig.6. Experimental comparison results of temperature sensing. (a1) Temperature demodulation results at room temperature based on single-pulse scheme. (a2) Temperature demodulation results at room temperature based on distorted detection signals. (a3) Temperature demodulation results at room temperature based on proposed EAC-coding scheme. (b1) SNR based on single-pulse scheme. (b2) SNR based on EAC-coding scheme (without pre-processing). (b3) SNR based on proposed EAC-coding scheme. (c1) Temperature resolution based on single-pulse scheme. (c2) Temperature resolution based on EAC-coding scheme (without pre-processing). (c3) Temperature resolution based on proposed EAC-coding scheme. (d1) Temperature demodulation results at 50 °C based on EAC-coding scheme and EAC-coding scheme (without pre-processing). (d2) Temperature demodulation results at 70 °C based on EAC-coding scheme and EAC-coding scheme (without pre-processing). (d3) Temperature demodulation results at 90 °C based on EAC-coding scheme and EAC-coding scheme (without pre-processing).

2) In the **(unmodified manuscript)** section of **Experimental results**, **page 9, right column, paragraph 2, line 15, delete** “The SNR measurement result after decoding is shown in Fig. 6(a). Compared with the conventional single-pulse scheme, the SNR at 70.0 km of sensing fiber is improved by 8.75 dB based on proposed EAC-coding scheme, which validates the superiority of the proposed EAC-coding scheme in SNR improvement.

The temperature demodulation result is shown in Fig. 6(b). Experimental measurement demonstrates that the proposed EAC-coding scheme achieves significant suppression of temperature fluctuation under ambient conditions, reducing the temperature fluctuation from ± 300 °C maximum in the conventional single-pulse scheme to ± 15.84 °C.

Furthermore, we conduct additional quantitative analyses by evaluating the temperature resolution of the system.

The temperature resolution based on the conventional single-pulse scheme and proposed EAC-coding scheme are shown in Fig. 7(a). At the sensing distance of 70.0 km, the conventional single-pulse scheme achieves a temperature resolution of 29.25 °C. In contrast, the proposed EAC-coding scheme demonstrates significant performance enhancement by effectively improving the SNR, which substantially reduces system temperature fluctuation, resulting in a remarkable improvement of temperature resolution to 5.39 °C at the same sensing distance. Meanwhile, to validate the spatial resolution of the proposed EAC-coding scheme, a

40-meter-long fiber under test is positioned at the distance of 70.0 km and immersed in a thermostatically controlled water bath maintained at 90 °C, while the remaining portion of the sensing fiber is exposed to the room temperature of 20 °C”.

- 3) In the section of **Experimental results**, page 15, left column, **paragraph 2, line 5**, add “The results of temperature demodulation and SNR are presented in Fig. 6. The temperature fluctuation of the traditional single-pulse scheme can be as high as approximately 300 °C, as shown in Fig. 6(a1). In this paper, first, by leveraging the synergistic demodulation mechanism integrating EAC-coding (which compensates for attenuation via coding gain) and the refined processing of Haar wavelet denoising, effective suppression of temperature fluctuation noise along the optical fiber is achieved. This significantly reduces the system’s temperature fluctuation to 22.51 °C while improving the SNR by 7.24 dB, as shown in Fig. 6(a2) and Fig. 6(b2). Furthermore, the preprocessing architecture within EAC-coding (autocorrelation characteristic analysis, waveform analysis and reconstruction of Raman backscattered signals) further reduces the system’s temperature fluctuation to 20.86 °C by dynamically compensating for EDFA transient gain fluctuation and reconstructing Raman scattered signals. Meanwhile, the SNR is further improved by an additional 1.48 dB, as shown in Fig. 6(a3) and Fig. 6(b3).

Compared with the temperature resolution of 29.25 °C achieved by the traditional single-pulse scheme, the EAC-coding scheme (integrating coding gain with Haar wavelet denoising) improves the system’s temperature resolution to 5.82 °C, as shown in Fig. 6(c2). Meanwhile, the preprocessing architecture within EAC-coding further enhances the system’s temperature resolution to 5.39 °C by dynamically compensating for EDFA transient gain fluctuation and reconstructing Raman scattered signals, as shown in Fig. 6(c3).

Meanwhile, we further compared the temperature measurement accuracy performance of the system, as shown in Fig. 6(d1) - (d3). It can be observed that the transient effects not only affect the SNR, temperature fluctuation, and temperature resolution but also exert a significant impact on the system’s temperature measurement accuracy. It can be observed that as the temperature of the fiber

under test (FUT) gradually increases, the temperature measurement error of the distorted signal demodulation results becomes increasingly large. This confirms that the influence of the transient effect also intensifies with rising temperature—with the temperature measurement accuracy deteriorating to a maximum of 14.98 °C at 90.0 °C (where the demodulation result is 104.98 °C). This is because with increasing temperature, the elevation of the Raman backscattered curve at the FUT becomes more pronounced, while transient effects exert an increasingly significant influence on signal intensity. After the preprocessing architecture in the EAC-coding scheme (autocorrelation characteristic analysis, waveform analysis and reconstruction of Raman backscattered signals), the temperature demodulation result is 90.91 °C, and the temperature measurement accuracy is 0.91 °C, representing an improvement of approximately 16.46 times. Therefore, the EAC-coding scheme proposed in this paper can effectively improve the system's temperature measurement accuracy, enhancing the temperature measurement accuracy of the long-distance Raman distributed optical fiber sensing system (with a sensing distance of 70.0 km) to within 1.0 °C.

Through the synergistic demodulation mechanism of EAC-coding (which utilizes coding gain to compensate for attenuation) combined with the refined processing of Haar wavelet denoising, the theoretical trade-off bottleneck between SNR and sensing distance in traditional schemes is broken through. Experimental results demonstrate that at an ultra-long distance of 70.0 km, the system achieves a spatial resolution of 1.58 m and a temperature resolution of 5.39 °C. Moreover, the preprocessing architecture in EAC-coding (integrating autocorrelation characteristic analysis of Raman backscattered signals and waveform analysis and reconstruction) significantly enhances the temporal fidelity and transmission stability of coded pulse sequences by dynamically compensating for EDFA transient gain fluctuation. This further improves the system's SNR and temperature measurement accuracy, enabling the temperature measurement accuracy to reach 0.91 °C. In this study, the improvements in SNR (accompanied by performance enhancements in sensing distance and

temperature resolution) are primarily attributed to the proposed EAC-coding (for coding gain compensation) and Haar wavelet denoising schemes. In contrast, the preprocessing architecture within EAC-coding (characterized by autocorrelation characteristic analysis, waveform analysis and reconstruction of Raman backscattered signals) serves to improve the system's temperature measurement accuracy by dynamically compensating for EDFA transient gain fluctuation and reconstructing Raman scattered signals.”

4. If the key contribution of this work is the compensation for transients during the amplification process, then the comparison of results should be between the uncompensated case and the compensated case.

Reply: We would like to express our sincere gratitude for the reviewer's professional comments, and we also apologize for any ambiguous language in our manuscript that may have caused inconvenience to the reviewer. In fact, in this manuscript, the improvements in SNR (accompanied by enhancements in sensing distance and temperature resolution performance) are primarily attributed to the proposed EAC-coding (coding gain) and Haar wavelet denoising schemes. In contrast, the preprocessing architecture of EAC-coding improves the system's temperature measurement accuracy by dynamically compensating for EDFA transient gain fluctuation and reconstructing Raman scattered signals.

Based on the reviewer's professional suggestions, we have added correlation demodulation experiments, which compare three Raman distributed optical fiber sensing temperature schemes: the traditional demodulation scheme, the Raman backscattered sensing scheme before compensation, and the Raman backscattered sensing scheme after compensation. The experimental results are presented in Fig. 5.

The temperature fluctuation of the traditional single-pulse scheme can be as high as approximately 300 °C. In this manuscript, first, by leveraging the synergistic demodulation mechanism integrating EAC-coding (which compensates for attenuation via coding gain) and the refined processing of Haar wavelet denoising, effective suppression of temperature fluctuation noise along the optical fiber is achieved. This significantly reduces the system's temperature fluctuation to 22.51 °C while improving the SNR by 7.24 dB. Furthermore, the preprocessing architecture within EAC-coding

(autocorrelation characteristic analysis, waveform analysis and reconstruction of Raman backscattered signals) further reduces the system's temperature fluctuation to 20.86 °C by dynamically compensating for EDFA transient gain fluctuation and reconstructing Raman scattered signals. Meanwhile, the SNR is further improved by an additional 1.48 dB.

Compared with the temperature resolution of 29.25 °C achieved by the traditional single-pulse scheme, the EAC-coding scheme (integrating coding gain with Haar wavelet denoising) improves the system's temperature resolution to 5.82 °C. Meanwhile, the preprocessing architecture within EAC-coding further enhances the system's temperature resolution to 5.39 °C by dynamically compensating for EDFA transient gain fluctuation and reconstructing Raman scattered signals.

Meanwhile, we further compared the temperature measurement accuracy performance of the system, as shown in Fig. 4(d1) - (d3). It can be observed that the transient effects not only affect the SNR, temperature fluctuation, and temperature resolution but also exert a significant impact on the system's temperature measurement accuracy. It can be observed that as the temperature of the fiber under test (FUT) gradually increases, the temperature measurement error of the distorted signal demodulation results becomes increasingly large. This confirms that the influence of the transient effect also intensifies with rising temperature—with the temperature measurement accuracy deteriorating to a maximum of 14.98 °C at 90.0 °C (where the demodulation result is 104.98 °C). This is because with increasing temperature, the elevation of the Raman backscattered curve at the FUT becomes more pronounced, while transient effects exert an increasingly significant influence on signal intensity. After the preprocessing architecture in the EAC-coding scheme (autocorrelation characteristic analysis, waveform analysis and reconstruction of Raman backscattered signals), the temperature demodulation result is 90.91 °C, and the temperature measurement accuracy is 0.91 °C, representing an improvement of approximately 16.46 times. Therefore, the EAC-coding scheme proposed in this manuscript can effectively improve the system's temperature measurement accuracy, enhancing the temperature measurement accuracy of the long-distance Raman distributed optical fiber sensing system (with a sensing distance of 70.0 km) to within 1.0 °C.

In conclusion, the innovation of this manuscript lies in two aspects. First, by leveraging the synergistic demodulation mechanism integrating EAC-coding (which compensates for attenuation via coding gain) and the refined processing of Haar wavelet denoising, this manuscript breaks through the theoretical trade-off bottleneck between SNR and sensing distance in traditional schemes. Experimental results demonstrate that at an ultra-long distance of 70.0 km, the system achieves a spatial resolution of 1.58 m and a temperature resolution of 5.39 °C, with the number of effective sensing points reaching 44,303. To the best of our knowledge, this number of effective sensing points represents the highest level in the field of Raman distributed optical fiber sensing. Second, the preprocessing architecture within EAC-coding (autocorrelation characteristic analysis, waveform analysis and reconstruction of Raman backscattered signals) significantly enhances the temporal fidelity and transmission stability of encoded sequences by dynamically compensating for EDFA transient gain fluctuation. This further improves the system's SNR and temperature measurement accuracy, enabling the temperature measurement accuracy to reach 0.91 °C.

Fig.5. Experimental comparison results of temperature sensing. (a1) Temperature demodulation results at room temperature based on single-pulse scheme. (a2) Temperature demodulation results at room temperature based on distorted detection signals. (a3) Temperature demodulation results at room temperature based on proposed EAC-coding scheme. (b1) SNR based on single-pulse scheme. (b2) SNR based on EAC-coding scheme (without pre-processing). (b3) SNR based on proposed EAC-coding scheme. (c1) Temperature resolution based on single-pulse scheme. (c2) Temperature resolution based on EAC-coding scheme (without pre-processing). (c3) Temperature resolution based on proposed EAC-coding scheme. (d1) Temperature demodulation results at 50 °C based on EAC-coding scheme and EAC-coding scheme (without pre-processing). (d2) Temperature demodulation results at 70 °C based on EAC-coding scheme and EAC-coding scheme (without pre-processing). (d3) Temperature demodulation results at 90 °C based on EAC-coding scheme and EAC-coding scheme (without pre-processing).

Thank you for your suggestion. We have made the following modifications in our revised manuscript:

- 1) In the section of **Experimental results**, page 15, left column,

paragraph 2, line 5, add “The results of temperature demodulation and SNR are presented in Fig. 6. The temperature fluctuation of the traditional single-pulse scheme can be as high as approximately 300 °C, as shown in Fig. 6(a1). In this paper, first, by leveraging the synergistic demodulation mechanism integrating EAC-coding (which compensates for attenuation via coding gain) and the refined processing of Haar wavelet denoising, effective suppression of temperature fluctuation noise along the optical fiber is achieved. This significantly reduces the system’s temperature fluctuation to 22.51 °C while improving the SNR by 7.24 dB, as shown in Fig. 6(a2) and Fig. 6(b2). Furthermore, the preprocessing architecture within EAC-coding (autocorrelation characteristic analysis, waveform analysis and reconstruction of Raman backscattered signals) further reduces the system’s temperature fluctuation to 20.86 °C by dynamically compensating for EDFA transient gain fluctuation and reconstructing Raman scattered signals. Meanwhile, the SNR is further improved by an additional 1.48 dB, as shown in Fig. 6(a3) and Fig. 6(b3).

Compared with the temperature resolution of 29.25 °C achieved by the traditional single-pulse scheme, the EAC-coding scheme (integrating coding gain with Haar wavelet denoising) improves the system’s temperature resolution to 5.82 °C, as shown in Fig. 6(c2). Meanwhile, the preprocessing architecture within EAC-coding further enhances the system’s temperature resolution to 5.39 °C by dynamically compensating for EDFA transient gain fluctuation and reconstructing Raman scattered signals, as shown in Fig. 6(c3).

Meanwhile, we further compared the temperature measurement accuracy performance of the system, as shown in Fig. 6(d1) - (d3). It can be observed that the transient effects not only affect the SNR, temperature fluctuation, and temperature resolution but also exert a significant impact on the system’s temperature measurement accuracy. It can be observed that as the temperature of the fiber under test (FUT) gradually increases, the temperature measurement error of the distorted signal demodulation results becomes increasingly large. This confirms that the influence of the transient effect also intensifies with rising temperature—with the temperature measurement accuracy deteriorating to a maximum of 14.98 °C at

90.0 °C (where the demodulation result is 104.98 °C). This is because with increasing temperature, the elevation of the Raman backscattered curve at the FUT becomes more pronounced, while transient effects exert an increasingly significant influence on signal intensity. After the preprocessing architecture in the EAC-coding scheme (autocorrelation characteristic analysis, waveform analysis and reconstruction of Raman backscattered signals), the temperature demodulation result is 90.91 °C, and the temperature measurement accuracy is 0.91 °C, representing an improvement of approximately 16.46 times. Therefore, the EAC-coding scheme proposed in this paper can effectively improve the system's temperature measurement accuracy, enhancing the temperature measurement accuracy of the long-distance Raman distributed optical fiber sensing system (with a sensing distance of 70.0 km) to within 1.0 °C.

Through the synergistic demodulation mechanism of EAC-coding (which utilizes coding gain to compensate for attenuation) combined with the refined processing of Haar wavelet denoising, the theoretical trade-off bottleneck between SNR and sensing distance in traditional schemes is broken through. Experimental results demonstrate that at an ultra-long distance of 70.0 km, the system achieves a spatial resolution of 1.58 m and a temperature resolution of 5.39 °C. Moreover, the preprocessing architecture in EAC-coding (integrating autocorrelation characteristic analysis of Raman backscattered signals and waveform analysis and reconstruction) significantly enhances the temporal fidelity and transmission stability of coded pulse sequences by dynamically compensating for EDFA transient gain fluctuation. This further improves the system's SNR and temperature measurement accuracy, enabling the temperature measurement accuracy to reach 0.91 °C. In this study, the improvements in SNR (accompanied by performance enhancements in sensing distance and temperature resolution) are primarily attributed to the proposed EAC-coding (for coding gain compensation) and Haar wavelet denoising schemes. In contrast, the preprocessing architecture within EAC-coding (characterized by autocorrelation characteristic analysis, waveform analysis and reconstruction of Raman backscattered

signals) serves to improve the system's temperature measurement accuracy by dynamically compensating for EDFA transient gain fluctuation and reconstructing Raman scattered signals.”

5. What do the authors mean by “destroyed detection” signals? Needs to be explained properly.

Reply: We sincerely apologize for the inappropriate terminology used in this manuscript. In the manuscript, the phrase “destroyed detection signals” refers to detection signals whose autocorrelation characteristics have been impaired by transient effects—specifically, detection signals with waveform distortion caused by transient effects, which correspond to the red curves $u(f)$, $\bar{u}(f)$, $w(f)$, $\bar{w}(f)$ in Fig. 3(a1) - (a4) of the manuscript. For the sake of clarity, we have replaced all instances of the phrase "destroyed detection signals" with "distorted detection signals" in both the main text and figure captions of the manuscript. Additionally, we have supplemented the original text with a glossary explanation of this signal (i.e., “distorted detection signals”).

Thank you for your suggestion. We have made the following modifications in our revised manuscript:

- 1) In the section of **Principles and methods**, page 5, change “Fig. 1” to “Fig. 1”.

Fig. 1. Physics principle of proposed EAC-coding scheme for Raman distributed optical fiber sensing. (a) Principle of encoding based on Golay complementary sequences. (b) The influence of transient effects on Raman backscattered signals. (c) RBSs generated by four distorted detection signals in sensing fiber; (d) Principle of decoding after analysis and waveform-reconstruction based on proposed EAC-coding scheme. (e)

Temperature demodulation principle.

- 2) In the section of **Principles and methods, page 5, left column, paragraph 3, line 18, change “destroyed detection signals” to “distorted detection signals”.**
- 3) In the section of **Experimental results, page 11, right column, paragraph 2, line 20, change “destroyed detection signals” to “distorted detection signals”.**
- 4) In the section of **Experimental results, page 11, right column, paragraph 2, line 27, change “destroyed detection signals” to “distorted detection signals”.**

6. In Fig. 6(a), the EAC scheme is shown to provide 8.75 dB improvement over the single pulse scheme. How is this 8.75 dB explained? For a 64 bit code, we expect an SNR improvement of $\sqrt{64}/2 = 4$, which is 6 dB.

Reply: We would like to express our sincere gratitude for your professional comments and also thank you for pointing out this issue. The maximum input optical power of the sensing fiber also affects the improvement of the theory SNR of its coding scheme. We are very sorry that some of the vague language in the manuscript overlooked this point, and we have added these contents in the revised manuscript. In this manuscript, based on the maximum optical power (0.2 mW) that can be achieved in conventional single-pulse demodulation scheme, the theoretical SNR of 64-bit code demodulation scheme is improved to 6 dB under this power condition. The EAC demodulation scheme proposed can enhance the stimulated Raman scattering (SRS) threshold of sensing fibers, enabling a significant increase in optical power input to the sensing fibers, reaching up to 0.7 mW. At this power level, the theoretical SNR of EAC encoding and demodulation is improved to 11.46 dB. In actual experiment, since the EDFA introduces a significant amount of amplified spontaneous emission (ASE) noise while amplifying the effective signal, the final experimental improvement in SNR is 8.72 dB, which is lower than the theoretical value of 11.46 dB. We have also revised this section in the updated manuscript. The specific reasons are as follows.

The establishment of the theoretical pulse gain is based on the strict premise that the input powers of the single-pulse demodulation scheme and the pulse coding scheme are identical. In the experimental scheme of this

paper, the input power is set to be close to the maximum value of the stimulated Raman scattering nonlinear effect threshold of the sensing fiber.

For the coding technology with a code length of 64 bits (i.e., the EAC-coding scheme adopted in this manuscript). Under the maximum input power achievable by a single pulse demodulation system (0.2 mW), the theoretical SNR improvement is 6 dB.

However, the theoretical 6 dB SNR improvement is based on the strict premise that the input powers of the single-pulse demodulation scheme and the pulse coding scheme are the same. In this experiment, to explore the extreme performance of the EAC-coding scheme and maximize the system SNR, we set the input power to be close to the maximum value of the SRS nonlinear effect threshold of the sensing fiber. Specifically, the average power input into the optical fiber is approximately 0.7 mW, and the calculated peak power is close to 1.4 W.

In contrast, the traditional single-pulse demodulation scheme, due to its higher peak power (under the same average power), is more likely to trigger the SRS nonlinear effect, leading to signal distortion and performance degradation. Therefore, to ensure the stable operation of the single-pulse scheme below the nonlinear threshold, its average input power must be significantly reduced, which is set to approximately 0.2 mW in the experiment.

The different incident powers of the two schemes are incorporated into the calculation of the theoretical SNR improvement. According to Eq. (4) in the manuscript, with all other factors remaining completely unchanged, the intensity of the Raman backscattered signal depends solely on the incident optical power. Thus, the modified theoretical SNR improvement is approximately $10\log_{10}[(\sqrt{64/2}) \times (0.7/0.2)] = 11.46$ dB. Meanwhile, since the EDFA introduces a significant amount of amplified spontaneous emission (ASE) noise while amplifying the effective signal, the final experimental improvement in SNR is 8.72 dB, which is slightly lower than the theoretical value of 11.46 dB.

In summary, it is precisely because the EAC-coding scheme allows a higher average input power without triggering significant nonlinear effects, and combined with its inherent theoretical coding gain, the SNR improvement observed in the experiment finally reaches 8.72 dB, which is

slightly lower than the theoretical SNR coefficient (11.46 dB) at this power setting (0.7 mW).

Thank you for your suggestion. We have made the following modifications in our revised manuscript:

- 1) In the section of **Discussion, page 22, right column, paragraph 3, line 18**, add “For the coding technology with a code length of 64 bits (i.e., the EAC-coding scheme adopted in this paper). Under the maximum input power achievable by a single pulse demodulation system (0.2 mW), the theoretical SNR improvement is 6 dB.

However, the theoretical 6 dB SNR improvement is based on the strict premise that the input powers of the single-pulse demodulation scheme and the pulse coding scheme are the same. In this experiment, to explore the extreme performance of the EAC-coding scheme and maximize the system SNR, we set the input power to be close to the maximum value of the stimulated Raman scattering (SRS) nonlinear effect threshold of the sensing fiber. Specifically, the average power input into the optical fiber is approximately 0.7 mW, and the calculated peak power is close to 1.4 W.

In contrast, the traditional single-pulse demodulation scheme, due to its higher peak power (under the same average power), is more likely to trigger the SRS nonlinear effect, leading to signal distortion and performance degradation. Therefore, to ensure the stable operation of the single-pulse scheme below the nonlinear threshold, its average input power must be significantly reduced, which is set to approximately 0.2 mW in the experiment.

The different incident powers of the two schemes are incorporated into the calculation of the theoretical SNR improvement. According to Eq. (4) in the paper, with all other factors remaining completely unchanged, the intensity of the Raman backscattered signal depends solely on the incident optical power. Thus, the modified theoretical SNR improvement is approximately $10\log_{10}[(\sqrt{64/2}) \times (0.7/0.2)] = 11.46$ dB. Since the EDFA introduces a significant amount of amplified spontaneous emission (ASE) noise while amplifying the effective signal, the final experimental improvement in SNR is 8.72 dB, which is slightly lower than the

theoretical value of 11.46 dB.

In summary, it is precisely because the EAC-coding scheme allows a higher average input power without triggering significant nonlinear effects, and combined with its inherent theoretical coding gain, the SNR improvement observed in the experiment finally reaches 8.72 dB, which is slightly lower than the theoretical SNR coefficient (11.46 dB) at this power setting (0.7 mW).”

7. Also, if we check the difference between the two curves at 28 km, the improvement seems to be more than 10 dB. How can you explain this?

Reply: We would like to express our gratitude to the reviewer for the valuable comments and in-depth corrections on this issue. In the experiment, within a range from the starting end of the sensing fiber to 56.0 km, the SNR improvement of the sensing signal is about 10.44 dB; while in the region beyond approximately 70.0 km at the tail end of the optical fiber, the SNR improvement is 8.72 dB. The improvement of SNR at each sensing distance point is lower than the theoretical value of 11.46 dB. The improvement of SNR at the end of sensing fiber is also lower than that at the front end of sensing fiber. This discrepancy primarily stems from the synergistic effect between the difference in SNR improvement between experiments and theory conditions, and the distance-dependent characteristics of fiber attenuation. The specific theoretical analysis is as follows.

After revising the SNR calculation formula, the overall SNR improvement along the optical fiber obtained from the experimental results is slightly lower than the theoretical SNR. Since the maximum input optical power of the sensing fiber affects the theoretical SNR improvement of its coding scheme, the final SNR magnitude calculated for the EAC-coding scheme is 11.46 dB. Meanwhile, due to the introduction of the EDFA, a large amount of amplified spontaneous emission (ASE) noise is introduced during the process of amplifying the detection signal—this results SNR improvement in the experiment being slightly lower than the theoretical value of 11.46 dB. Consequently, the SNR improvement along the optical fiber derived from the experimental results (with a maximum of 10.44 dB) is slightly lower overall than the theoretical SNR (11.46 dB).

Furthermore, fiber attenuation exhibits a cumulative effect with transmission distance. At 70.0 km, the single-pulse signal is already close to

the system's detection limit due to severe attenuation, making it difficult to extract effectively. Although the EAC-coding scheme still maintains a certain signal advantage, the intensity difference between the EAC-coding signal and the single-pulse signal is smaller than that at 28.0 km, as the signal amplitude decreases significantly after long-distance transmission. Consequently, the SNR improvement at this location decreases to 8.72 dB. This result clearly reflects the distance-dependent characteristic of fiber signal attenuation and also highlights the performance difference of the coding scheme at different transmission distances.

Thank you for your suggestion. We have made the following modifications in our revised manuscript:

- 1) In the section of **Discussion, page 15, right column, paragraph 2, line 5**, add “In the experiment, within a range from the starting end of the sensing fiber to 56.0 km, the SNR improvement of the sensing signal is about 10.44 dB; while in the region beyond approximately 70.0 km at the tail end of the optical fiber, the SNR improvement is 8.72 dB. The improvement of SNR at each sensing distance point is lower than the theoretical value of 11.46 dB. The improvement of SNR at the end of sensing fiber is also lower than that at the front end of sensing fiber. This discrepancy primarily stems from the synergistic effect between the difference in SNR improvement between experiments and theory conditions, and the distance-dependent characteristics of fiber attenuation.

After revising the SNR calculation formula, the overall SNR improvement along the optical fiber obtained from the experimental results is slightly lower than the theoretical SNR. Since the maximum input optical power of the sensing fiber affects the theoretical SNR improvement of its coding scheme, the final SNR magnitude calculated for the EAC-coding scheme is 11.46 dB. Meanwhile, due to the introduction of the EDFA, a large amount of amplified spontaneous emission (ASE) noise is introduced during the process of amplifying the detection signal—this results SNR improvement in the experiment being slightly lower than the theoretical value of 11.46 dB. Consequently, the SNR improvement along the optical fiber derived from the experimental results (with a

maximum of 10.44 dB) is slightly lower overall than the theoretical SNR (11.46 dB).

Furthermore, fiber attenuation exhibits a cumulative effect with transmission distance. At 70.0 km, the single-pulse signal is already close to the system's detection limit due to severe attenuation, making it difficult to extract effectively. Although the EAC-coding scheme still maintains a certain signal advantage, the intensity difference between the EAC-coding signal and the single-pulse signal is smaller than that at 28.0 km, as the signal amplitude decreases significantly after long-distance transmission. Consequently, the SNR improvement at this location decreases to 8.72 dB. This result clearly reflects the distance-dependent characteristic of fiber signal attenuation and also highlights the performance difference of the coding scheme at different transmission distances.”

8. What is being plotted in Fig. 7(a)? Absolute temperature or uncertainty in temperature? I presume it is the latter, in which case it should be clearly mentioned. How is 6x improvement achieved? Need proper explanation?

Reply: We wish to express our sincere agreement with the reviewers' insightful comments, the Fig. 7(a) (figure number of the manuscript before revision) depicts the temperature uncertainty, which is used to define the temperature resolution. Temperature resolution is defined as the minimum temperature change that a system can reliably distinguish, and it is one of the key parameters for evaluating Raman distributed optical fiber sensing systems.

In Raman distributed optical fiber sensing, the standard calculation method for temperature resolution is based on the statistical fluctuation of temperature demodulation results under steady-state conditions. Specifically, it is characterized and calculated by the standard deviation of temperature demodulation values within a specific length under a constant temperature environment ^{[1][2]}.

The proposed scheme compares the temperature resolution performance between the traditional single-pulse demodulation scheme and the proposed EAC-coding scheme. In this experiment, temperature resolution is

characterized by the temperature standard deviation of each distributed temperature measurement point within a fixed length of the sensing fiber. Experimental results show that at a sensing distance of 70.0 km, the temperature resolution of the EAC-coding scheme is significantly improved compared with that of the single-pulse scheme: the temperature resolution is enhanced from 29.25 °C (of the single-pulse scheme) to 5.39 °C, representing an improvement of up to 5.43 times.

[1] X. Liu, *et al.* High-speed and high-resolution YAG fiber based distributed high temperature sensing system empowered by a 2D image restoration algorithm. *Opt. Express*, **31**(4): 6170-6183 (2023).

[2] J. Gasser, *et al.* Distributed temperature sensor combining centimeter resolution with hundreds of meters sensing range. *Opt. Express*, **30**(5): 6768-6777 (2022).

Thank you for your suggestion. We have made the following modifications in our revised manuscript:

- 1) In the section of **Experimental results**, page 16, left column, paragraph 3, line 37, add “In this experiment, temperature resolution is characterized by the temperature standard deviation of each distributed temperature measurement point within a fixed length of the sensing fiber. Experimental results show that at a sensing distance of 70.0 km, the temperature resolution of the EAC-coding scheme is significantly improved compared with that of the single-pulse scheme: the temperature resolution is enhanced from 29.25 °C (of the single-pulse scheme) to 5.39 °C, representing an improvement of up to 5.43 times.”

Corrections:

We would like to express our heartfelt thanks to all reviewers for the careful review of our manuscript and your valuable comments amid your busy schedules. The professional suggestions you provided have offered important guidance for improving the quality of the manuscript, and we would like to extend our sincere gratitude for this.

To clearly present the full picture of revisions to the manuscript and facilitate the reviewers' verification of the implementation of these revisions, we have summarized all the revisions and supplements made based on the reviewers' comments.

Once again, we appreciate the attentive guidance from all reviewers. Should you have any details requiring further communication or additional suggestions for improvement during subsequent verification, we will respond promptly and fully cooperate, and continue to polish the manuscript with the most rigorous attitude to ensure it meets the high standards for publication in the journal.

Thank you for your suggestion. We have made the following modifications in our revised manuscript:

- 1) In the section of **Experimental results, page 17, right column, paragraph 2, line 11, add** “In the proposed scheme, the pulse width of a single code element is set to 10 ns, corresponding to a theoretical spatial resolution of 1.0 m. Furthermore, single-mode fiber is employed to avoid the degradation of spatial resolution caused by intermodal dispersion in the sensing fiber. However, the intensity of Raman scattering signals excited in single-mode fiber is relatively low. Therefore, in the experiment, to further improve the SNR of the sensing signals, Haar wavelet denoising processing is applied to the Raman anti-Stokes scattering signals after the decoding process. It is suitable for processing low-complexity signals with mutation characteristics, thus making it well-suited for handling fiber optic Raman scattering signals. Meanwhile, in this paper, after temperature demodulation, the db10 wavelet method is synchronously applied to the distributed temperature demodulation curve to further improve the SNR of the sensing signal. However, this denoising method will filter out the edge information of the temperature envelope in the sensing fiber region, resulting in a final experimentally measured spatial resolution of 1.58 m, which differs from the theoretical spatial resolution of 1.0 m.”
- 2) In the section of **Experimental setup, page 9, right column, paragraph 2, line**

4, add “In the experimental process of this scheme, only the Raman anti-Stokes scattering signals were analyzed and processed, and the subsequent temperature demodulation was also performed solely based on the Raman anti-Stokes scattering signals. Compared with the dual-path demodulation scheme (where Raman Stokes scattering light is used to demodulate Raman anti-Stokes scattering signals), this scheme has the following two specific advantages. The Raman anti-Stokes scattering signal based on single-path demodulation exhibits higher sensitivity to the temperature distribution along the sensing fiber; The single-path demodulation scheme can enhance the system's measurement rate and reduce system costs.”

- 3) In the section of **Experimental setup, page 8, right column, paragraph 2, line 5, add** “The equipment used in the experiment, along with their corresponding model numbers, manufacturers, and key optical parameters, are listed as follows: Laser (KEYANG PHOTONICS; DFB Continuous laser; Center Wavelength:1550 nm);SOA (OPEAK; OAM-SOA-PL-15-15-S; Operating Wavelength: 1520-1570 nm); DDPG (CIQTEK; ASG8100; Maximum Coding bit: 99 bit); EDFA (OPEAK; EDFA-C-PL-MB-100-S; Operating Wavelength: 1550 nm); WDM (OPEAK; WDM-1*3-1550; Isolation: ≥ 60 dB); APD (KEYANG PHOTONICS; KY-DTS-200M; Bandwidth:200 MHz); DAC (CIQTEK; DAQ2100; Sampling Rate: 1 GSa/s). And all experimental equipment used in this experiment are conventional commercial products available on the market.”
- 4) In the section of **Experimental results, page 12, right column, change** “Fig. 5” to “Fig. 5”

Fig. 5. Intensity of Raman backscattered signals based on EAC-coding scheme and conventional single-pulse. (a) Intensity of Raman backscattered signals based on EAC-coding scheme and conventional single-pulse with and without Haar wavelet. (b1) Intensity of Raman backscattered signals based on conventional single-pulse at the end of sensing fiber. (b2) Intensity of Raman backscattered signals based on EAC-coding scheme at the end of sensing fiber. (c1) SNR of Raman backscattered signals based on conventional single-pulse scheme at the end of sensing fiber. (c2) SNR of Raman backscattered signals based on EAC-coding scheme at the end of sensing fiber.

5) In the section of **Experimental results**, page 13, right column, paragraph 2, line 11, add “As shown in Fig. 5(c1) and 5(c2), the Haar wavelet function significantly improves the SNR for both the EAC-coding scheme proposed in this paper and the traditional single-pulse demodulation scheme. Experimental results indicate that after Haar wavelet denoising, the SNR of the EAC-coding scheme and the traditional single-pulse demodulation scheme is increased by 7.63 dB and 2.92 dB, respectively. It is worth noting that a relatively significant difference emerges between the two demodulation schemes at the end of the sensing fiber. For single-pulse scheme, the effective Raman scattering signal is basically submerged by system noise beyond 42.0 km. After incorporating the Haar wavelet function, although the noise level is reduced, the signal is still submerged

by noise beyond 56.0 km. Therefore, the improvement in SNR at the end of the sensing fiber is relatively limited for the single-pulse scheme. This also confirms that the wavelet denoising algorithm has a limited capability to improve SNR in long-distance distributed optical fiber sensing demodulation schemes. Hence, we combine the EAC-coding scheme with Haar wavelet denoising. Only by avoiding the effective Raman scattering signal being submerged by system noise as much as possible can the wavelet denoising algorithm maximize the SNR of the sensing signal.”

- 6) In the **(unmodified manuscript)** section of **Introduction, page 2, left column, paragraph 3, line 35, delete** “The pulse coding scheme employs specific functional equations to encode detection signals, and then decodes the collected Raman backscattered signals through a decoding program [21-22]. It increases the intensity of the detection signals without the stimulated Raman generating nonlinear effects and thus improve the SNR of system. For example, Sun *et al.* proposed a genetic-optimised aperiodic coding scheme achieving a sensing spatial resolution of 1.0 m at the sensing distance of 39.0 km [23]. With the development of digital signal processing, various denoising schemes are used in Raman distributed optical fiber sensing, including wavelet denoising algorithm and dynamic sampling-correction method [24-25]. For example, M. Yu, *et al.* proposed a dynamic sampling-correction scheme achieving a sensing distance of 20.0 km [26]. H. S. Pradhan, *et al.* proposed a Fourier wavelet regularised deconvolution scheme achieving a sensing distance of 30.0 km with a spatial resolution of 3.0 m [27].

Furthermore, the neural network scheme can learn the characteristic patterns of Raman scattering signals, perform convolutional operations and subsequent processing on the input signals to effectively identify and remove noise components, thereby improving the SNR [28]. For example, Zhang *et al.* proposed a deep 1-D denoising convolutional neural network scheme, which realized a sensing distance of 10 km with a spatial resolution of 3 m [29].

However, in large-scale engineering infrastructures, fiber deployment typically adopts U-shaped or Hilbert-curve configurations, necessitating length of sensing fiber several times infrastructures. Despite employing advanced denoising schemes in Raman distributed fiber sensing [30-32], achieving a further breakthrough in sensing distance remains challenging under meter-scale spatial resolution constraints.”

- 7) In the section of **Introduction, page 2, right column, paragraph 4, line 34, add** “Pulse coding schemes have demonstrated significant advantages in

long-distance Raman distributed optical fiber sensing system. On one hand, by encoding the detection optical pulses, this technology can significantly increase the optical power injected into the sensing fiber while avoiding fiber nonlinear effects, thereby enhancing the intensity of effective sensing signals at the level of signal generation mechanism. On the other hand, at the data processing level, it leverages the separability between coded signals and uncoded noise to effectively suppress noise and improve the SNR of system.

Currently, pulse coding schemes applied in Raman distributed optical fiber sensing system are primarily classified into five categories, which specifically include the Simplex coding scheme, Pre-Shaped Simplex Coding scheme, low-repetition-rate cyclic pulse coding scheme, Genetic-optimised aperiodic coding scheme, Derived Sequences coding scheme.

Simplex coding scheme, as a linear coding technique based on Hadamard matrix transformation, has its code length (in L bits) determining the number of detection signals required (L groups). For instance, in 2006, J. Park, *et al.* achieved a spatial resolution of 17.0 m, a temperature resolution of 3.0 °C, and a total of 2176 effective sensing points over a sensing distance of 37.0 km by utilizing this coding scheme and link optimization techniques^[31].

Low-repetition-rate cyclic pulse coding scheme generates Simplex coding-based cyclic coding sequences using an acousto-optic modulator (AOM), enabling pulses to be injected into the optical fiber periodically at a low repetition rate. For example, in 2011, M. A. Soto, *et al.* combined the low-repetition-rate quasi-periodic cyclic pulse coding with a high-power fiber laser, achieving a spatial resolution of 1.0 m, a temperature resolution of 3.0 °C, and 26,000 effective sensing points over a sensing distance of 26.0 km^[27].

Pre-Shaped Simplex Coding scheme is based on simplex coding technology and adopts a linearly increasing profile to adjust pulse amplitude, compensating for the amplitude variation caused by EDFA to suppress the transient effect of the EDFA and improve system performance. For example, in 2017, J. B. Rosolem, *et al.* achieved a spatial resolution of 10.0 m, a temperature resolution of 8.4 °C, and 6,200 effective sensing points over a sensing distance of 62.0 km by using the pre-shaped Simplex coding and a gain-controlled erbium-doped fiber amplifier^[32].

Genetic-optimised aperiodic coding scheme utilizes a distributed genetic algorithm (DGA) to generate a single-sequence aperiodic code (GO-code), which is then converted into an optical pulse sequence and injected into the optical fiber. For example, in 2020, X. Z. Sun, *et al.* achieved a spatial resolution of 1.0 m, a

temperature resolution of 3.9 °C, and 39,000 effective sensing points over a sensing distance of 39.0 km based on the Genetic-optimised aperiodic coding scheme [29].

Derived Sequences coding scheme addresses the transient effect of EDFA through a derived sequence decoding method, thereby enhancing the performance of long-distance sensing. For instance, in 2022, D. D. Chai, *et al.* achieved a spatial resolution of 5.0 m, a temperature resolution of 2.5 °C, and 8,800 effective sensing points over a sensing distance of 44.0 km based on the Derived Sequences coding scheme [33].”

- 8) In the section of **Experimental setup, page 9, right column, paragraph 3, line 20, add** “To ensure the environmental temperature stability of the sensing fiber during the long-term experiment, the entire 70.0 km sensing fiber is placed in an ultra-clean constant-temperature laboratory, where the ambient temperature is maintained at 24.0±1.0 °C. The FUT is placed in a high-precision constant-temperature water bath with a temperature fluctuation of less than 0.10 °C. Thus, the above experimental setup can guarantee the temperature stability of the testing environment for the sensing fiber during long-term measurements.”
- 9) In the section of **Experimental results, page 14, change** “Fig. 6, Fig .7” to “Fig. 6”

Fig.6. Experimental comparison results of temperature sensing. (a1) Temperature demodulation results at room temperature based on single-pulse scheme. (a2) Temperature demodulation results at room temperature based on distorted detection signals. (a3) Temperature demodulation results at room temperature based on proposed EAC-coding scheme. (b1) SNR based on single-pulse scheme. (b2) SNR based on EAC-coding scheme (without pre-processing). (b3) SNR based on proposed EAC-coding scheme. (c1) Temperature resolution based on single-pulse scheme. (c2) Temperature resolution based on EAC-coding scheme (without pre-processing). (c3) Temperature resolution based on proposed EAC-coding scheme. (d1) Temperature demodulation results at 50 °C based on EAC-coding scheme and EAC-coding scheme (without pre-processing). (d2) Temperature demodulation results at 70 °C based on EAC-coding scheme and EAC-coding scheme (without pre-processing). (d3) Temperature demodulation results at 90 °C based on EAC-coding scheme and EAC-coding scheme (without pre-processing).

10) In the section of **Experimental setup**, page 8, right column, change “Fig. 2” to “Fig. 2”

Fig. 2. Experimental setup based on proposed EAC-coding scheme for Raman distributed optical fiber sensing.

11) In the section of **Experimental setup**, page 10, left column, change “Fig. 3” to “Fig. 3”

Fig. 3. Timing of detection signals and autocorrelation function. (a1) Timing of the detection signal u before and after transient effects. (a2) Timing of the detection signal \bar{u} before and after transient effects. (a3) Timing of the detection signal w before and after transient effects. (a4) Timing of the detection signal \bar{w} before and after transient effects. (b1) Autocorrelation function of the detection signals before transient effects. (b2) Autocorrelation function of the detection signals after transient effects.

12) In the section of **Introduction**, page 2, right column, paragraph 2, line 3, add “Algorithm denoising schemes aim to improve the SNR of the system by processing optical fiber scattering signals. For example, M. Yu, *et al.* proposed a dynamic sampling-correction scheme achieving a sensing distance of 20.0 km^[23].”

H. S. Pradhan, *et al.* proposed a Fourier wavelet regularised deconvolution scheme achieving a sensing distance of 30.0 km with a spatial resolution of 3.0 m [24]. Essentially derived from mathematical optimization theory, such schemes only denoise signals from the perspective of signal processing without achieving breakthroughs in physical mechanisms. Furthermore, they have limitations including parameter dependence and difficulty in preserving the true characteristics of signals [30].

In addition, as one of the current mainstream denoising methods, convolutional neural networks (CNNs) scheme is a typical data-driven approach. It optimizes model parameters using large-scale training datasets, with the goal of achieving effective processing of test data. For example, Z. Zhang, *et al.* proposed a deep 1-D denoising convolutional neural network scheme, which realized a sensing distance of 10 km with a spatial resolution of 3 m [26]. However, the performance of this method is significantly constrained by the specificity of training data; when there is a notable distribution difference between test signals and the training set, its effect on enhancing sensing performance will be limited.”

- 13) In the **(unmodified manuscript)** section of **Introduction, page 3, left column, paragraph 2, line 3, delete**“ In this work, an enhanced autocorrelation pulse coding (EAC-coding) scheme is proposed, which introduces Golay complementary sequences into Raman distributed optical fiber sensing, and at the same time restores the autocorrelation which is destroyed by the transient effects of erbium-doped fiber amplifier (EDFA) by preprocessing the Raman backscattered signals for analysis and waveform-reconstruction. The proposed scheme further incorporates Haar wavelet denoising algorithm to enhance system performance. In the experiment, a Raman distributed optical fiber sensing with sensing distance of 70.0 km, spatial resolution of 1.58 m and temperature resolution of 5.39 °C is achieved, which provides a new solution for the research of long-distance Raman distributed optical fiber sensing.”
- 14) In the section of **Introduction, page 4, left column, paragraph 3, line 22, add** “Currently, existing pulse coding techniques in Raman distributed optical fiber sensing remain constrained by several prevalent bottlenecks: high hardware implementation complexity, excessive computational overhead in encoding and decoding, limited nonlinear effect suppression capability, and inadequate resistance to signal distortion coupled with poor stability during long-distance transmission. Collectively, these factors constrain the system’s application potential in high-performance scenarios—including low SNR environments and long-distance sensing applications.

To address the aforementioned issues, an Enhanced anti-distortion coding (EAC-coding) scheme is proposed in this paper. This scheme effectively overcomes the trade-off dilemma among SNR, sensing distance, and hardware cost in traditional methods through a three-fold synergistic mechanism: compensating for EDFA fluctuation via a preprocessing framework, suppressing transmission losses through coding gain, and finely removing residual noise using wavelet transform. While maintaining relatively low complexity, the EAC-coding scheme significantly enhances the anti-distortion capability and adaptability of the system, providing a new solution for long-distance, high-performance Raman distributed optical fiber sensing. Experimental results demonstrate that at an ultra-long distance of 70.0 km, the system achieves a spatial resolution of 1.58 m and a temperature resolution of 5.39 °C, with the number of effective sensing points reaching 44,303. To the best of our knowledge, this number of effective sensing points represents the highest level in the field of Raman distributed optical fiber sensing. In addition, the preprocessing architecture within EAC-coding (autocorrelation characteristic analysis, waveform analysis and reconstruction of Raman backscattered signals) significantly enhances the temporal fidelity and transmission stability of encoded sequences by dynamically compensating for EDFA transient gain fluctuation. This further improves the SNR and temperature measurement accuracy performance, enabling the temperature measurement accuracy to reach 0.91 °C, verifying the engineering feasibility of the scheme.”

- 15) In the **(unmodified manuscript)** section of **Experimental results**, **page 10, right column, paragraph 2, line 29**, delete “Raman distributed optical fiber sensing has emerged as a pivotal technology for continuous temperature monitoring in long-range infrastructure and environmental applications. However, its practical implementation over distances exceeding 30 km has been fundamentally hindered by the inherently low signal-to-noise ratio (SNR) of Raman scattering signals, which creates a critical trade-off between sensing distance and spatial resolution. In this study, we present a novel Raman distributed optical fiber sensing system utilizing the proposed EAC-coding scheme, which effectively suppresses transient effects and significantly enhances the SNR of system while incorporating Haar wavelet. This system enables long-distance Raman distributed temperature measurement with high spatial resolution, achieving precise temperature demodulation and accurate identification of FUT position and length. The experimental results demonstrate

that the proposed scheme achieves a spatial resolution of 1.58 m, temperature resolution of 5.39 °C, and temperature measurement accuracy of 0.88 °C at the sensing distance of 70.0 km, establishing a new approach for long-distance Raman distributed optical fiber sensing system. Future research will integrate more advanced denoising techniques to extend the system's sensing distance while preserving its spatial resolution capabilities.”

- 16) In the section of **Experimental results, page 18, left column, paragraph 2, line 10**, add “EAC-coding scheme proposed in this paper incorporates a novel preprocessing framework for autocorrelation characteristic analysis, waveform analysis and reconstruction of Raman backscattered signals. The preprocessing architecture for autocorrelation characteristic analysis, waveform analysis and reconstruction in this scheme can effectively suppress the transient effects introduced by EDFA, thereby significantly improving the amplitude fidelity and temporal stability of the encoded pulse sequence. This innovative active reconstruction mechanism in the signal domain, which is based on analyzing the waveforms of Raman backscattered signals via forward/inverse Fourier transforms, directly eliminates the phenomenon of detection pulse distortion existing in traditional demodulation schemes. The scope of action of the EAC-coding scheme encompasses two key aspects: specifically, one is addressing the fundamental limitation of detection signal distortion through physical-layer waveform reconstruction, and the other is performing correlation operations between the proposed coding sequence and the reconstructed Raman signal to restore the ideal cross-correlation characteristics of Golay sequences, and achieve a theoretical gain value, thereby breaking through the SNR-distance trade-off limit. It achieves the coordinated optimization of “waveform correction, correlation characteristic restoration, signal intensity enhancement”, laying a physical layer foundation for long-distance sensing.

Furthermore, the proposed EAC-coding scheme features a synergistically enhanced two-stage noise suppression mechanism, demonstrating innovations in multi-level joint denoising.

First, EAC-coding scheme enhances the SNR of sensing signals through its inherent coding gain. Based on the cross-correlation characteristics of its sequence design, this coding scheme provides a significant inherent gain, which compensates for signal attenuation caused by long-distance transmission and effectively improves the system's baseline SNR. Furthermore, this mechanism not only significantly boosts optical power but also leverages the irrelevance between the decoding algorithm and incoherent noise to achieve preliminary suppression

of transmission link losses at the encoding-decoding level.

Second, aiming at the residual noise with specific time-frequency domain distribution characteristics in the decoded signal, this scheme further introduces Haar wavelet transform for adaptive noise filtering. The multi-resolution analysis capability of the Haar wavelet enables accurate capture of the local time-frequency features of noise. By performing sparse representation of the signal at different scales (corresponding to different frequency bands) and time-domain positions, it achieves effective separation between useful signals and non-stationary noise components. While preserving the key details of the distributed temperature signal, this method efficiently eliminates residual noise, significantly enhancing the clarity of the signal in the time-frequency domain and the overall measurement accuracy of the system.

This three-fold synergistic mechanism, consisting of the preprocessing framework compensating for EDFA fluctuation, coding gain suppressing transmission losses, and wavelet transform finely removing residual noise, systematically breaks through the bottlenecks of traditional schemes in terms of SNR and sensing distance, providing a novel decoding framework for long-distance and high-precision Raman distributed optical fiber sensing.”

- 17) In the section of **Experimental results, page 13, left column, paragraph 2, line 18, add** “In this study, a denoising algorithm based on the Haar wavelet is adopted to process distributed temperature sensing signals with abrupt change characteristics. The inherent step-like form of the Haar wavelet basis function endows it with excellent feature extraction capability for singular points in the signal, especially for the abrupt temperature field change regions along the sensing fiber.”
- 18) In the section of **Introduction, page 3, left column, paragraph 3, line 27, add** “The main limitation of this scheme lies in the linear growth of data acquisition volume and computational complexity with code length, which restricts its practical efficiency in applications requiring long code lengths and high-speed sensing.”
- 19) In the section of **Introduction, page 3, right column, paragraph 1, line 9, add** “However, higher pulse energy also makes it more likely to induce fiber nonlinear effects, and when approaching the stimulated scattering threshold, it restricts the system’s sensing distance and SNR performance.”
- 20) In the section of **Introduction, page 3, right column, paragraph 2, line 25, add** “Although this scheme effectively suppresses the transient response of the EDFA at the hardware level, its hardware structure is more complex, leading to a

significant increase in implementation cost.”

- 21) In the section of **Introduction, page 4, left column, paragraph 1, line 3**, add “The core challenge of this scheme lies in its high dependence on optimization algorithms to search for the optimal code, which typically requires substantial computational resources and iterative testing.”
- 22) In the section of **Introduction, page 4, right column, paragraph 2, line 17**, add “However, this scheme’s decoding process imposes a heavy computational burden, and the core algorithm involves high-complexity matrix operations, making it difficult to meet real-time requirements.”
- 23) In the section of **Discussion, page 19, right column, paragraph 4, line 7**, add “Currently, the pulse coding techniques applied in Raman distributed optical fiber sensing system mainly include the following: Simplex coding scheme, Pre-Shaped Simplex coding scheme, Low-repetition-rate cyclic pulse coding scheme, Genetic-optimised aperiodic coding scheme, and Derived Sequences coding scheme. Their key performance indicators are presented in Table 2.

Table 2. Sensing performance of various coding schemes.

Scheme	Sensing Distance	Spatial Resolution	Sensing Points	Temperature Resolution	Measurement Time
Simplex coding ^[31]	37.0 km	17.0 m	2176	3.0 °C	/
Low-repetition-rate cyclic pulse coding ^[27]	26.0 km	1.0 m	26000	3.0 °C	30 s
Pre-Shaped Simplex Coding ^[32]	62.0 km	10.0 m	6200	8.4 °C	/
Genetic-optimised aperiodic coding ^[29]	39.0 km	1.0 m	39000	3.9 °C	13.6 min
Derived Sequences coding ^[33]	44.0 km	5.0 m	8800	2.5 °C	/
EAC-coding	70.0 km	1.58 m	44303	5.39 °C	252 min

Sensing Points refers to the number of independent spatial sampling points in the fiber optic sensing link where the system can achieve effective measurements. The specific calculation method is the ratio of the sensing distance to the spatial resolution.
Temperature resolution is defined as the minimum temperature change that a system can reliably distinguish, and its calculation method is based on the standard deviation under steady-state conditions.

We have compared our proposed scheme with other coding methods such as simplex coding and Genetic-optimised aperiodic coding in terms of coding gain, measurement time, and system complexity. The specific revisions are as follows.

In terms of coding gain, the EAC-coding scheme proposed in this paper exhibits significant advantages in SNR improvement. The core of this scheme lies in the analysis and accurate reconstruction of the waveform of Raman backscattered signals, which effectively compensates for the detection signals distortion caused by the transient effects of the system. This mechanism enables EAC-coding to overcome the performance bottleneck of traditional derived sequences coding schemes induced by signals distortion, achieving a coding gain

closer to the theoretical limit. Specifically, existing derived sequence coding scheme construct coding sequences based on Golay complementary sequences and perform correlation decoding using their derived sequences. The upper limit of their theoretical coding gain is $\sqrt{L}/2$ (where L denotes the length of the coding sequence). However, due to the unavoidable distortion (especially transient distortion) in actual detection signals, the results of their correlation operations deteriorate, leading to a significant gap between the actually achieved coding gain and this theoretical value. In contrast, the EAC-coding scheme, through the aforementioned signals analysis and reconstruction mechanism, significantly suppresses the impact of transient distortion on the detection signals waveform, thereby enabling its coding gain to approach the theoretical upper limit of $\sqrt{L}/2$. Benefiting from this, the proposed system ultimately achieves temperature sensing capability in a long-distance Raman distributed optical fiber sensing system with a sensing distance of 70.0 km.

In terms of measurement time, the core advantage of the EAC-coding scheme stems from its extremely low decoding computational complexity. This scheme achieves signals reconstruction solely through Fourier transform, which significantly reduces the computational cost and time delay of real-time data processing, thereby effectively improving the overall response speed of the system. Specifically, for a Simplex code with length L , it is necessary to inject L groups of detection signals into the optical fiber and collect L groups of Raman backscattered signals, with its theoretical minimum acquisition time being $T_{Simplex}=L \times t$ (where t denotes the acquisition time of one group of Raman backscattered curves). Although the Genetic-optimised aperiodic coding scheme only requires injecting one group of detection signals into the optical fiber, it needs additional time to design the coding sequence based on the distributed genetic algorithm. The Derived sequences coding scheme requires injecting 4 groups of detection signals into the optical fiber, with a theoretical acquisition time of $4t$. However, in the decoding stage, it involves complex steps such as extracting the attenuation envelope, establishing a compensation model, and solving the compensation envelope via matrix operations, which increases the time cost of the decoding process. The EAC-coding scheme proposed in this paper performs encoding based on Golay complementary sequences, with a theoretical acquisition time of $4t$, and only needs Fourier transform to complete the analysis and reconstruction of the Raman backscattered signals waveform. This significant efficiency improvement at the algorithm level greatly reduces the real-time data processing delay and effectively enhances the system response

speed, providing crucial real-time guarantee for the final realization of temperature sensing in a 70.0 km long-distance Raman distributed optical fiber sensing system.

In terms of system complexity, the core innovation of the EAC-coding scheme lies in achieving a high degree of simplification in the decoding process through algorithm optimization, while keeping the hardware architecture unchanged. This scheme only needs to perform efficient Fourier transform operations to complete signal reconstruction, which significantly reduces the algorithmic complexity and computational burden of real-time data processing. The specific reasons are as follows. Simplex coding scheme requires the injection of L groups of detection signals, resulting in a long acquisition time and a large volume of raw data. This not only imposes high requirements on hardware performance (such as data acquisition rate and storage capacity) and long-term stability, but also the large-scale data processing involved in its decoding process significantly increases the computational complexity. Genetic-optimised aperiodic coding scheme greatly shortens the acquisition time without increasing hardware costs. However, due to its unique design, it is necessary to design coding sequences based on a distributed genetic algorithm and conduct extensive tests to find the optimal coding sequence. Derived sequences coding scheme also does not require additional hardware overhead. However, this scheme introduces high computational complexity in the decoding stage, as it relies on computationally expensive matrix operations to generate and process derived sequences.

EAC-coding scheme proposed in this paper incorporates a novel preprocessing framework for autocorrelation characteristic analysis, waveform analysis and reconstruction of Raman backscattered signals. This framework significantly improves the fidelity and transmission stability of the encoded pulse sequence by dynamically compensating for the transient gain fluctuation of EDFA. This technology leverages the inherent gain of EAC-coding pulse coding to effectively offset fiber scattering attenuation during 70.0 km long-distance transmission, achieving a significant improvement in the baseline SNR. On this basis, a two-stage noise suppression method is further constructed. The first-stage processing involves compensating for signal attenuation using the theoretical coding gain of EAC-coding. For the second-stage processing, a Haar wavelet denoising algorithm is adopted. Its step-matching characteristic enables accurate extraction of the abrupt temperature change features of the sensing fiber and

adaptive filtering of the residual time/frequency domain coupled noise in the decoded signal. This three-fold synergistic mechanism, consisting of the preprocessing framework compensating for EDFA fluctuation, coding gain suppressing transmission losses, and wavelet transform finely removing residual noise, breaks through the theoretical trade-off bottleneck between SNR and sensing distance in traditional schemes. Experiment results indicate that, at an ultra-long distance of 70.0 km, this scheme can achieve a spatial resolution of 1.58 m, a temperature resolution of 5.39 °C, and an effective number of sensing points reaching 44,303, verifying the engineering feasibility of the scheme. To the best of our knowledge, the number of effective sensing points achieved by this scheme ranks the highest in the field of Raman distributed optical fiber sensing systems.”

24) In the section of **Experimental setup, page 9, left column, paragraph 2, line 21, add** “Furthermore, the measurement time of the scheme proposed in this paper is mainly limited by the massive data volume (with the effective number of sensing points reaching 44,303) and the hardware accumulation processing time of the acquisition card, which are caused by its long sensing distance and high spatial resolution—rather than the complexity of its processing architecture. The current measurement time (252 min) is constrained by the processing capability of the data acquisition card (acquisition rate: 1.0 GSa/s; storage depth: 512 Mpts). The massive data volume (approximately 700 k points per frame corresponding to a 70.0 km sensing distance) exceeds the on-board real-time processing limit. In the future, high-performance acquisition cards will be upgraded to optimize the data processing pipeline, thereby eliminating the delay deterioration caused by hardware bottlenecks.”

25) In the section of **Discussion, page 21, right column, paragraph 3, line 35, add** “In this proposed scheme, the physical limitations on the optimization of sensing distance and SNR mainly stem from two aspects: the insufficient performance of experimental devices and the physical demodulation principle.

In terms of experimental devices, the maximum coding bit length of the digital delayed pulse generator (DDPG) and the sampling rate of the data acquisition card are key factors restricting the improvement of the sensing distance and SNR of this scheme. The CIQTEK ASG8100 model DDPG used in the experiment has a maximum coding bit length of 99 bits. Since the bit length of Golay complementary sequences is typically 2^n , a coding length of 64 bits was selected in the experiment. The coding length directly determines the coding gain

of the system, which in turn affects the extent of improvement in sensing performance. Therefore, it is planned to adopt a DDPG with a higher maximum coding bit length (e.g., ≥ 128 bits) in the future to increase the system's coding length, thereby achieving better sensing distance performance. In addition, the currently used data acquisition card has a sampling rate of 1.0 GSa/s and a maximum number of sampling points of 700,000, corresponding to a theoretical maximum sensing distance of approximately 70.0 km. Future plans involve upgrading to a data acquisition card with a higher sampling rate (e.g., ≥ 2 GSa/s) and a larger number of sampling points (e.g., $\geq 1,000,000$) to eliminate the limitations imposed by hardware devices on the sensing distance.

In terms of the physical demodulation principle, the stimulated scattering threshold and fiber dispersion effect also affect the improvement of sensing distance and SNR in this scheme. Firstly, the stimulated scattering threshold of single-mode fibers is relatively low, which limits the maximum optical power injected into the fiber. As a result, the intensity of the Raman backscattered signal excited in the fiber is relatively weak, leading to a low SNR. In the future, we will attempt to use few-mode fibers to increase the optical power injected into the fiber while minimizing intermodal dispersion as much as possible. Finally, fiber dispersion introduced by long-distance transmission causes pulse broadening and intersymbol interference, which destroys the orthogonality of sequences. This results in the broadening of demodulated correlation peaks and a reduction in their amplitude, impairing both SNR and distance resolution. This issue can be compensated for by introducing dispersion-compensating fibers.”

- 26) In the section of **Experimental results, page 13, left column, paragraph 2, line 26**, **add** “The Haar wavelet function generates a set of orthogonal basis functions through translation and scaling, enabling multi-scale decomposition of signals. Its inherent advantage in processing low-complexity signals with abrupt change characteristics makes it particularly suitable for the analysis of Raman backscattered signals.

In the experiment, to further improve the SNR of Raman backscattered signals, we adopted a 6-level decomposition Haar wavelet. Furthermore, in the post-processing stage of temperature demodulation, to further suppress noise and enhance signal smoothness, we additionally applied the db10 (Daubechies wavelet 10) wavelet function with 4-level decomposition. It should be noted that while this step improves signal quality (by reducing temperature fluctuation), it

moderately degrades the spatial resolution of the system. Such parameter adjustment reflects the necessary trade-off between spatial resolution and temperature measurement accuracy.”

- 27) In the section of **Experimental results, page 15, left column, paragraph 2, line 28, add** “Temperature fluctuation is expressed as the difference between the measured temperature and the ambient temperature. Specifically, in a Raman distributed fiber optic sensing system, there exists a discrepancy between the demodulated temperature (T') at the sampling points of the sensing fiber and the ambient reference temperature (T_0). This discrepancy is defined as temperature fluctuation ($\Delta T=T'-T_0$).”
- 28) In the section of **Experimental results, page 16, left column, paragraph 3, line 29, add** “Temperature resolution refers to temperature uncertainty, which characterizes the system's ability to distinguish temperature changes, is generally defined as the standard deviation (σ) of measured temperatures under constant temperature conditions (e.g., a section of reference fiber) ^[34-35].”
- 29) In the section of **Experimental results, page 13, left column, paragraph 1, line 10, add** “(with Haar wavelet).”
- 30) In the section of **Experimental results, page 11, left column, change** “Fig. 4” to “Fig. 4”.

Fig. 4. Intensity of Raman backscattered signals and SNR. (a1) Intensity of Raman backscattered signal generated by detection signal u before and after pre-processing. (a2) Intensity of Raman backscattered signal generated by detection signal \bar{u} before and after pre-processing. (a3) Intensity of Raman backscattered signal generated by detection signal w before and after pre-processing. (a4) Intensity of Raman backscattered signal generated by detection signal \bar{w} before and after pre-processing. (b1) Intensity of Raman backscattered signals decoded by distorted detection signals before and after pre-processing. (b2) SNR of Raman backscattered signals decoded by detection signals before and after pre-processing. (c1) Autocorrelation function of the detection signals before pre-processing. (c2) Autocorrelation function of the distorted detection signals after pre-processing.

31) In the section of **Experimental results**, page 12, left column, paragraph 2, line 3, add “We have added an autocorrelation processing experiment on the detection signals coupled into the sensing fiber after reconstruction, aiming to further verify that it eliminates the transient effects of the system. The details of the newly added experiment are as follows. The primary objective of the newly added experiment is to verify that the reconstructed Raman backscattered signals

is equivalent to the Raman backscattered signal of the undistorted detection signals.

Taking the detection signal u as an example, its attenuation envelope function can be expressed by Eq. (8). Thus, the detection signal coupled into the sensing fiber after reconstruction can be calculated and is given by Eq. (17).

$$u' = \text{iff}t\left(\frac{\text{fft}[u(f)]}{\text{fft}(f')}\right) \quad (17)$$

Then, we performed the same processing on the four detection signals and conducted autocorrelation operations on the processed detection signals. If Eq. (18) is satisfied, it can be verified that their autocorrelation characteristics are restored after reconstruction, with the transient effects effectively suppressed.

$$(u' - \bar{u}') * (u' - \bar{u}') + (w' - \bar{w}') * (w' - \bar{w}') = 2l\delta \quad (18)$$

The experimental results are shown in Fig. 4. Fig. 4 (c1) presents the autocorrelation function of the detection signals with waveform distortion caused by transient effects, while Fig. 4(c2) shows the autocorrelation function of the equivalent detection signals coupled into the sensing fiber after reconstruction. It can be clearly observed from Fig. 4(c2) that the sidelobe noise of its autocorrelation function is significantly reduced, which verifies that its autocorrelation characteristics are restored and the transient effects are effectively suppressed.”

- 32) In the **(unmodified manuscript) section of Experimental results, page 9, right column, paragraph 2, line 15, delete** “The SNR measurement result after decoding is shown in Fig. 6(a). Compared with the conventional single-pulse scheme, the SNR at 70.0 km of sensing fiber is improved by 8.75 dB based on proposed EAC-coding scheme, which validates the superiority of the proposed EAC-coding scheme in SNR improvement.”

The temperature demodulation result is shown in Fig. 6(b). Experimental measurement demonstrates that the proposed EAC-coding scheme achieves significant suppression of temperature fluctuation under ambient conditions, reducing the temperature uncertainty from ± 300 °C maximum in the conventional single-pulse scheme to ± 15.84 °C.

Furthermore, we conduct additional quantitative analyses by evaluating the temperature resolution of the system.

The temperature resolution based on the conventional single-pulse scheme and proposed EAC-coding scheme are shown in Fig. 7(a). At the sensing distance

of 70.0 km, the conventional single-pulse scheme achieves a temperature resolution of 29.25 °C. In contrast, the proposed EAC-coding scheme demonstrates significant performance enhancement by effectively improving the SNR, which substantially reduces system temperature fluctuation, resulting in a remarkable improvement of temperature resolution to 5.39 °C at the same sensing distance. Meanwhile, to validate the spatial resolution of the proposed EAC-coding scheme, a 40-meter-long fiber under test is positioned at the distance of 70.0 km and immersed in a thermostatically controlled water bath maintained at 90 °C, while the remaining portion of the sensing fiber is exposed to the room temperature of 20 °C.”

- 33) In the section of **Experimental results**, page 15, left column, paragraph 2, line 5, add “The results of temperature demodulation and SNR are presented in Fig. 6. The temperature fluctuation of the traditional single-pulse scheme can be as high as approximately 300 °C, as shown in Fig. 6(a1). In this paper, first, by leveraging the synergistic demodulation mechanism integrating EAC-coding (which compensates for attenuation via coding gain) and the refined processing of Haar wavelet denoising, effective suppression of temperature fluctuation noise along the optical fiber is achieved. This significantly reduces the system’s temperature fluctuation to 22.51 °C while improving the SNR by 7.24 dB, as shown in Fig. 6(a2) and Fig. 6(b2). Furthermore, the preprocessing architecture within EAC-coding (autocorrelation characteristic analysis, waveform analysis and reconstruction of Raman backscattered signals) further reduces the system’s temperature fluctuation to 20.86 °C by dynamically compensating for EDFA transient gain fluctuation and reconstructing Raman scattered signals. Meanwhile, the SNR is further improved by an additional 1.48 dB, as shown in Fig. 6(a3) and Fig. 6(b3).

Compared with the temperature resolution of 29.25 °C achieved by the traditional single-pulse scheme, the EAC-coding scheme (integrating coding gain with Haar wavelet denoising) improves the system’s temperature resolution to 5.82 °C, as shown in Fig. 6(c2). Meanwhile, the preprocessing architecture within EAC-coding further enhances the system’s temperature resolution to 5.39 °C by dynamically compensating for EDFA transient gain fluctuation and reconstructing Raman scattered signals, as shown in Fig. 6(c3).

Meanwhile, we further compared the temperature measurement accuracy performance of the system, as shown in Fig. 6(d1) - (d3). It can be observed that

the transient effects not only affect the SNR, temperature fluctuation, and temperature resolution but also exert a significant impact on the system's temperature measurement accuracy. It can be observed that as the temperature of the fiber under test (FUT) gradually increases, the temperature measurement error of the distorted signal demodulation results becomes increasingly large. This confirms that the influence of the transient effect also intensifies with rising temperature—with the temperature measurement accuracy deteriorating to a maximum of 14.98 °C at 90.0 °C (where the demodulation result is 104.98 °C). This is because with increasing temperature, the elevation of the Raman backscattered curve at the FUT becomes more pronounced, while transient effects exert an increasingly significant influence on signal intensity. After the preprocessing architecture in the EAC-coding scheme (autocorrelation characteristic analysis, waveform analysis and reconstruction of Raman backscattered signals), the temperature demodulation result is 90.91 °C, and the temperature measurement accuracy is 0.91 °C, representing an improvement of approximately 16.46 times. Therefore, the EAC-coding scheme proposed in this paper can effectively improve the system's temperature measurement accuracy, enhancing the temperature measurement accuracy of the long-distance Raman distributed optical fiber sensing system (with a sensing distance of 70.0 km) to within 1.0 °C.

Through the synergistic demodulation mechanism of EAC-coding (which utilizes coding gain to compensate for attenuation) combined with the refined processing of Haar wavelet denoising, the theoretical trade-off bottleneck between SNR and sensing distance in traditional schemes is broken through. Experimental results demonstrate that at an ultra-long distance of 70.0 km, the system achieves a spatial resolution of 1.58 m and a temperature resolution of 5.39 °C. Moreover, the preprocessing architecture in EAC-coding (integrating autocorrelation characteristic analysis of Raman backscattered signals and waveform analysis and reconstruction) significantly enhances the temporal fidelity and transmission stability of coded pulse sequences by dynamically compensating for EDFA transient gain fluctuation. This further improves the system's SNR and temperature measurement accuracy, enabling the temperature measurement accuracy to reach 0.91 °C. In this study, the improvements in SNR (accompanied by performance enhancements in sensing distance and temperature resolution) are primarily attributed to the proposed EAC-coding (for coding gain

compensation) and Haar wavelet denoising schemes. In contrast, the preprocessing architecture within EAC-coding (characterized by autocorrelation characteristic analysis, waveform analysis and reconstruction of Raman backscattered signals) serves to improve the system's temperature measurement accuracy by dynamically compensating for EDFA transient gain fluctuation and reconstructing Raman scattered signals.”

34) In the section of **Principles and methods**, page 5, change “Fig. 1” to “Fig. 1”.

Fig. 1. Physics principle of proposed EAC-coding scheme for Raman distributed optical fiber sensing. (a) Principle of encoding based on Golay complementary sequences. (b) The influence of transient effects on Raman backscattered signals. (c) RBSs generated by four distorted detection signals in sensing fiber; (d) Principle of decoding after analysis and waveform-reconstruction based on proposed EAC-coding scheme. (e) Temperature demodulation principle.

35) In the section of **Principles and methods**, page 5, left column, paragraph 3, line 18, change “destroyed detection signals” to “distorted detection signals”.

36) In the section of **Experimental results**, page 11, right column, paragraph 2, line 20, change “destroyed detection signals” to “distorted detection signals”.

37) In the section of **Experimental results**, page 11, right column, paragraph 2, line 27, change “destroyed detection signals” to “distorted detection signals”.

38) In the section of **Discussion**, page 22, right column, paragraph 3, line 18, add “For the coding technology with a code length of 64 bits (i.e., the EAC-coding scheme adopted in this paper). Under the maximum input power achievable by a single pulse demodulation system (0.2 mW), the theoretical SNR improvement is 6 dB.”

However, the theoretical 6 dB SNR improvement is based on the strict premise that the input powers of the single-pulse demodulation scheme and the

pulse coding scheme are the same. In this experiment, to explore the extreme performance of the EAC-coding scheme and maximize the system SNR, we set the input power to be close to the maximum value of the stimulated Raman scattering (SRS) nonlinear effect threshold of the sensing fiber. Specifically, the average power input into the optical fiber is approximately 0.7 mW, and the calculated peak power is close to 1.4 W.

In contrast, the traditional single-pulse demodulation scheme, due to its higher peak power (under the same average power), is more likely to trigger the SRS nonlinear effect, leading to signal distortion and performance degradation. Therefore, to ensure the stable operation of the single-pulse scheme below the nonlinear threshold, its average input power must be significantly reduced, which is set to approximately 0.2 mW in the experiment.

The different incident powers of the two schemes are incorporated into the calculation of the theoretical SNR improvement. According to Eq. (4) in the paper, with all other factors remaining completely unchanged, the intensity of the Raman backscattered signal depends solely on the incident optical power. Thus, the modified theoretical SNR improvement is approximately $10\log_{10}[(\sqrt{64/2}) \times (0.7/0.2)] = 11.46$ dB. Since the EDFA introduces a significant amount of amplified spontaneous emission (ASE) noise while amplifying the effective signal, the final experimental improvement in SNR is 8.72 dB, which is slightly lower than the theoretical value of 11.46 dB.

In summary, it is precisely because the EAC-coding scheme allows a higher average input power without triggering significant nonlinear effects, and combined with its inherent theoretical coding gain, the SNR improvement observed in the experiment finally reaches 8.72 dB, which is slightly lower than the theoretical SNR coefficient (11.46 dB) at this power setting (0.7 mW).”

- 39) In the section of **Discussion, page 15, right column, paragraph 2, line 5**, add “In the experiment, within a range from the starting end of the sensing fiber to 56.0 km, the SNR improvement of the sensing signal is about 10.44 dB; while in the region beyond approximately 70.0 km at the tail end of the optical fiber, the SNR improvement is 8.72 dB. The improvement of SNR at each sensing distance point is lower than the theoretical value of 11.46 dB. The improvement of SNR at the end of sensing fiber is also lower than that at the front end of sensing fiber. This discrepancy primarily stems from the synergistic effect between the difference in SNR improvement between experiments and theory conditions, and the distance-dependent characteristics of fiber attenuation.

After revising the SNR calculation formula, the overall SNR improvement along the optical fiber obtained from the experimental results is slightly lower than the theoretical SNR. Since the maximum input optical power of the sensing fiber affects the theoretical SNR improvement of its coding scheme, the final SNR magnitude calculated for the EAC-coding scheme is 11.46 dB. Meanwhile, due to the introduction of the EDFA, a large amount of amplified spontaneous emission (ASE) noise is introduced during the process of amplifying the detection signal—this results SNR improvement in the experiment being slightly lower than the theoretical value of 11.46 dB. Consequently, the SNR improvement along the optical fiber derived from the experimental results (with a maximum of 10.44 dB) is slightly lower overall than the theoretical SNR (11.46 dB).

Furthermore, fiber attenuation exhibits a cumulative effect with transmission distance. At 70.0 km, the single-pulse signal is already close to the system's detection limit due to severe attenuation, making it difficult to extract effectively. Although the EAC-coding scheme still maintains a certain signal advantage, the intensity difference between the EAC-coding signal and the single-pulse signal is smaller than that at 28.0 km, as the signal amplitude decreases significantly after long-distance transmission. Consequently, the SNR improvement at this location decreases to 8.72 dB. This result clearly reflects the distance-dependent characteristic of fiber signal attenuation and also highlights the performance difference of the coding scheme at different transmission distances.”

- 40) In the section of **Experimental results, page 16, right column, paragraph 3, line 37, add** “In this experiment, temperature resolution is characterized by the temperature standard deviation of each distributed temperature measurement point within a fixed length of the sensing fiber. Experimental results show that at a sensing distance of 70.0 km, the temperature resolution of the EAC-coding scheme is significantly improved compared with that of the single-pulse scheme: the temperature resolution is enhanced from 29.25 °C (of the single-pulse scheme) to 5.39 °C, representing an improvement of up to 5.43 times.”
- 41) In the **(unmodified manuscript) section of References, page 12, right column, delete** “[21] M. A. Soto, *et al.* Raman-based distributed temperature sensor with 1 m spatial resolution over 26 km SMF using low-repetition-rate cyclic pulse coding. *Opt. Lett.* **36**(13), 2557-2559 (2011).

[22] G. D. B. Vazquez, *et al.* Distributed temperature sensing using cyclic pseudorandom sequences. *IEEE Sens. J.* **17**(6), 1686-1691 (2017).

[23] X. Z. Sun, *et al.* Genetic-optimised aperiodic code for distributed optical fibre sensors. *Nat. Commun.* **11**(1), 5774 (2020).

[24] Z. L. Wang, *et al.* Application of wavelet transform modulus maxima in Raman distributed temperature sensors. *Photonic Sens.* **4**, 142-146(2014).

[25] Z. L. Wang, *et al.* An improved denoising method in RDTS based on wavelet transform modulus maxima. *IEEE Sens. J.* **15**(2), 1061-1067 (2015).

[26] M. Yu, *et al.* Ambient condition desensitization of a fiber Raman temperature sensing system based on a dynamic sampling-correction scheme. *Appl. Opt.* **54**(15), 4823-4827(2015).

[27] H. S. Pradhan, *et al.* Characterisation of Raman distributed temperature sensor using deconvolution algorithms. *IET Optoelectron.* **9**(2), 101-107(2015).

[28] H. S. Guo, *et al.* High Precision Raman Distributed Fiber Sensing Using Residual Composite Dual-Convolutional Neural Network. *J. Lightwave Technol.* **42**(10), 3918-3928 (2024)

[29] Z. Zhang, *et al.* High-performance Raman distributed temperature sensing powered by deep learning. *J. Lightwave Technol.* **39**(2), 654-659(2020).

[30] Y. Liu, *et al.* Long-range Raman distributed temperature sensor with high spatial and temperature resolution using graded-index few-mode fiber. *Opt. Express.* **26**(16), 20562-20571 (2018).

[31] M. Wang, *et al.* Few-mode fiber-based Raman distributed temperature sensing. *Opt. Express.* **25**(5), 4907-4916 (2017).

[32] H. Wu, *et al.* 24 km High-performance Raman distributed temperature sensing using low water peak fiber and optimized denoising neural network. *Sensors.* **22**(6), 2139 (2022)”

42) In the **section of References, page 24, right column, add “[21] Z. L. Wang, *et al.* Application of wavelet transform modulus maxima in Raman distributed temperature sensors. *Photonic Sens.* **4**, 142-146 (2014).**

[22] Z. L. Wang, *et al.* An improved denoising method in RDTS based on wavelet transform modulus maxima. *IEEE Sens. J.* **15**(2), 1061-1067 (2015).

[23] M. Yu, *et al.* Ambient condition desensitization of a fiber Raman temperature sensing system based on a dynamic sampling-correction scheme. *Appl. Opt.* **54**(15), 4823-4827 (2015).

[24] H. S. Pradhan, *et al.* Characterisation of Raman distributed temperature sensor using deconvolution algorithms. *IET Optoelectron.* **9**(2), 101-107 (2015).

[25] H. S. Guo, *et al.* High Precision Raman Distributed Fiber Sensing

Using Residual Composite Dual-Convolutional Neural Network. *J. Lightwave Technol.* **42**(10), 3918-3928 (2024)

[26] Z. Zhang, *et al.* High-performance Raman distributed temperature sensing powered by deep learning. *J. Lightwave Technol.* **39**(2), 654-659 (2020).

[27] M. A. Soto, *et al.* Raman-based distributed temperature sensor with 1 m spatial resolution over 26 km SMF using low-repetition-rate cyclic pulse coding. *Opt. Lett.* **36**(13), 2557-2559 (2011).

[28] G. D. B. Vazquez, *et al.* Distributed temperature sensing using cyclic pseudorandom sequences. *IEEE Sens. J.* **17**(6), 1686-1691 (2017).

[29] X. Z. Sun, *et al.* Genetic-optimised aperiodic code for distributed optical fibre sensors. *Nat. Commun.* **11**(1), 5774 (2020).

[30] L. Xu, *et al.* RDTS noise reduction method based on ICEEMDAN-FE-WSTD. *IEEE Sens. J.* **22**(18), 17854-17863 (2022).

[31] J. Park, *et al.* Raman-based distributed temperature sensor with simplex coding and link optimization. *IEEE Photonics Technol. Lett.* **18**(17), 1879-1881 (2006).

[32] B. Rosolem, *et al.* Raman DTS based on OTDR improved by using gain-controlled EDFA and pre-shaped simplex code. *IEEE Sens. J.* **17**(11), 3346-3353 (2017).

[33] D. D. Chai, *et al.* Derived sequences decoding approach for long-range distributed temperature sensors. *IEEE Sens. J.* **23**(3), 2204-2210 (2022).

[34] X. Liu, *et al.* High-speed and high-resolution YAG fiber based distributed high temperature sensing system empowered by a 2D image restoration algorithm. *Opt. Express.* **31**(4), 6170-6183 (2023).

[35] J. Gasser, *et al.* Distributed temperature sensor combining centimeter resolution with hundreds of meters sensing range. *Opt. Express.* **30**(5), 6768-6777 (2022).”

Again, we are grateful for the insightful suggestions from the reviewers.

Sincerely Yours,

Jian Li, Mingjiang Zhang

Response to the reviewers' comments

Thank you very much for your consideration of our manuscript entitled “70 km Long-Range Raman Distributed Optical Fibre Sensing Through Enhanced Anti-distortion Coding and Waveform Reconstruction” by Fan Zhang, Jian Li, Lulei Li, Kangyi Cao and Mingjiang Zhang.

We would like to thank the reviewers for their efforts to review our manuscript and their appreciation of the novelty and quality of our work. The constructive suggestions from the reviewers have been thoroughly considered and implemented in the revised manuscript, which now has been significantly improved.

In the next several pages, the changes made to the manuscript are detailed to fully address the concerns raised according to each of the reviewers' comments. We hope the manuscript is now appropriate for publication in *Nature Communications*.

In addition, we would like to declare:

- (1) All authors agree with the submission.
- (2) The work has not been published or submitted for publication elsewhere, either completely or in part, or in another form or language.
- (3) No materials are reproduced from another source.
- (4) The authors declare no conflicts of interest.

We look forward to hearing from you!

With best regards,

Jian Li, Mingjiang Zhang

All the reviewers' suggestions have been considered. The following are the changes made in the manuscript to fully address the reviewers' concerns:

Blue: Original comments from the reviewer.

Black: Our response including action taken.

Responds to the reviewers' comments:

Reviewer #1 (Comments to the Author):

Recommendations for the Author(s):

My comments have been well addressed, and all the questions have been answered in detail. The current manuscript can be accepted for publication.

Reply: We sincerely appreciate your careful and thorough review and positive feedback. Your detailed questions and constructive guidance have been invaluable in refining the manuscript. We feel greatly honored and encouraged to receive the "accept for publication" decision. Once again, we would like to extend our sincere gratitude to you.

Responds to the reviewers' comments:

Reviewer #2 (Comments to the Author):

Recommendations for the Author(s):

I can see that the authors have been trying very hard to carefully address all the comments that the reviewers raised. While all the comments have been comprehensively discussed, the current manuscript is way too lengthy, especially for the introduction and the discussion parts. Much of the content should be much more concise or moved to the supplementary material.

Reply: We sincerely appreciate your careful review and constructive feedback on our manuscript. Your recognition of our earnest efforts to address all the previous comments is truly encouraging, and we are grateful for your thoughtful insights that help refine the work further.

Following your valuable suggestions, we have substantially streamlined redundant content and relocated relevant materials to the Supplementary Information, ensuring that the core information of the study remains concise and focused while retaining all critical information.

Thank you for your suggestion. We have made the following modifications in our revised manuscript:

- 1) In the section of **Introduction**, the following content has been relocated to the **Supplementary Information**. “Simplex codingscheme, as a linear coding technique based on Hadamard matrix transformation, has its code length (in L bits) determining the number of detection signals required (L groups). For instance, in 2006, J. Park, et al. achieved a spatial resolution of 17.0 m, a temperature resolution of 3.0 °C, and a total of 2176 effective sensing points over a sensing distance of 37.0 km by utilizing this coding scheme and link optimization techniques. The main limitation of this scheme lies in the linear growth of data acquisition volume and computational complexity with code length, which restricts its practical efficiency in applications requiring long code lengths and high-speed sensing.

Low-repetition-rate cyclic pulse coding scheme generates Simplex coding-based cyclic coding sequences using an acousto-optic modulator (AOM), enabling pulses to be injected into the optical fiber periodically at a low repetition rate. For example, in 2011, M. A. Soto, et al. combined the low-repetition-rate quasi-periodic cyclic

pulse coding with a high-power fiber laser, achieving a spatial resolution of 1.0 m, a temperature resolution of 3.0 °C, and 26,000 effective sensing points over a sensing distance of 26.0 km. However, higher pulse energy also makes it more likely to induce fiber nonlinear effects, and when approaching the stimulated scattering threshold, it restricts the system's sensing distance and SNR performance.

Pre-Shaped Simplex Coding scheme is based on simplex coding technology and adopts a linearly increasing profile to adjust pulse amplitude, compensating for the amplitude variation caused by EDFA to suppress the transient effect of the EDFA and improve system performance. For example, in 2017, J. B. Rosolem, et al. achieved a spatial resolution of 10.0 m, a temperature resolution of 8.4 °C, and 6,200 effective sensing points over a sensing distance of 62.0 km by using the pre-shaped Simplex coding and a gain-controlled erbium-doped fiber amplifier. Although this scheme effectively suppresses the transient response of the EDFA at the hardware level, its hardware structure is more complex, leading to a significant increase in implementation cost.

Genetic-optimised aperiodic coding scheme utilizes a distributed genetic algorithm (DGA) to generate a single-sequence aperiodic code (GO-code), which is then converted into an optical pulse sequence and injected into the optical fiber. For example, in 2020, X. Z. Sun, et al. achieved a spatial resolution of 1.0 m, a temperature resolution of 3.9 °C, and 39,000 effective sensing points over a sensing distance of 39.0 km based on the Genetic-optimised aperiodic coding scheme. The core challenge of this scheme lies in its high dependence on optimization algorithms to search for the optimal code, which typically requires substantial computational resources and iterative testing.

Derived Sequences coding scheme addresses the transient effects of EDFA through a derived sequence decoding method, thereby enhancing the performance of long-distance sensing. For instance, in 2022, D. D. Chai, et al. achieved a spatial resolution of 5.0 m, a temperature resolution of 2.5 °C, and 8,800 effective sensing points over a sensing distance of 44.0 km based on the Derived Sequences coding scheme. However, this scheme's decoding process imposes a heavy computational burden, and the core algorithm

involves high-complexity matrix operations, making it difficult to meet real-time requirements.”

2) In the section of **Discussion**, the following content has been relocated to the **Supplementary Information**. “We have compared our proposed scheme with other coding methods such as simplex coding and Genetic-optimised aperiodic coding in terms of coding gain, measurement time, and system complexity. The specific revisions are as follows.

Table 2. Sensing performance of various coding schemes.

Scheme	Sensing Distance	Spatial Resolution	Sensing Points	Temperature Resolution	Measurement Time
Simplex coding ^[31]	37.0 km	17.0 m	2176	3.0 °C	/
Low-repetition-rate cyclic pulse coding ^[27]	26.0 km	1.0 m	26000	3.0 °C	30 s
Pre-Shaped Simplex Coding ^[32]	62.0 km	10.0 m	6200	8.4 °C	/
Genetic-optimised aperiodic coding ^[29]	39.0 km	1.0 m	39000	3.9 °C	13.6 min
Derived Sequences coding ^[33]	44.0 km	5.0 m	8800	2.5 °C	/
EAC-coding	70.0 km	1.58 m	44303	5.39 °C	252 min

Sensing Points refers to the number of independent spatial sampling points in the fiber optic sensing link where the system can achieve effective measurements. The specific calculation method is the ratio of the sensing distance to the spatial resolution.
Temperature resolution is defined as the minimum temperature change that a system can reliably distinguish, and its calculation method is based on the standard deviation under steady-state conditions.

In terms of coding gain, the EAC-coding scheme proposed in this paper exhibits significant advantages in SNR improvement. The core of this scheme lies in the analysis and accurate reconstruction of the waveform of Raman backscattered signals, which effectively compensates for the detection signals distortion caused by the transient effects of the system. This mechanism enables EAC-coding to overcome the performance bottleneck of traditional derived sequences coding schemes induced by signals distortion, achieving a coding gain closer to the theoretical limit. Specifically, existing derived sequence coding scheme construct coding sequences based on Golay complementary sequences and perform correlation decoding using their derived sequences. The upper limit of their theoretical coding gain is $\sqrt{L}/2$ (where L denotes the length of the coding sequence). However, due to the unavoidable distortion (especially transient distortion) in actual detection signals, the results of their correlation operations deteriorate, leading to a significant gap between the actually achieved coding gain and this theoretical value. In contrast, the EAC-coding scheme, through the aforementioned signals analysis and reconstruction mechanism, significantly suppresses the impact of transient distortion on the detection signals

waveform, thereby enabling its coding gain to approach the theoretical upper limit of $\sqrt{L}/2$. Benefiting from this, the proposed system ultimately achieves temperature sensing capability in a long-distance Raman distributed optical fiber sensing system with a sensing distance of 70.0 km.

In terms of measurement time, the core advantage of the EAC-coding scheme stems from its extremely low decoding computational complexity. This scheme achieves signals reconstruction solely through Fourier transform, which significantly reduces the computational cost and time delay of real-time data processing, thereby effectively improving the overall response speed of the system. Specifically, for a Simplex code with length L, it is necessary to inject L groups of detection signals into the optical fiber and collect L groups of Raman backscattered signals, with its theoretical minimum acquisition time being $T_{\text{Simplex}}=L \times t$ (where t denotes the acquisition time of one group of Raman backscattered curves). Although the Genetic-optimised aperiodic coding scheme only requires injecting one group of detection signals into the optical fiber, it needs additional time to design the coding sequence based on the distributed genetic algorithm. The Derived sequences coding scheme requires injecting 4 groups of detection signals into the optical fiber, with a theoretical acquisition time of $4t$. However, in the decoding stage, it involves complex steps such as extracting the attenuation envelope, establishing a compensation model, and solving the compensation envelope via matrix operations, which increases the time cost of the decoding process. The EAC-coding scheme proposed in this paper performs encoding based on Golay complementary sequences, with a theoretical acquisition time of $4t$, and only needs Fourier transform to complete the analysis and reconstruction of the Raman backscattered signals waveform. This significant efficiency improvement at the algorithm level greatly reduces the real-time data processing delay and effectively enhances the system response speed, providing crucial real-time guarantee for the final realization of temperature sensing in a 70.0 km long-distance Raman distributed optical fiber sensing system.

In terms of system complexity, the core innovation of the EAC-coding scheme lies in achieving a high degree of simplification in the

decoding process through algorithm optimization, while keeping the hardware architecture unchanged. This scheme only needs to perform efficient Fourier transform operations to complete signal reconstruction, which significantly reduces the algorithmic complexity and computational burden of real-time data processing. The specific reasons are as follows. Simplex coding scheme requires the injection of L groups of detection signals, resulting in a long acquisition time and a large volume of raw data. This not only imposes high requirements on hardware performance (such as data acquisition rate and storage capacity) and long-term stability, but also the large-scale data processing involved in its decoding process significantly increases the computational complexity. Genetic-optimised aperiodic coding scheme greatly shortens the acquisition time without increasing hardware costs. However, due to its unique design, it is necessary to design coding sequences based on a distributed genetic algorithm and conduct extensive tests to find the optimal coding sequence. Derived sequences coding scheme also does not require additional hardware overhead. However, this scheme introduces high computational complexity in the decoding stage, as it relies on computationally expensive matrix operations to generate and process derived sequences.

EAC-coding scheme proposed in this paper incorporates a novel preprocessing framework for autocorrelation characteristic analysis, waveform analysis and reconstruction of Raman backscattered signals. This framework significantly improves the fidelity and transmission stability of the encoded pulse sequence by dynamically compensating for the transient gain fluctuation of EDFA. This technology leverages the inherent gain of EAC-coding pulse coding to effectively offset fiber scattering attenuation during 70.0 km long-distance transmission, achieving a significant improvement in the baseline SNR. On this basis, a two-stage noise suppression method is further constructed. The first-stage processing involves compensating for signal attenuation using the theoretical coding gain of EAC-coding. For the second-stage processing, a Haar wavelet denoising algorithm is adopted. Its step-matching characteristic enables accurate extraction of the abrupt temperature change features of the sensing fiber and adaptive filtering of the residual time/frequency domain

coupled noise in the decoded signal. This three-fold synergistic mechanism, consisting of the preprocessing framework compensating for EDFA fluctuation, coding gain suppressing transmission losses, and wavelet transform finely removing residual noise, breaks through the theoretical trade-off bottleneck between SNR and sensing distance in traditional schemes. Experiment results indicate that, at an ultra-long distance of 70.0 km, this scheme can achieve a spatial resolution of 1.58 m, a temperature resolution of 5.39 °C, and an effective number of sensing points reaching 44,303, verifying the engineering feasibility of the scheme. To the best of our knowledge, the number of effective sensing points achieved by this scheme ranks the highest in the field of Raman distributed optical fiber sensing systems.

Responds to the reviewers’ comments:

Reviewer #3 (Comments to the Author):

Recommendations for the Author(s):

The response seems reasonable. Only remaining issue is the overhead in terms of the time taken for the processing. Perhaps the authors can include that in the manuscript.

Reply: We sincerely appreciate your positive feedback and valuable suggestion on our manuscript. Your recognition that our responses are reasonable is truly encouraging, and we are grateful for your thoughtful point regarding the time costs—this is a critical aspect we will promptly address.

Following your valuable suggestion, we will incorporate relevant content related to time costs into the manuscript, ensuring the information is clear, detailed, and aligned with the study’s completeness. Details are as follows.

The current single measurement time is 252 minutes, limited by a pulse period of 700 μs and a number of averages of three million, with a theoretical measurement time of $35\text{ min} \times 4 = 140\text{ min}$. Due to the massive data volume exceeding the on-board real-time processing limit of the data acquisition card, the measurement time is prolonged. In the future, a high-performance data acquisition card will be upgraded and the data processing pipeline optimized, thereby eliminating the delay degradation caused by hardware bottlenecks.

- 1) In the section of **Discussion, page 13, right column, paragraph 1, line 1**, add “In terms of acquisition time cost, the current measurement time is 252 minutes, limited by a pulse period of 700 μs and a number of averages of three million, with a theoretical measurement time of $35\text{ min} \times 4 = 140\text{ min}$. Due to the massive data volume exceeding the on-board real-time processing limit of the DAC, the measurement time is prolonged. In the future, a high-performance DAC will be upgraded and the data processing pipeline optimized, thereby eliminating the delay degradation caused by hardware bottlenecks.”

Corrections:

We would like to express our heartfelt thanks to all reviewers for the careful review of our manuscript and your valuable comments amid your busy schedules. The professional suggestions you provided have offered important guidance for improving the quality of the manuscript, and we would like to extend our sincere gratitude for this.

To clearly present the full picture of revisions to the manuscript and facilitate the reviewers' verification of the implementation of these revisions, we have summarized all the revisions and supplements made based on the reviewers' comments.

Once again, we appreciate the attentive guidance from all reviewers.

Thank you for your suggestion. We have made the following modifications in our revised manuscript:

- 1) In the section of **Introduction**, the following content has been relocated to the Supplementary Information. “Simplex coding scheme, as a linear coding technique based on Hadamard matrix transformation, has its code length (in L bits) determining the number of detection signals required (L groups). For instance, in 2006, J. Park, et al. achieved a spatial resolution of 17.0 m, a temperature resolution of 3.0 °C, and a total of 2176 effective sensing points over a sensing distance of 37.0 km by utilizing this coding scheme and link optimization techniques. The main limitation of this scheme lies in the linear growth of data acquisition volume and computational complexity with code length, which restricts its practical efficiency in applications requiring long code lengths and high-speed sensing.

Low-repetition-rate cyclic pulse coding scheme generates Simplex coding-based cyclic coding sequences using an acousto-optic modulator (AOM), enabling pulses to be injected into the optical fiber periodically at a low repetition rate. For example, in 2011, M. A. Soto, et al. combined the low-repetition-rate quasi-periodic cyclic pulse coding with a high-power fiber laser, achieving a spatial resolution of 1.0 m, a temperature resolution of 3.0 °C, and 26,000 effective sensing points over a sensing distance of 26.0 km. However, higher pulse energy also makes it more likely to induce fiber nonlinear effects, and when approaching the stimulated scattering threshold, it restricts the system's sensing distance and SNR performance.

Pre-Shaped Simplex Coding scheme is based on simplex coding technology and adopts a linearly increasing profile to adjust pulse amplitude, compensating for the amplitude variation caused by EDFA to suppress the transient effect of the EDFA and improve system performance. For example, in 2017, J. B. Rosolem, et al. achieved a spatial resolution of 10.0 m, a temperature resolution of 8.4 °C, and

6,200 effective sensing points over a sensing distance of 62.0 km by using the pre-shaped Simplex coding and a gain-controlled erbium-doped fiber amplifier. Although this scheme effectively suppresses the transient response of the EDFA at the hardware level, its hardware structure is more complex, leading to a significant increase in implementation cost.

Genetic-optimised aperiodic coding scheme utilizes a distributed genetic algorithm (DGA) to generate a single-sequence aperiodic code (GO-code), which is then converted into an optical pulse sequence and injected into the optical fiber. For example, in 2020, X. Z. Sun, et al. achieved a spatial resolution of 1.0 m, a temperature resolution of 3.9 °C, and 39,000 effective sensing points over a sensing distance of 39.0 km based on the Genetic-optimised aperiodic coding scheme. The core challenge of this scheme lies in its high dependence on optimization algorithms to search for the optimal code, which typically requires substantial computational resources and iterative testing.

Derived Sequences coding scheme addresses the transient effects of EDFA through a derived sequence decoding method, thereby enhancing the performance of long-distance sensing. For instance, in 2022, D. D. Chai, et al. achieved a spatial resolution of 5.0 m, a temperature resolution of 2.5 °C, and 8,800 effective sensing points over a sensing distance of 44.0 km based on the Derived Sequences coding scheme. However, this scheme's decoding process imposes a heavy computational burden, and the core algorithm involves high-complexity matrix operations, making it difficult to meet real-time requirements.”

- 2) In the section of **Discussion**, the following content has been relocated to the Supplementary Information. “Currently, the pulse coding techniques applied in Raman distributed optical fiber sensing system mainly include the following: Simplex coding scheme, Pre-Shaped Simplex coding scheme, Low-repetition-rate cyclic pulse coding scheme, Genetic-optimised aperiodic coding scheme, and Derived Sequences coding scheme. Their key performance indicators are presented in Table 2.

Scheme	Sensing Distance	Spatial Resolution	Sensing Points	Temperature Resolution	Measurement Time
Simplex coding ^[31]	37.0 km	17.0 m	2176	3.0 °C	/
Low-repetition-rate cyclic pulse coding ^[27]	26.0 km	1.0 m	26000	3.0 °C	30 s
Pre-Shaped Simplex Coding ^[32]	62.0 km	10.0 m	6200	8.4 °C	/
Genetic-optimised aperiodic coding ^[29]	39.0 km	1.0 m	39000	3.9 °C	13.6 min
Derived Sequences coding ^[33]	44.0 km	5.0 m	8800	2.5 °C	/
EAC-coding	70.0 km	1.58 m	44303	5.39 °C	252 min

Sensing Points refers to the number of independent spatial sampling points in the fiber optic sensing link where the system can achieve effective measurements. The specific calculation method is the ratio of the sensing distance to the spatial resolution.
Temperature resolution is defined as the minimum temperature change that a system can reliably distinguish, and its calculation method is based on the standard deviation under steady-state conditions.

We have compared our proposed scheme with other coding methods such as simplex coding and Genetic-optimised aperiodic coding in terms of coding gain, measurement time, and system complexity. The specific revisions are as follows.

In terms of coding gain, the EAC-coding scheme proposed in this paper exhibits significant advantages in SNR improvement. The core of this scheme lies in the analysis and accurate reconstruction of the waveform of Raman backscattered signals, which effectively compensates for the detection signals distortion caused by the transient effects of the system. This mechanism enables EAC-coding to overcome the performance bottleneck of traditional derived sequences coding schemes induced by signals distortion, achieving a coding gain closer to the theoretical limit. Specifically, existing derived sequence coding scheme construct coding sequences based on Golay complementary sequences and perform correlation decoding using their derived sequences. The upper limit of their theoretical coding gain is $\sqrt{L}/2$ (where L denotes the length of the coding sequence). However, due to the unavoidable distortion (especially transient distortion) in actual detection signals, the results of their correlation operations deteriorate, leading to a significant gap between the actually achieved coding gain and this theoretical value. In contrast, the EAC-coding scheme, through the aforementioned signals analysis and reconstruction mechanism, significantly suppresses the impact of transient distortion on the detection signals waveform, thereby enabling its coding gain to approach the theoretical upper limit of $\sqrt{L}/2$. Benefiting from this, the proposed system ultimately achieves temperature sensing capability in a long-distance Raman distributed optical fiber sensing system with a sensing distance of 70.0 km.

In terms of measurement time, the core advantage of the EAC-coding scheme stems from its extremely low decoding computational complexity. This scheme achieves signals reconstruction solely through Fourier transform, which significantly reduces the computational cost and time delay of real-time data processing, thereby effectively improving the overall response speed of the system.

Specifically, for a Simplex code with length L, it is necessary to inject L groups of detection signals into the optical fiber and collect L groups of Raman backscattered signals, with its theoretical minimum acquisition time being $T_{\text{Simplex}}=L \times t$ (where t denotes the acquisition time of one group of Raman backscattered curves). Although the Genetic-optimised aperiodic coding scheme only requires injecting one group of detection signals into the optical fiber, it needs additional time to design the coding sequence based on the distributed genetic algorithm. The Derived sequences coding scheme requires injecting 4 groups of detection signals into the optical fiber, with a theoretical acquisition time of $4t$. However, in the decoding stage, it involves complex steps such as extracting the attenuation envelope, establishing a compensation model, and solving the compensation envelope via matrix operations, which increases the time cost of the decoding process. The EAC-coding scheme proposed in this paper performs encoding based on Golay complementary sequences, with a theoretical acquisition time of $4t$, and only needs Fourier transform to complete the analysis and reconstruction of the Raman backscattered signals waveform. This significant efficiency improvement at the algorithm level greatly reduces the real-time data processing delay and effectively enhances the system response speed, providing crucial real-time guarantee for the final realization of temperature sensing in a 70.0 km long-distance Raman distributed optical fiber sensing system.

In terms of system complexity, the core innovation of the EAC-coding scheme lies in achieving a high degree of simplification in the decoding process through algorithm optimization, while keeping the hardware architecture unchanged. This scheme only needs to perform efficient Fourier transform operations to complete signal reconstruction, which significantly reduces the algorithmic complexity and computational burden of real-time data processing. The specific reasons are as follows. Simplex coding scheme requires the injection of L groups of detection signals, resulting in a long acquisition time and a large volume of raw data. This not only imposes high requirements on hardware performance (such as data acquisition rate and storage capacity) and long-term stability, but also the large-scale data processing involved in its decoding process significantly increases the computational complexity. Genetic-optimised aperiodic coding scheme greatly shortens the acquisition time without increasing hardware costs. However, due to its unique design, it is necessary to design coding sequences based on a distributed genetic algorithm and conduct extensive tests to find the optimal coding sequence. Derived sequences coding scheme also does not require additional hardware overhead. However, this scheme introduces high computational complexity in the

decoding stage, as it relies on computationally expensive matrix operations to generate and process derived sequences.

EAC-coding scheme proposed in this paper incorporates a novel preprocessing framework for autocorrelation characteristic analysis, waveform analysis and reconstruction of Raman backscattered signals. This framework significantly improves the fidelity and transmission stability of the encoded pulse sequence by dynamically compensating for the transient gain fluctuation of EDFA. This technology leverages the inherent gain of EAC-coding pulse coding to effectively offset fiber scattering attenuation during 70.0 km long-distance transmission, achieving a significant improvement in the baseline SNR. On this basis, a two-stage noise suppression method is further constructed. The first-stage processing involves compensating for signal attenuation using the theoretical coding gain of EAC-coding. For the second-stage processing, a Haar wavelet denoising algorithm is adopted. Its step-matching characteristic enables accurate extraction of the abrupt temperature change features of the sensing fiber and adaptive filtering of the residual time/frequency domain coupled noise in the decoded signal. This three-fold synergistic mechanism, consisting of the preprocessing framework compensating for EDFA fluctuation, coding gain suppressing transmission losses, and wavelet transform finely removing residual noise, breaks through the theoretical trade-off bottleneck between SNR and sensing distance in traditional schemes. Experiment results indicate that, at an ultra-long distance of 70.0 km, this scheme can achieve a spatial resolution of 1.58 m, a temperature resolution of 5.39 °C, and an effective number of sensing points reaching 44,303, verifying the engineering feasibility of the scheme. To the best of our knowledge, the number of effective sensing points achieved by this scheme ranks the highest in the field of Raman distributed optical fiber sensing systems.

- 3) In the section of **Discussion, page 13, right column, paragraph 1, line 1**, add “In terms of time cost, the current single measurement time is 252 minutes, limited by a pulse period of 700 μs and a number of averages of three million, with a theoretical measurement time of $35 \text{ min} \times 4 = 140 \text{ min}$. Due to the massive data volume exceeding the on-board real-time processing limit of the data acquisition card, the measurement time is prolonged. In the future, a high-performance data acquisition card will be upgraded and the data processing pipeline optimized, thereby eliminating the delay degradation caused by hardware bottlenecks.”

Review of Manuscript titled

“Overcoming the Longstanding Challenge of Long-Range Raman Distributed Optical Fiber Sensing Through Golay-Encoded Autocorrelation and Waveform Reconstruction”

It is well known that Distributed Anti-Stokes Raman Thermometry (aka Raman OTDR) is an attractive approach for distributed temperature sensing without much cross-sensitivity due to strain (unlike Brillouin distributed sensors). A key challenge for such a Raman OTDR is its operation in the spontaneous Raman scattering regime, which severely limits the peak power that can be used thereby limiting the sensing range to 20 km.

The use of a coded sequence of pulses is a well-established approach in Raman OTDR to increase the backscattered energy without losing spatial resolution. Specifically, Golay encoded sequences are attractive since they use a complementary pair of sequences that theoretically eliminates side lobes leading to excellent spatial resolution and low temperature uncertainty. In this work, the authors present their efforts on utilizing this idea to extend the measurement range to a distance of 70 km which seem impressive at first sight.

However, the above manuscript is not recommended to be published in Nature Communications in its current form due to the following reasons:

1. There is a fundamental question on the improvement achieved in this work compared to previous work. The authors claim that their work “introduces Golay complementary sequences into Raman distributed optical fiber sensing”, which is not true. In fact, it is surprising that they are not aware of previous work on this (10.1109/JSEN.2022.3229417) from the same University. In general, the authors need to present a thorough literature survey of previous work in this area and provide a clear justification of how their work is different from the previous approaches/results.
2. The authors explain the process of reconstructing the Raman backscattered signals to eliminate transient effects in Sec. 2, but it would be better to demonstrate experimental proof of their approach. The results shown in Fig. 4 are alluding to this, but not very convincing.
3. The improvement claimed in Fig. 4 is with respect to RBS decoded using “destroyed detection” signals, but the comparison presented for the Raman OTDR (Figs. 5-7) is with respect to single pulse Raman OTDR. Why is that? How about comparing with the uncompensated RBS?
4. If the key contribution of this work is the compensation for transients during the amplification process, then the comparison of results should be between the uncompensated case and the compensated case.
5. What do the authors mean by “destroyed detection” signals? Needs to be explained properly.
6. In Fig. 6(a), the EAC scheme is shown to provide 8.75 dB improvement over the single pulse scheme. How is this 8.75 dB explained? For a 64 bit code, we expect an SNR improvement of $\sqrt{64}/2 = 4$, which is 6 dB.
7. Also, if we check the difference between the two curves at 28 km, the improvement seems to be more than 10 dB. How can you explain this?
8. What is being plotted in Fig. 7(a)? Absolute temperature or uncertainty in temperature? I presume it is the latter, in which case it should be clearly mentioned. How is 6x improvement achieved? Need proper explanation.